# Long-term memory plasticity in a decade-long connectivity study post anterior temporal lobe resection

Marine N. Fleury [1,2] ✉, Lawrence P. Binding[1,2,3], Peter Taylor[1,4],
Fenglai Xiao [1,2] ✉, Davide Giampiccolo[1,5,6], Sarah Buck[1,2], Gavin P. Winston[1,2,7],
Pamela J. Thompson[1,8], Sallie Baxendale[1,8], Andrew W. McEvoy[1,5],
Matthias J. Koepp[1,2], John S. Duncan [1,2] & Meneka K. Sidhu[1,2]

Approximately 40% of individuals undergoing anterior temporal lobe resection for temporal lobe epilepsy experience episodic memory decline. There has been a focus on early memory network changes; longer-term plasticity and its impact on memory function are unclear. Our study investigates neural mechanisms of memory recovery and network plasticity over nearly a decade post-surgery. We assess memory network changes, from 3–12 months to 10 years postoperatively, in 25 patients (12 left-sided resections) relative to 10 healthy matched controls, using longitudinal task-based functional MRI and standard neuropsychology assessments. We observe key adaptive changes in memory networks of a predominantly seizure-free cohort. Ongoing neuroplasticity in posterior medial temporal regions and contralesional cingulum or pallidum contribute to long-term verbal and visual memory recovery. Here, we show the potential for sustained cognitive improvement and importance of strategic approaches in epilepsy treatment, advocating for conservative surgeries and long-term use of cognitive rehabilitation for ongoing recovery.

In medication-refractory focal epilepsy, seizures are associated with progressive cognitive decline over multiple domains, including memory and executive functions[1–3]. Surgical resection can bring seizure control[1,4] which is the strongest predictor of cognitive remission, regardless of the effect of surgery on function[1,2]. Anterior temporal lobe resection (ATLR) is the most commonly performed surgical treatment for refractory temporal lobe epilepsy (TLE)[5]. This involves the partial removal of temporal neocortical and mesial temporal structures including the anterior hippocampus, parahippocampal and fusiform gyri; structures critical for successful memory formation[6,7].

Although ATLR may prevent the progressive decline observed in refractory epilepsy, up to 40% of individuals experience significant episodic memory deficits postoperatively[5,8]. Identifying regions crucial for effective memory reorganization that could be spared in surgery may help mitigate long-term post-operative memory loss. Additionally, understanding post-operative plasticity could assist the development of strategies to promote long-term memory recovery.

Longitudinal functional MRI (fMRI) studies have highlighted plasticity of episodic memory function up to 2 years following ATLR[9,10]. Task-based fMRI research primarily compared mid-term (up to 2 years) postoperative memory activations with preoperative or short-term

¹Department of Clinical and Experimental Epilepsy, Queen Square Institute of Neurology, UCL, London WC1N 3BG, UK. ²MRI Unit, Epilepsy Society, Chalfont St Peter SL9 0RJ, UK. ³Department of Computer Science, Centre for Medical Image Computing, UCL, London WC1V 6LJ, UK. ⁴CNNP lab, Interdisciplinary Computing and Complex BioSystems Group, School of Computing Science, Newcastle University, Newcastle NE4 5TG, UK. ⁵Victor Horsley Department of Neurosurgery, National Hospital for Neurology and Neurosurgery, Queen Square, London WC1N 3BG, UK. ⁶Department of Neurosurgery, Institute of Neurosciences, Cleveland Clinic London, London W1T 4AJ, UK. ⁷Department of Medicine, Division of Neurology, Queen's University, Kingston K7L 3N6, Canada. ⁸Psychology Department, Epilepsy Society, Chalfont St Peter SL9 0RJ, UK. ✉e-mail: marine.fleury.20@ucl.ac.uk; f.xiao@ucl.ac.uk

**Table 1 | Demographics and clinical details of people who had anterior temporal lobe resection and controls**

| | Sex M (F) | Age median (IQR) | Duration median (IQR) | Seizure frequency median (IQR) | ILAE seizure outcome | | Anti-seizure medications (ASM) | |
|---|---|---|---|---|---|---|---|---|
| | | | | | T2 | T3 | T2 to T3 | T3 |
| Left ATLR (n = 12) | 7 (5) | 38 (13) | 12.5 (16) | 6 (0) | Class 1-2: 10 Class 4-5: 2 | Class 1: 10 Class 3-5: 2 | Less/off: 6 Same: 4 Extra 1-2: 2 | None: 4 1-2 ASMs: 7 3 ASMs: 1 |
| Right ATLR (n = 13) | 4 (9) | 38 (21) | 16 (21) | 10 (5) | Class 1: 11 Class 2-3: 2 | Class 1: 12 Class 2: 1 | Less/off: 7 Same: 6 | None: 6 1-2 ASMs: 6 3 ASMs: 1 |
| Controls (n = 10) | 4 (5) | 37 (23) | NA | NA | NA | NA | NA | NA |
| P-value | 0.40 | 0.80 | 0.37 | 0.91 | 0.48 | 0.44 | 0.76 | 0.77 |

Statistical difference between groups is presented, for gender using two-sided Fisher exact p-value, and with Kruskal–Wallis p-values for age, duration, and frequency (all at time of surgery), ILAE outcome and ASM intake. Median (IQR) age and duration at the time of surgery are presented in years, while seizure frequency in the group median per months.

*ASM* anti-seizure medications, *ATLR* anterior temporal lobe resection, *ILAE* International League Against Epilepsy, *IQR* interquartile range, *F* females, *M* males, *N* number, *NA* not applicable, *T2* 3–12-month follow-up (median 11 months), *T3* 10-year follow-up (median 9 years).

(3 months) post-surgical activations[9–12]. These showed that increased engagement of contralesional hippocampus and neocortex, including the insula and orbitofrontal cortex (OFC), was supportive of postoperative memory function[9,11,13].

Network-level analyses however have only been performed on resting-state fMRI[14,15]. Task-based fMRI connectomics is needed to comprehensively model postoperative changes in the functional architecture of episodic memory and its association with memory function following ATLR. This comes in light of recent evidence of distributed functional[9,12,16] and structural alterations[17,18] before and after ATLR, together with postoperative deficits across both verbal and non-verbal memory domains, regardless of surgical laterality[1].

Network studies using psychophysiological interaction (PPI)[19] analysis model the changes in functional couplings between a seed and whole-brain regions specific to task performance. Using PPI, we showed widespread functional connectivity increases in TLE between bilateral medial temporal lobes (MTLs) and to contralesional temporal and extra-temporal regions that supported better memory[16]. Investigating longitudinal connectivity changes specific to the memory connectome would inform on the network's potential for adaptive plasticity beyond the two-year postoperative mark.

A critical research question is whether memory functional plasticity after surgery represents a short-term response to surgical insult, or is an ongoing, dynamic process that evolves over time. If plasticity is ongoing, identifying patterns of memory-node abnormalities in longitudinal studies may shed light on the biology of rehabilitation and aid stratifying patients for optimal cognitive strategies. Additionally, comprehending the long-term memory outcome and its supporting network is vital for effective pre-operative counseling of people with epilepsy (PWE) and their families.

This study aimed to identify neural mechanisms underlying verbal and visual memory network plasticity and recovery long-term after epilepsy surgery. In a longitudinal design, we (1) assessed changes in task-associated functional connectivity from 3–12 months to 10 years after ATLR using generalized PPI, and (2) correlated memory network alterations with improvement in episodic memory function. We hypothesized that there will be continued long-term plasticity of episodic memory networks[9,20], particularly in people who are seizure-free post-operatively[1]. Given our previous research and short-term plasticity effects, changes in local MTL connectivity including connections to the fusiform gyrus are of a priori interest[9,13,20].

In this work, we show cognitive stabilization or improvement 10 years post-operation, alongside the enduring adaptability of memory networks. Longitudinal connectivity increases within posterior medial temporal regions and with the cingulum and globus pallidus support long-term verbal and visual memory recovery, endorsing the importance of tailored surgeries.

## Results

The study cohort was followed-up 3 months, 12 months, and 10 years after ATLR (and similar in controls). Short-term and long-term respectively denote a combined 3–12-month (median = 11, interquartile range (IQR) = 8-12) timepoint and a 10-year (median = 9, IQR = 8-10) follow-up, respectively.

### Subjects

Nine of 10 healthy controls were right-handed, and one left-handed, as were 10 of 12 in the left ATLR group, two left-handed. In the right ATLR group, 11 of 13 were right-handed (1 ambidextrous and 2 left-handed). Patient demographics and clinical features at 3–12 months and 10 years are summarized in Table 1. All subjects who underwent left ATLR were MRI positive; 10/12 had hippocampal sclerosis (HS) preoperatively, 1 cavernoma, and 1 ependymoma. Among right ATLR subjects, 11/13 were MRI positive; 8 had HS, 1 cavernoma, 1 dysembryoplastic neuroepithelial tumor, and 3 had gliosis (only evident on pathology in 2). Long-term postoperatively, complete seizure-freedom (ILAE outcome 1)[21] was observed in 83% of left and 92% of right ATLR.

There were no significant differences in age and sex between patient and control groups (Kruskal-Wallis, two-sided Fisher exact tests; Table 1). There was no significant difference between both resection groups in clinical features (Kruskal-Wallis tests).

### Long-term changes in postoperative memory

Longitudinal changes in memory were determined as clinically significant based on a reliable change index (RCI), with 95% confidence intervals.

From 3–12 months to 10 years, healthy controls' memory was as follows: for verbal memory, 12.5% improved, 12.5% no change, while 75% declined. For visual memory, 12.5% improved, 50% no change, while 37.5% declined. At the group-level, there was a clinically significant improvement in verbal memory from baseline to 3–12 months assessments, which was reversed from 3–12 months to 10 years (i.e., significant decline). For visual memory, group-level performance remained stable over time.

In left ATLR, clinically significant changes in postoperative memory from 3–12-month to 10-year assessments were as follows: for verbal memory, 50% improved, 17% showed no change, while 33% declined. For visual memory, 42% improved, 8% remained stable, while 50% declined. At the group-level, verbal memory performance was stable from preoperative to 3–12-month follow-ups, and there was a clinically significant improvement from 3–12-months to 10 years. Comparing the 10-year follow-up to preoperatively, 75% of left ATLR patients demonstrated positive outcomes in verbal memory, with 33% returning to baseline performance and 42% showing improved memory. Visual memory significantly improved from

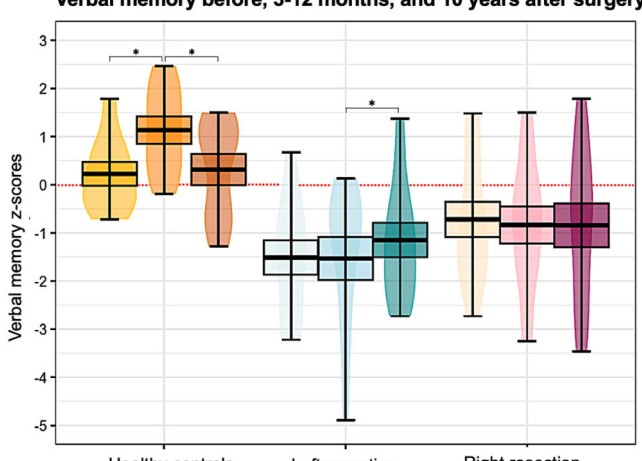

**Fig. 1 | Verbal and visual memory performance in each resection group and healthy controls at the preoperative, 3–12-month, and 10-year follow-ups.** Box plots display the mean (center line), the 25th–75th percentiles (bounds of the box), and the minimum and maximum values (whiskers). Violin plots illustrate the data distribution. Data are presented as mean ± standard error of the mean for memory z-scores from list and design learning tests (verbal and visual memory, respectively). This was assessed preoperatively, at 3–12 months, and at 10 years in people with epilepsy, and at similar intervals in controls. Memory improvement or decline is determined using a reliable change index with a 95% confidence interval. $N = 10$ controls, 12 left resections, 13 right resections.

preoperative to 3–12-month assessments and remained stable to the longer term (Fig. 1).

In the right ATLR, clinically significant changes were as follows: for verbal memory, 31% improved, 46% no change, and 23% declined. For visual memory, 42% improved, 17% no change, and 41% declined. At the group-level from preoperative to 3–12-month assessments, there was an initial visual memory decline that resolved in the longer term: group average increased from 3–12 months to 10 years that did not reach significance (Fig. 1 and Table 2). Ten years postoperatively relative to preoperatively, 58% of right ATLR patients showed positive outcomes in visual memory, with 33% returning to baseline performance and 25% exhibiting better memory. For verbal memory, group-level performance remained stable over time.

## Group differences in memory outcome
Longitudinal Group*Time interaction effects on memory function were assessed using linear mixed-effect models, and the impact of preoperative HS pathology, ASM cessation and seizure-freedom at 10 years using Fisher-Pitman permutation tests. Group averages in memory z-scores and statistics are shown Table 2. Supplementary information (SI) details preoperative neuropsychology, confidence intervals, and effects of clinical factors.

**Group differences.** At both 3–12-month and 10-year assessments, healthy individuals' verbal memory was significantly better than that of people in left ATLR, ($t$(40.83) = 4.64, $p = 0.0002$, β = 2.7, 95% CI = [1.56, 3.76]) and right ATLR groups ($t$(40.83) = 3.49, $p = 0.0035$, β = 2.0, 95% CI = [0.89, 3.05]). For visual memory, controls performed significantly better than individuals in the right ATLR ($t$(52.69) = 3.54, $p = 0.005$, β = 1.5, 95% CI [0.68, 2.27]) but not left ATLR group ($t$(52.69) = 1.51, $p = 0.27$, β = 0.63, 95% CI [−0.16, 1.42]). Across postoperative time-points, there was no significant difference in verbal or visual memory z-scores between ATLR groups (see SI).

**Interaction effect.** For verbal memory, the effect of time on memory z-scores differed between controls and left ATLR, with a greater memory change (increase) over time in left ATLR than controls ($t$(32) = 2.79, $p = 0.016$, β = 1.13 95% CI = [0.35, 1.90]). There was no significant Group*Time interaction effect between controls and the right ATLR group for verbal memory ($t$(32) = 1.87, $p = 0.071$, β = 0.74, 95% CI = [−0.02, 1.50]), and between controls and either ATLR group for visual memory (left ATLR: t(31.00)= −0.05, $p = 0.96$, β = −0.020, 95% CI [−0.88, 0.84]; right ATLR: $t$(31.00)= −0.85, $p = 0.60$, β =−0.38, 95% CI [−1.25, 0.48]).

**Impact of clinical factors.** Continued ASM medication at 10 years and HS pathology adversely impacted long-term memory functions. This included worse visual memory recovery in people who continued ASMs (Exact Fisher–Pitman permutation test: $Z = 2.08$; $p = 0.036$), and worse 10-year verbal and visual memory in those with compared to those without HS (verbal memory: Z = −2.70, $p = 0.0045$; visual memory: $Z = −2.11$, $p = 0.029$).

## Differences in resection extent
There was no significant difference in the extent of hippocampus resection (Mann–Whitney tests, $U = 91$, $p = 0.13$, Rank-Biserial Correlation ($r$) = 0.31), and of anterior temporal lobe structures' resection ($U = 83$, $p = 0.32$, r = 0.21) between left and right ATLR groups. Figure 2 illustrates the preoperative and 10-year volumes of the surgically spared, remnant hippocampus in each resection group.

Correspondingly, linear regressions showed no predictive effect of the remnant hippocampus volume at 10 years on verbal and visual memory recovery from 3 to 12 months to 10 years, irrespective of memory type and surgical laterality; verbal memory, left ATLR: $t$(10)= −0.73, $p = 0.48$, β = −0.0003, 95% CI [−0.0004, 0.0008], right ATLR: $t$(11) = 0.66, $p = 0.53$, β = 0.0002, 95% CI [−0.001, 0.0005]; visual memory, left ATLR: $t$(10) = 2.0, $p = 0.08$, β = 0.0006, 95% CI [−7.8 × 10⁻⁵, 0.001], right ATLR: $t$(11) = −0.85, $p = 0.42$, β = −0.0003, 95% CI [−0.001, 0.0005]. This lack of predictive effect is possibly associated with the non-significant variations in the extents of hippocampus resection.

## Changes in functional connectivity 3–12 months to 10 years post-surgery relative to changes in controls
Table 3 shows detailed results from the flexible factorial design analysis.

For successful word encoding in left ATLR compared to controls, there was increased connectivity from the remnant MTL seed with both bilateral posterior parahippocampal and fusiform gyri from

**Table 2 | Neuropsychometry measures across experimental assessments in healthy controls and in left or right resection groups**

|  | Preoperative IQ | | Verbal memory z-scores | | | Visual memory z-scores | | |
|---|---|---|---|---|---|---|---|---|
|  | **VIQ** | **PIQ** | **T1** | **T2** | **T3** | **T1** | **T2** | **T3** |
| Controls (n = 10) | 0.44 (0.69) FSIQ |  | 0.22 (0.78) | 1.13 (0.90) | 0.55 (1.01) | −0.43 (0.88) | 0.71 (0.58) | 0.83 (0.36) |
| Left ATLR (n = 12) | −0.74 (0.60) | −0.45 (0.49) | −0.72 (1.33) | −1.53 (1.55) | −0.38 (0.87) | 0.08 (0.84) | 0.08 (0.84) | 0.16 (1.04) |
| Right ATLR (n = 13) | −0.70 (0.91) | −0.45 (0.77) | −1.51 (1.22) | −0.84 (1.38) | 0.71 (0.50) | 0.71 (0.50) | −0.77(1.15) | −0.33(1.35) |
| P-value | 0.001 | 0.004 | 0.006 | 0.0002 | 0.029 | 0.003 | 0.002 | 0.076 |

In controls, full scale IQ (FSIQ) is reported, while in patients, premorbid IQ was assessed via the NART-2; verbal and performance IQ subtests evaluate verbal and non-verbal reasoning. Clinically meaningful changes in z-scores are calculated based on the reliable change index (RCI), using a 95% confidence interval. For verbal memory from 3–12 months to 10 years, this meant an RCI improvement of ≥ 0.31 and a RCI decline of ≥ 0.40. For visual memory from 3–12 months to 10 years, an RCI improvement of ≥0.58 and RCI decline of ≥0.16 were deemed clinically significant. All values are shown as mean (standard deviation). P-values are reported for each one-way ANOVA.

ATLR anterior temporal lobe resection, N number; T2 3–12-month follow-up (median 11 months), T3 10-year follow-up (median 9 years), VIQ verbal IQ, PIQ performance IQ.

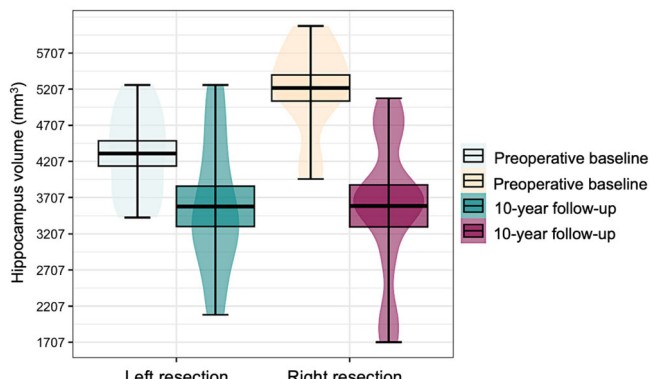

**Fig. 2 | Volume of the epileptogenic hippocampus before and 10 years after anterior temporal lobe resection.** Box plots display the mean (center line), the 25th–75th percentiles (bounds of the box), and minimum and maximum values (whiskers). Violin plots illustrate the data distribution. Data are presented as mean ± standard error for each group at preoperative and 10-year follow-up time points. N = 12 left resections, 13 right resections.

3–12 months to 10 years postoperatively. Extra-temporally, connectivity was longitudinally increased between the remnant MTL and both the right anterior cingulate cortex and bilateral occipital cortices.

For successful face encoding, there was enhanced functional connectivity between the remnant left MTL seed and posterior MTL regions, including the remnant parahippocampal and bilateral fusiform gyri, long-term compared to short-term after left ATLR. Neocortically, there was increased connectivity between the contralesional hippocampal seed and the right superior frontal gyrus from 3–12 months to 10 years.

For successful word encoding in right ATLR compared to controls, there was increased long-term functional connectivity compared to short-term, from the remnant right MTL seed with the remnant posterior parahippocampal and fusiform gyri, and extra-temporally with the left inferior parietal gyrus.

For successful face encoding, functional connectivity from 3–12 months to 10 years after right ATLR was enhanced mainly from the contralesional hippocampal seed; posteriorly with the remnant parahippocampal and bilateral fusiform gyri, and extra-temporally with the right inferior frontal and right supramarginal gyri. The remnant right MTL seed showed stronger functional couplings with the remnant posterior parahippocampus.

### Correlation between longitudinal changes in connectivity and memory recovery 3-12 months to 10 years postoperatively

Generalized PPI analyses inherently modeled memory-encoding networks underlying successful memory formation. Therefore, all reported results represent connectivity that is supportive of memory function. To look at neural substrates of more efficient memory recovery, correlations were performed, revealing the most adaptive longitudinal connectivity changes. Figure 3 and Table 4 present longitudinal changes in task-based connectivity significantly correlated with verbal memory improvement from 3–12 months to 10 years. Positive correlations for visual memory are reported Table 5, Fig. 4 in left ATLR, and Fig. 5 in right ATLR.

From 3–12 months to 10 years after left ATLR compared to controls, improvement in verbal memory was significantly correlated with increased functional connectivity between the remnant left MTL seed and both the right posterior parahippocampus and the remnant posterior fusiform gyrus. At the extra-MTL level, verbal memory improvement was significantly associated with enhanced functional couplings between the remnant MTL seed and the right anterior cingulate cortex, along with reduced connectivity with the left parietal gyrus, 10 years relative to 3–12 months post-surgery.

Visual memory improvement after left ATLR compared to controls correlated with longitudinal increases in connectivity between the remnant left MTL seed and bilateral posterior fusiform gyri, and between the contralesional hippocampal seed and both remnant hippocampus and fusiform gyrus, from short-term to long-term follow-ups. Neocortically, people with long-term improvement in visual memory exhibited a longitudinal increase in contralesional frontal and bilateral temporal connectivity; between the remnant MTL and bilateral temporal areas (right middle and left inferior), and between the contralesional hippocampal seed and the right anterior OFC and right superior frontal gyrus.

Similar to left ATLR, from 3–12 months to 10 years after right ATLR, verbal memory improvement was significantly correlated with increased connectivity with posterior MTL regions; from the remnant right MTL seed with the remnant posterior fusiform, and from the contralesional left hippocampal seed with the remnant parahippocampal gyrus. At the extra-MTL level, long-term verbal memory improvement was significantly supported by reduced connectivity between the remnant MTL and left amygdala, right thalamus, and right supramarginal gyrus.

Visual memory improvement after right ATLR compared to controls correlated with longitudinal increases in functional connectivity with the bilateral posterior MTL. This involved enhanced connectivity from the remnant right MTL seed with the remnant parahippocampal and contralesional fusiform gyri, and from the contralesional hippocampal seed with the contralesional parahippocampal and bilateral fusiform gyri. At the extra-MTL level, improvement in visual memory significantly correlated with increased functional connections between the contralesional hippocampal seed and the bilateral middle and inferior temporal gyri, contralesional basal ganglia (external

**Table 3 | Functional connectivity changes from 3–12 months to 10 years after anterior temporal lobe resection in people with epilepsy, over and above changes seen in controls**

| | Connectivity T2 to T3 | Left ATLR | | Right ATLR | |
|---|---|---|---|---|---|
| | | *Left MTL seed* | *Right MTL seed* | *Left MTL seed* | *Right MTL seed* |
| **Words remembered** | Connectivity increases | L post fusiform *(occ.)* −34 −84 −14 *P* = 0.014[a], *T* = 3.31 | *No suprathreshold connectivity* | *No suprathreshold connectivity* | R post fusiform *(temp-occ.)* 36 −42 −16 *P* = 0.018[a], *T* = 3.21 |
| | | L post parahippocampus −22 −30 −20 *P* = 0.022[b], *T* = 2.65 | | | R post fusiform *(occ.)* 34 −76 −14 *P* = 0.017[a], *T* = 3.24 |
| | | R post parahippocampus 18 −42 −8 *P* = 0.023[a], *T* = 3.06 | | | R post parahippocampus 36 −40 −12 *P* = 0.032[b], *T* = 2.45 |
| | | R post fusiform *(occ.)* 28 −72 −2 *P* = 0.043[a], *T* = 2.74 | | | L inf parietal G −62 −42 42, *P* < 0.001, *T* = 4.39 |
| | | R subgenual ACC 10 −28 −8 *P* < 0.001, *T* = 3.46 | | | |
| | | L inf occipital G −30 −92 −10 *P* < 0.001, *T* = 4.68 | | | |
| | | R sup occipital G 26 −82 32 *P* < 0.001, *T* = 3.71 | | | |
| | Connectivity decreases | *No suprathreshold connectivity* | *No suprathreshold connectivity* | *No suprathreshold connectivity* | *No suprathreshold connectivity* |
| **Faces Remembered** | Connectivity increases | L post parahippocampus −24 −38 −14 *P* = 0.021[b], *T* = 2.76 | Right sup frontal G 18 −10 60 *P* = 0.001, *T* = 3.69 | L post fusiform *(temporal)* −38 −26 −16 *P* = 0.034[a], *T* = 3.00 | R post parahippocampus. 28 −26 −22 *P* = 0.037[b], *T* = 2.44 |
| | | L post fusiform *(temp-occ.)* −20 −48 −16 *P* = 0.001[a], *T* = 4.36 | | L post fusiform *(occ.)* −36 −62 −10 *P* = 0.039[a], *T* = 2.71 | |
| | | R post fusiform *(temp-occ.)* 36 −62 −14 *P* = 0.012[a], *T* = 3.54 | | R post fusiform *(occ.)* 30 −70 −2 *P* = 0.011[a], *T* = 3.47 | |
| | | R mid fusiform *(temporal)* 32 −38 −16 *P* = 0.021[a], *T* = 3.25 | | R post fusiform *(temporal.)* 34 −38 −16 *P* = 0.021[a], *T* = 3.47 | |
| | | | | R post parahippocampus 36 −40 −12 *P* = 0.020[a], *T* = 2.78 | |
| | | | | R inf frontal gyrus 40 14 14 *P* < 0.001, *T* = 3.84 | |
| | | | | Right supramarginal G 54 −44 26 *P* = 0.001, *T* = 3.53 | |
| | Connectivity decreases | R sup temporal G 56 0 −14 *P* < 0.001, *T* = 3.77 | *No suprathreshold connectivity* | Left insula −34, 4, −12 *P* < 0.001, *T* = 4.78 | *No suprathreshold connectivity* |
| | | | | R sup temporal G 54 0 −14 *P* < 0.001, *T* = 3.72 | |

One-sided T-contrast results from two-way mixed ANOVAs using a flexible factorial design, with IQ as confound variable. Changes in functional connectivity across postoperative follow-ups are presented for postsurgical groups over and above changes in controls, seeding from the left and right medial temporal lobes (MTL). MTL-to-neocortex connectivity results are shown at *P* < 0.001 (uncorrected).

*ACC* anterior cingulate cortex, *ATLR* anterior temporal lobe resection, *Fusiform* fusiform gyrus, *FWE* Family Wise Error, *G* gyrus, *Inf* inferior, *LGN* lateral geniculate nucleus, *Mid* middle, *MTL* medial temporal lobe, *Post* posterior, *Occ* occipital, *Sup* superior; *T2* 3–12-month follow-up (median 11 months), *T3* 10-year follow-up (median 9 years).

[a]*P* < 0.05, FWE correction, using a 6 mm sphere for contralesional MTL regions.
[b]*P* < 0.05, FWE correction 3 mm sphere in remnant MTL regions to avoid resection cavity-related activations.

globus pallidus), right inferior frontal and right supramarginal gyri, 10 years compared to 3–12 months after right ATLR.

**Impact of hippocampal sclerosis on long-term plasticity effects**
Post-hoc two-sample t-tests evaluated the potential effect of HS on long-term memory plasticity. The longitudinal connectivity changes in people with HS were compared with those in people without HS, separately in each ATLR group for each MTL seed and subsequent memory fMRI task. Due to the maladaptive impact of HS on pre-operative network plasticity[16], we hypothesized that people with HS will have less network changes to regions supportive of memory functions from 3-12 months to 10 years. Results are reported within a binary thresholded mask of the longitudinal PPI's main effect, using the same thresholding methods as in above analyses. Refer

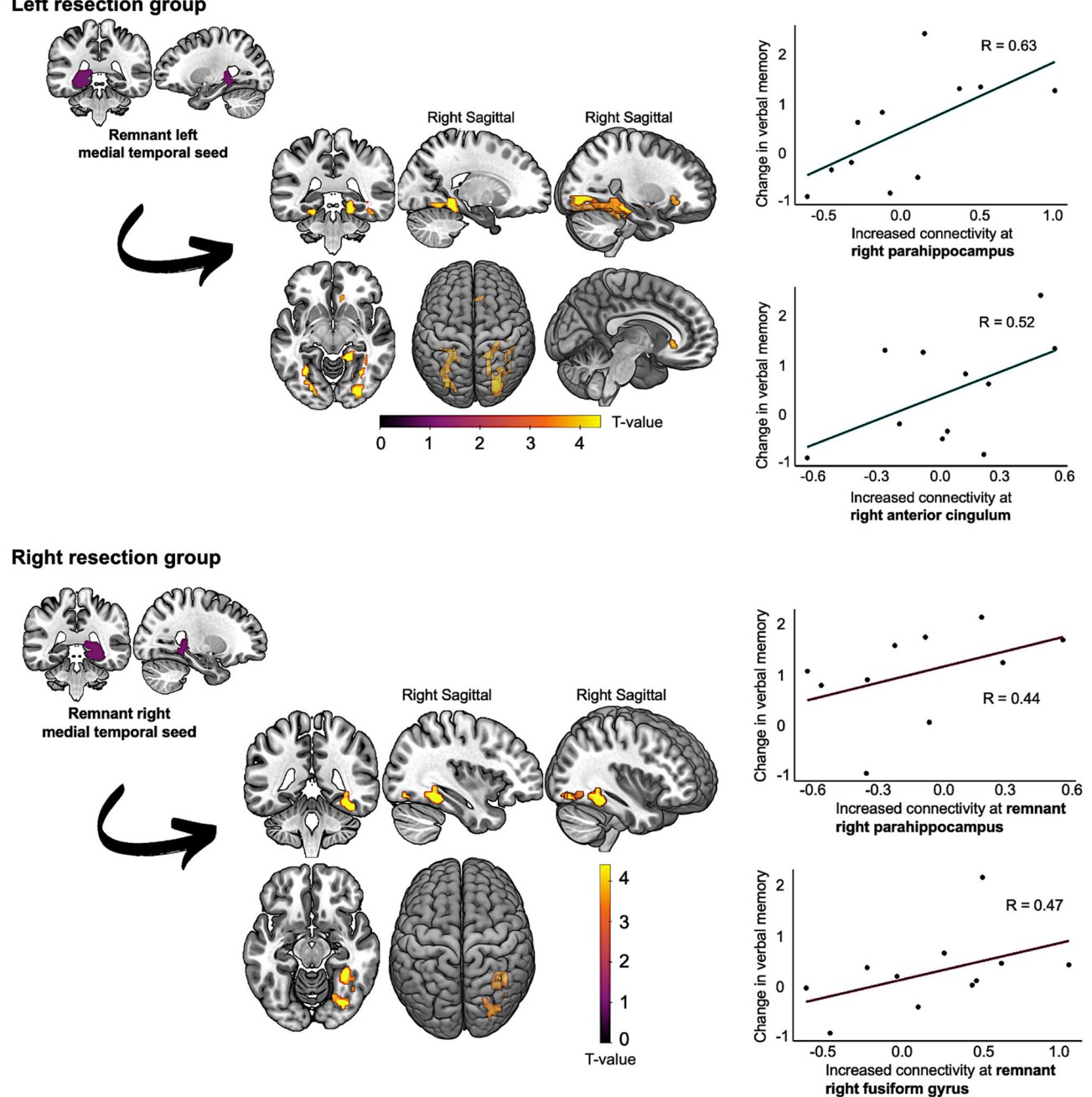

**Fig. 3 | Longitudinal functional connectivity changes correlated with post-operative verbal memory improvement from 3–12 months to 10 years after anterior temporal lobe resection.** Brain slices show longitudinal connectivity increases between the surgically spared MTL seed and whole-brain regions, correlating with verbal memory recovery. Top and bottom rows depict neuroplasticity after left and right-sided resections. Scatter plots present eigenvariates from one-sided T-contrasts of connectivity increases, plotted against verbal memory (list learning) changes. Top blue graphs represent adaptive connectivity increases with the contralesional parahippocampus ($P = 0.018$) and anterior cingulum ($P = 0.001$) after left-sided surgery; bottom purple graphs show connectivity increases with the remnant right parahippocampus ($P = 0.037$) and fusiform ($P = 0.022$) after right-sided surgery. All connectivity is presented controlling for medication and IQ, significant at $P < 0.001$ (uncorrected) in neocortical regions and $P < 0.05$ family-wise error small volume corrected in the MTL. MTL medial temporal lobe.

to SI for detailed methods and results with MNI coordinates and T-statistics.

For successful word encoding after left ATLR, individuals with HS compared to those without HS exhibited reduced plasticity (i.e., less longitudinal increases in connectivity) from 3-12 months to 10 years between the remnant MTL seed and the bilateral posterior fusiform gyrus. There was no difference in plasticity patterns from the contralesional hippocampus seed.

For successful face encoding after left ATLR, people with HS had reduced plasticity between the contralesional hippocampus seed and the remnant posterior fusiform gyrus. There was no difference in plasticity effects from the remnant seed.

**Table 4 | Correlations between long-term verbal memory improvement and longitudinal changes in task-based functional connectivity from 3–12 months to 10 years after anterior temporal lobe resection**

| | Left resection group | | Right resection group | |
|---|---|---|---|---|
| | *Left MTL seed* | *Right MTL seed* | *Left MTL seed* | *Right MTL seed* |
| Connectivity increases T2 to T3 | L post fusiform gyrus −34 −84 −14 $P = 0.008^a$, $T = 3.61$ | *No suprathreshold connectivity* | R post parahippocampus 36 −44 −4 $P = 0.037^b$, $T = 2.15$ | R post fusiform gyrus 36 −42 −16 $P = 0.024^a$, $T = 3.07$ |
| | L post fusiform gyrus −22 −34 −20 $P = 0.029^a$, $T = 2.96$ | | | R post fusiform gyrus 44 −42 −22 $P = 0.025^a$, $T = 3.04$ |
| | R post parahippocampus 16 −40 −8 $P = 0.018^a$, $T = 2.37$ | | | R post fusiform gyrus 32 −76 −14 $P = 0.022^a$, $T = 3.11$ |
| | R subgenual ACC 10 28 −8, $P = 0.001$, $T = 3.46$ | | | |
| Connectivity decreases T2 to T3 | L supramarginal gyrus −54 −22 42 $P = 0.001$, $T = 3.64$ | *No suprathreshold connectivity* | *No suprathreshold connectivity* | L amygdala −28 8 −20 $P = 0.042^a$, $T = 2.77$ |
| | | | | R thalamus 18 −28 −4 $P = 0.001$, $T = 3.58$ |
| | | | | R supramarginal gyrus 66 −30 26 $P = 0.001$, $T = 3.63$ |

Results of three-way ANCOVAs for each MTL seed during encoding of words subsequently remembered. These are the functional connectivity changes from short- to long-term follow-ups in people with epilepsy that were significantly correlated with improvement in verbal memory z-scores (i.e., from list-learning test) over that timeline. These were analyzed relative to connectivity changes and memory of healthy controls. MTL seed to extra-MTL connectivity is displayed at $P < 0.001$, uncorrected.

*FWE* family-wise error, *G* gyrus, *Inf* inferior, *Mid* middle, *MTL* medial temporal lobe, *Post* posterior, *sup* superior, *T2* = 3–12-month follow-up (median 11 months), T3 = 10-year follow-up (median 9 years).

a $P < 0.05$, FWE correction, using a 6 mm sphere for contralesional MTL regions.

b $P < 0.05$, FWE correction 3 mm sphere in remnant MTL regions.

For successful word encoding after right ATLR, individuals with HS showed less plasticity between the remnant MTL seed and the remnant fusiform gyrus than those without HS.

For successful face encoding after right ATLR, people with HS had more plasticity (i.e., greater increases in connectivity) from 3-12 months to 10 years between the contralesional hippocampus seed and the left posterior fusiform gyrus, but reduced plasticity with the remnant posterior fusiform gyrus, compared to those without HS. From the remnant MTL seed, people with HS had less plasticity with the remnant posterior parahippocampus than those without HS.

### Shorter-term plasticity (3-12 months) supportive of longer-term improvement in postoperative memory

Post-hoc assessments evaluated further the neural mechanisms specific to long-term recovery of memory. We investigated the changes from before to 3-12 months after ATLR that were supportive of ongoing improvement over the longer term (3-12 months to 10 years) in PWE, compared to changes observed in healthy controls. Similar methods and thresholding to the primary longer-term plasticity analyses were employed (see *Methods*) and results are detailed with MNI coordinates and T-statistics in SI.

In left ATLR compared to controls, long-term improvement in verbal memory was associated with increased functional connectivity short-term compared to preoperatively, from the remnant MTL seed with the right middle cingulate and right superior parietal cortices, and reduced short-term connectivity with the remnant posterior fusiform gyrus. From the contralesional hippocampus seed, people with long-term verbal memory improvement exhibited reduced connectivity with the right inferior frontal cortex (gyrus and inferior OFC) and right angular gyrus, but increased connectivity with the right anterior and posterior OFC, right middle frontal gyrus, and bilateral parietal areas at 3-12 months compared to pre-surgery.

Long-term improvement in visual memory in left ATLR compared to controls correlated with reduced short-term connectivity from the remnant MTL seed with the right posterior fusiform gyrus, compared to preoperatively. From the contralesional hippocampus, 3-12-month connectivity that was reduced with the right superior temporal gyrus and increased with the right middle temporal gyrus correlated with improvement in longer term memory.

In right ATLR compared to controls, people with improvement in longer term verbal memory showed increased short-term connectivity between the contralesional hippocampus and anterior MTL (left amygdala), right middle temporal and right middle cingulate cortices, and reduced with the right posterior MTL (remnant posterior fusiform) and right angular gyrus, compared to pre-surgery. From the remnant MTL seed, people with long-term verbal memory recovery exhibited enhanced connectivity with the right anterior OFC at 3-12 months compared to before surgery.

Long-term improvement in visual memory in right ATLR correlated with increased connectivity seeding from the contralesional hippocampus with the right cuneus, and seeding from the remnant MTL with the right the calcarine and basal ganglia (external globus pallidus), and decreased remnant MTL connectivity with the left inferior OFC.

## Discussion

This longitudinal study explored the networks underlying episodic memory recovery long-term after temporal lobe epilepsy surgery. Patient-specific connectivity changes of the memory-encoding network were (1) assessed from 3–12 months to 10 years postoperatively relative to changes in healthy controls, and (2) correlated with improvements in long-term verbal and visual memory functions.

We identified key increases in functional connections to posterior MTL regions, including the right posterior parahippocampus and remnant /bilateral posterior fusiform gyri, which were central to long-term recovery of both verbal and visual memory. Long-term improvements in

**Table 5 | Correlations between long-term visual memory improvement and longitudinal changes in task-based functional connectivity from 3–12 months to 10 years after anterior temporal lobe resection**

| | Left resection group | | Right resection group | |
|---|---|---|---|---|
| | *Left MTL seed* | *Right MTL seed* | *Left MTL seed* | *Right MTL seed* |
| **Connectivity increases T2 to T3** | L post fusiform G −18 −48 −14 P = 0.003[a], T = 4.29 | Left post hippocampus −22 −36 0 P = 0.042[b], T = 2.24 | L post parahippocampus −22 −34 −14, P = 0.049[a], T = 2.82 | L post fusiform (*occ.*) −20 −86 −6 P = 0.023[a], T = 3.19 |
| | L post fusiform G −34 −62 −10 P = 0.003[a], T = 4.25 | L post fusiform G −20 −48 −16 P = 0.050[a], T = 2.77 | L post fusiform G (*temp*) −38 −26 −18 P = 0.008[a], T = 3.79 | L post fusiform (*temp-occ.*) −40 −52 −10 P = 0.032[a], T = 3.03 |
| | R post fusiform G 36 −62 −16 P = 0.001[a], T = 4.72 | R anterior OFC 28 36 −18 P = 0.001 T = 3.34 | R post fusiform G (*temp-occ.*) 36 −64 −16 P = 0.006[a], T = 3.94 | R post parahippocampus 34 −34 −16 P = 0.038[b], T = 2.44 |
| | L inf temporal G −40 −46 −14 P = 0.001, T = 3.63 | R sup frontal G (dorsomedial) 18 −10 60 P < 0.001, T = 3.76 | R post fusiform G (*temp.*) 34 −38 −16 P = 0.034[a], T = 3.03 | L inf temporal G −44 −52 −6 P < 0.001, T = 3.91 |
| | R mid temporal G 38 −56 16 P < 0.001, T = 4.14 | R mid occipital G 26 −86 2 P = 0.001, T = 3.62 | L pallidum - VL thalamus −18 −8 8 P < 0.001, T = 4.37 | |
| | R calcarine G 10 −90 0 P < 0.001, T = 4.04 | | L mid temporal G −40 −66 10 P < 0.001, T = 3.76 | |
| | | | R mid temporal G 46 −68 16 P < 0.001, T = 3.78 | |
| | | | R inf temporal G 52 −18 −18 P = 0.001, T = 3.57 | |
| | | | R inf frontal G (anterolateral) 40 14 14 P = 0.001, T = 3.52 | |
| | | | R supramarginal G 54 −44 26 P < 0.001, T = 4.00 | |
| **Connectivity decreases T2 to T3** | *No suprathreshold connectivity* | *No suprathreshold connectivity* | *No suprathreshold connectivity* | *No suprathreshold connectivity* |

Results of three-way ANCOVAs for each MTL seed during encoding of faces subsequently remembered. These represent longitudinal connectivity changes in people with epilepsy that were significantly correlated with improvement in visual memory z-scores (i.e., from design learning test) from short to long-term post-surgery. These analyses were performed relative to connectivity changes and memory of healthy controls. MTL-to-neocortex connectivity is displayed at P < 0.001, uncorrected.

*FWE* = family-wise error, *G* gyrus, *Inf* inferior, *Mid* middle, *MTL* medial temporal lobe, *OFC* orbitofrontal cortex, *Post* posterior, *Sup* superior, *T2* 3–12-month follow-up (median 11 months), *T3* 10-year follow-up (median 9 years).

[a]*P* < 0.05, FWE correction, using a 6 mm sphere for contralesional MTL regions.
[b]*P* < 0.05, FWE correction 3 mm sphere in remnant MTL regions.

verbal and visual memory after left and right ATLR, respectively, correlated with longitudinal connectivity increases with the contralesional anterior cingulate cortex and external globus pallidus.

**Long-term postoperative memory changes**
Over almost a decade, healthy controls exhibited an initial improvement in verbal memory from the baseline to 3–12-month assessments followed by a significant decline at 10 years. This pattern suggests an early practice effect that resolved over time, with similar mean performances at baseline and 10 years indicating overall stable cognitive function.

Neuropsychology studies demonstrate that ATLR is characterized by an immediate impact on episodic memory functions[1,22], which in seizure-free people is followed by either cognitive stabilization or improvement as recovery time increases (beyond two years)[1,2,23]. In line with current literature, postsurgical PWE did not present with the learning/test-retest effect in verbal memory, suggesting an early surgical insult on memory function. Yet beyond 3–12 months, our predominantly seizure-free cohort demonstrated cognitive stability at the group-level and patient-specific recovery was evident.

From 3–12-months to 10 years, 50% of left ATLR and 38% of right ATLR showed clinically significant improvement in verbal and visual memory, respectively. Visual memory improved after left-sided

surgery, with group-level performance significantly increasing above baseline within the first year post-surgery and stabilizing over the longer term. Ten years postoperatively, 75% of left ATLR and 58% of right ATLR had good verbal and visual memory outcomes compared to preoperatively, with improved performance or return to baseline. Enhancement of baseline performance possibly translates to the removal of nociferous cortex, and corresponding release of functions and reserve capacities previously suppressed by epileptic activity[24,25].

This has important clinical implications for presurgical discussions and cognitive rehabilitation endorsement[26]. In intractable TLE, recurrent seizures and a heavy ASM load negatively impact mood, cognition, and integrity of memory-related brain regions[27,28]. In this study, most postsurgical PWE achieved seizure freedom, approximately half reduced or ceased ASMs during follow-ups (50% left, 54% right ATLR), and had cognitive stabilization or improvement long-term post-operation.

**Efficient medial temporal plasticity over a decade: change of trajectory**
Early after surgery, activation-based fMRI studies have shown plasticity effects in the contralesional hippocampus, amygdala and anterior parahippocampal gyrus, supportive of memory function beyond 5 years post-ATLR[9–11]. In our study, impaired preoperative connectivity

## Left resection group

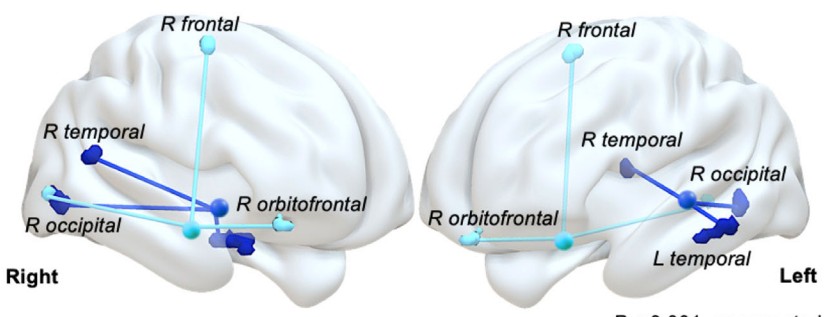

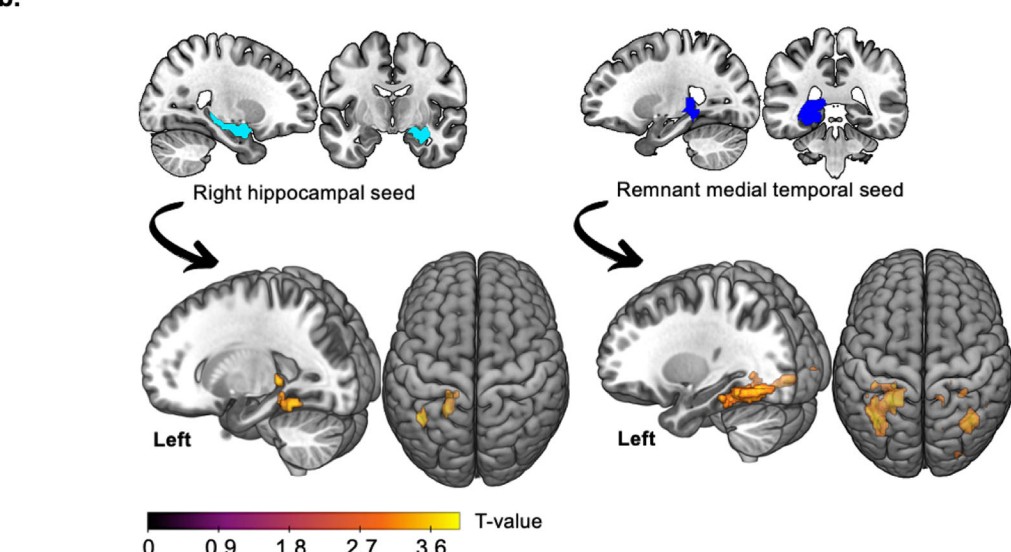

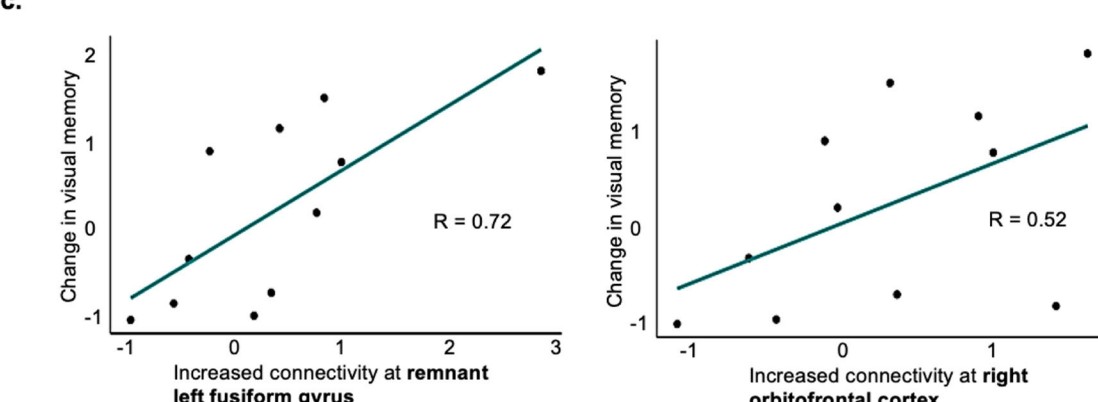

**Fig. 4 | Longitudinal increases in functional connectivity correlated with visual memory improvement from 3–12 months to 10 years after left-sided surgery.** **a** 3-D brain renderings show long-term adaptive connectivity increases after left ATLR between MTL seeds and extra-MTL regions (fMRI clusters), represented by cyan (contralesional hippocampus) and dark blue (remnant MTL) connecting lines. **b** Brain slices depict adaptive connectivity increases at medial temporal-level from both MTLs. **c** Scatter plots display eigenvariates from one-sided T-contrasts of connectivity increases, contralesionally with the orbitofrontal cortex ($P = 0.001$) and ipsilesionally with the remnant posterior fusiform gyrus ($P = 0.003$), plotted against long-term visual memory (design learning) changes. All connectivity is presented controlling for medication and IQ, significant at $P < 0.001$ (uncorrected) in neocortical regions and $P < 0.05$ family-wise error small volume corrected in the MTL. *ATLR* anterior temporal lobe resection, *MTL* medial temporal lobe.

## Right resection group

**a.**

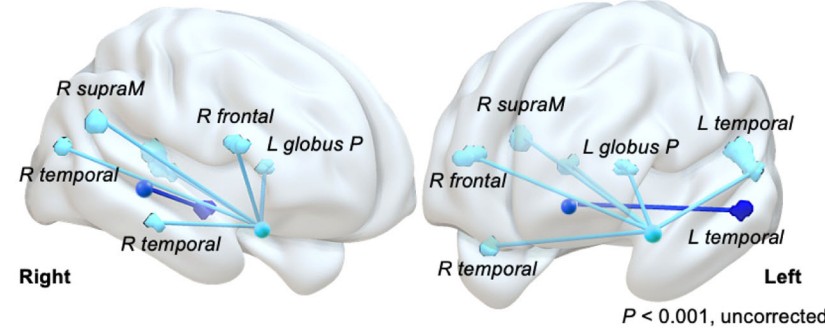

*P < 0.001, uncorrected*

**b.**

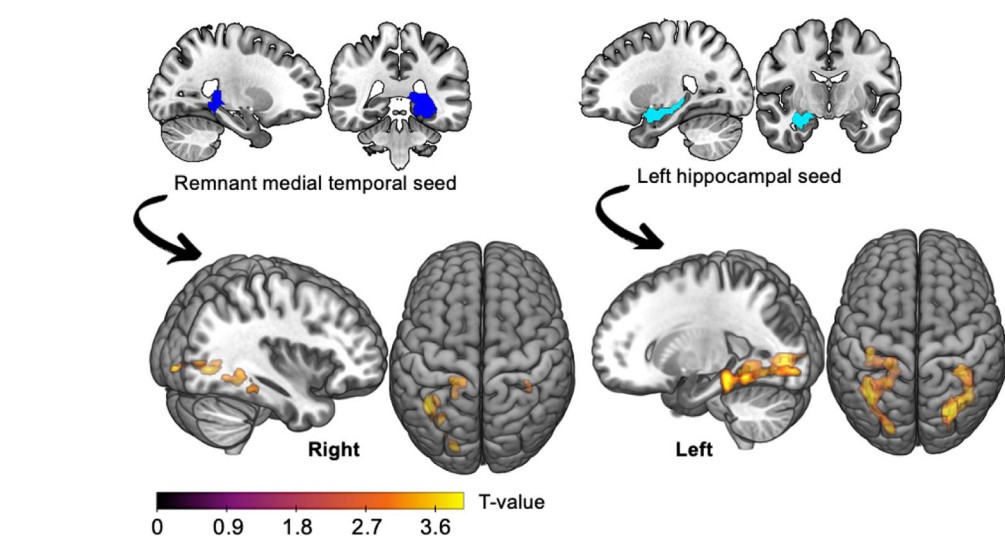

**c.**

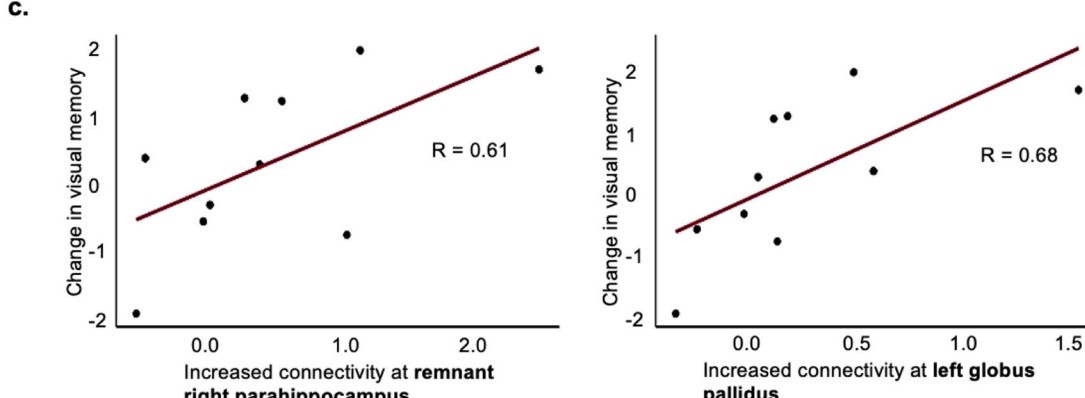

**Fig. 5 | Increased functional connectivity correlated with visual memory improvement over a decade after right-sided surgery. a** 3-D brain renderings show adaptive connectivity increases from 3–12 months to 10 years post-right ATLR between extra-MTL regions and each MTL seed, represented by cyan (contralesional hippocampus) and dark blue (remnant MTL) connecting lines. **b** Brain slices highlight medial temporal-level connectivity increases seeding from both MTLs. **c** Scatter plots display eigenvariates from one-sided T-contrasts of connectivity increases, contralesionally with the globus pallidus (*P* < 0.001) and ipsilesionally with the remnant posterior parahippocampal gyrus (*P* = 0.038), plotted against long-term visual memory (design learning) changes. All connectivity is reported controlling for medication and IQ, significant at *P* < 0.001 (uncorrected) in neocortical regions and *P* < 0.05 family-wise error small volume corrected in the MTL. ATLR anterior temporal lobe resection, Globus P globus pallidus, MTL medial temporal lobe, SupraM supramarginal gyrus.

was followed by new functional connections between the contralesional hippocampus and amygdala 3–12 months after right ATLR, which efficiently supported long-term improvement in verbal memory.

Conversely, increased memory fMRI activity in the surgically spared posterior hippocampus and associated regions is reportedly not functionally efficient four months post-ATLR[20]. This aligns with the observed adaptive decrease in remnant fusiform connectivity at 3–12 months compared to preoperatively, irrespective of surgical laterality and memory type. The surgically induced, partial severance of the inferior longitudinal fasciculus (ILF) may result in this observed functional disconnection, as the ILF connects anterior and posterior MTL regions and its resection can affect verbal and visual cognition[29,30].

It is of note that PWE with continued memory improvement beyond 3–12 months developed new functional connections with posterior and surgically spared regions (i.e., fusiform and parahippocampal gyri), that were efficiently integrated in the cognitive network at 10 years. Conversely, people with HS exhibited reduced connectivity to the remnant fusiform gyrus, a region herein demonstrated to be critical for memory recovery. Preoperatively, people with HS had greater functional network disruption than those without[16,31]. Memory function at 10 years was significantly worse in people with HS, suggesting that this represented a maladaptive reduction in long-term connectivity.

Overall, we propose that early post-surgery, there is efficient new contralateral connectivity with anterior medial temporal regions, laying the foundation for longer-term plasticity, as posterior remnant regions become involved in recovery of mnemonic processes.

### Resection extent and successful network reorganizations

In line with our long-term volumetric findings, Bonelli and colleagues[20,32] found that short-term post-ATLR, memory function does not correlate with the volumes of preoperative and remnant postoperative hippocampus. This lack of correlation may be attributed to the observed consistency in resection extent irrespective of surgical laterality, resulting in a clinical cohort with homogeneous remnant MTL volumes.

Moving beyond volumetric analyses, connectivity measures highlighted crucial differences in plasticity patterns. These included successful network reorganization toward structures that putatively underly the processing and specialized pattern recognition of graphic and contextual stimuli[33,34]. Memory formation may rely on the conceptual representation of features supported by specialized visual medial temporal regions. Recent evidence denotes positive engagement of the fusiform's visual word form and face areas during recognition of both letters and faces, with functional lateralization independent of material type[35]. This raises questions about external drivers of hemispheric dominance, beyond category-selective representations[36] – for instance, the surgical removal of homologous medial temporal regions.

Our findings emphasize the value of using connectivity metrics to assess individual mechanistic variations in cognitive recovery and the importance of preserving posterior MTL areas during surgery. Enhancing their integration with the broader brain network through preoperative pre-habilitation and postoperative rehabilitation presents a promising clinical avenue to facilitate long-term cognitive recovery.

### Long-term support from structures of goal-directed behaviors

Long-term visual memory recovery following right ATLR was associated with new functional connections with the external globus pallidus bilaterally; ipsilaterally in the short term post-surgery and also contralesionally by 10 years.

Although known modulators of the motor control system, the globus pallidus interna and externa are also part of frontal-striatal loops subserving spatial memory encoding and controlling working memory capacity to store relevant information[37,38]. In concert with the preparatory engagement of bilateral frontal regions, left external pallidal activity allows subsequent information filtering during encoding of visuo-spatial working memory and its task-related recruitment reduces unnecessary storage of distractors[37]. Correspondingly, people with right-sided basal ganglia injury exhibit deficits in encoding and recall of visual working memory under equally weighted attentional demands[39]. This implies corpus striatal engagement beyond memory processes that depend on selective attention mechanisms.

For verbal memory recovery, the importance of right frontal engagement was enhanced 3–12 months after both ATLR compared to preoperatively, including the right middle cingulate and anterior orbitofrontal cortices. The absence of long-term decreases in frontal connectivity suggests this plasticity was sustained and adaptive 10 years post-surgery. There were ongoing connectivity increases between remnant MTL and contralesional anterior cingulum long-term after left ATLR, possibly reflecting a greater impact of left than right-sided resection on verbal mnemonic processes.

Reorganization of verbal memory function to the cingulate cortex is critical in TLE. PWE exhibit memory fMRI activations in the anterior cingulate cortex that support successful word encoding both before and short-term after surgery[7,40] and correlate with improvement 12 months after left ATLR[9]. The anterior cingulate cortex shares direct structural connections with the anterior parahippocampal gyrus[41] and is an essential node of the default mode and salience networks[42]. In healthy people, heightened functional connectivity between right cingulum and temporal regions is associated with increased cognitive load and sustained attention[43]. Increased functional connectivity and structural integrity of the contralesional anterior cingulum may subserve better global cognition[44] and flexible learning[45], shifting attention allocation for higher processing speed, as evidenced in post-stroke cohorts.

Recent hippocampal pathology reports have shed light on the hippocampus' role in binding elements regardless of duration of retention; both for short-term working and long-term memories. They denote similar activity of hippocampal neurons underpinning long-term memory during short-term processes requiring great memory load and cognitive control[46,47]. This activity is synchronized with frontal engagement based on cognitive demand[47]. It is therefore plausible that the surgical strain on function is compensated by new functional couplings with extra-temporal areas subserving goal-directed behaviors, such as the globus pallidus and cingulate cortex, and facilitating successful formation of long-term memory.

### Task-specific patterns of extra-MTL plasticity

The long-term plasticity of extra-temporal memory networks varied with memory tasks. Visual memory recovery underpinned a bilateral network involvement with compensatory mechanisms engaging both temporal and frontal regions a decade postoperatively, while verbal memory displayed a more lateralized pattern of compensation, primarily involving the reinstatement of posterior MTL plasticity disconnected during surgery.

Preoperatively, successful visual memory encoding correlates with activation fMRI maxima in both anterior hippocampi, and with the bilateral middle and inferior temporal cortices and right frontal areas such as the right orbitofrontal cortex in healthy subjects[7,16]. This extensive network's architecture may explain the observed, long-term increases in connectivity with neocortical areas typically part of the episodic memory network. Such neuroplasticity adaptively compensated for functional loss in the ipsilateral MTL following right ATLR, and for the disruption of the bilateral temporal cognitive network after left ATLR.

Verbal memory encoding is predominantly lateralized to the left hemisphere in healthy subjects[7,29,48]. As such, individuals undergoing

right-sided resection may maintain nearly intact, established neocortical networks of verbal memory arising from the left MTL. This may account for the observed long-term plasticity after right ATLR, which, beyond 3–12 months, primarily involved increased connectivity within the remnant right MTL.

These differential plasticity patterns underscore the dynamic nature of cognitive network architectures, adapting according to the task's functional demands and learning strategies involved, as previously proposed[49].

### Strengths and limitations

We present a unique longitudinal design spanning many years, with data acquired from the same participants at four time-points compared to a consistent control population. Despite a small sample size, we showed statistically significant network changes and correlations.

Our study has several strengths and limitations. A methodological strength is the use of gPPI as a connectivity tool. This models the entire experimental span, allowing analysis of neural correlates highly specific to subsequent memory effects, and better controls for both type I and II errors compared to other tools such as standard PPI[50]. Next, in our analyses we took into account number of ASMs at each timepoint and included these as regressors of no interest. This modeled for any changes attributed to the change in number of ASMs at each timepoint. Due to the variability of ASMs prescribed clinically however, we were unable to assess the specific impact of individual ASMs on memory network changes.

The postsurgical cohort was primarily seizure-free, limiting the generalizability of our results to people with a good prognosis. The small number of participants (3/25) with ongoing seizures may also restrict the applicability of our finding that ongoing seizures did not affect memory recovery. Future studies with larger numbers should investigate the modulatory effect of specific ASMs and differential seizure outcomes on network recovery. In particular, verbal memory changes may be more sensitive to ASMs known to disrupt language-related function[51–54] (see SI), compared to visual memory recovery. The RCI provided a standardized approach to assess cognitive change but is limited by its reliance on published norms and subjective confidence intervals, which may affect its generalizability and sensitivity.

Over a 10-year period, scanner changes were inevitable. To address this, we used a flexible factorial design that accounted for potential heteroskedasticity and consulted with the UCL Wellcome Centre for Human Neuroimaging; longitudinal differences in connectivity reflected interaction effects between time-point and memory task, which are orthogonal to the main time-point effect and therefore distinguishable from effects related to scanner change. Healthy controls showed a significant test-retest influence on verbal memory, at the initial short-interval follow-up, which was not seen in PWE likely due to the immediate surgical impact on function. The shorter-term timepoint consisted of a combination of 12-month and 3-month data for subjects who did not return at 12 months. We performed sensitivity analyses to verify that there was no significant difference in neuroplasticity effects between the mixed 3–12-month and 12-month-only cohorts. Future studies should assess the plasticity of memory networks at more frequent, regular intervals to better understand the temporal evolution of the described plasticity changes.

In conclusion, our longitudinal functional connectivity study investigated the neural mechanisms underlying memory recovery in a unique dataset, extending over a decade after epilepsy surgery. ATLR enabled ASM reduction, sustained seizure freedom, and cognitive stabilization or improvement 10 years post-operation. Memory network plasticity was an ongoing phenomenon that continued to reshape over time. Increases in functional connectivity that supported long-term memory recovery involved surgically spared medial temporal and extra-temporal structures, including the contralesional cingulum and pallidum. While preliminary, these findings can impact surgical intervention to avoid crucial regions of memory reorganization, shed light on the biology of long-term memory rehabilitation, and advocate ongoing cognitive rehabilitation to optimize cognitive outcomes years after surgery.

## Methods

### Subjects

Twenty-five individuals with medically refractory TLE underwent ATLR from 2009 to 2012 at the National Hospital for Neurology and Neurosurgery (NHNN), London, United Kingdom[9]. There were 12 left-sided ATLR (seven males, median preoperative age 38 years, interquartile range (IQR) 28-41) and 13 right-sided (four males, median preoperative age 38 years, IQR 29-50).

Neuropsychology assessment, as well as structural and memory functional MRI (fMRI) were acquired at four timepoints: preoperatively, and at a median 3-month (IQR = 3–4) and 12-month (IQR = 11–13.5) post-surgery, and up to 10 years postoperatively (median = 9, IQR = 8–10). Neuropsychological and memory fMRI evaluations were conducted on the same day, ensuring comprehensive assessments at each follow-up. Eight left-sided and 10 right-sided ATLR cases completed assessments at all postoperative follow-ups. All subjects included in this study returned for the 10-year neuropsychometry and fMRI assessments (Table S1 for details).

To optimize acquired data and analysis, a short-term assessment timepoint was introduced. It encompassed all data collected at 12 months, alongside data at 3 months from patients unable to attend 12-month follow-up (four left-sided and three right-sided). Throughout this manuscript, short-term data refers to assessments conducted during the 3–12-month follow-up period (median = 11 months, IQR = 8–12), occurring from 2009 to 2013. Conversely, long-term data pertains to the 10-year follow-up, conducted between 2019–2022.

Ten healthy, English-proficient, matched-controls (four males, aged 27-50) were assessed at similar intervals. Controls and left/right ATLR groups were comparable for language dominance, handedness, and age. Sex was considered in study design, with sex-matched controls. Sex was determined by self-report or health records. No further sex-based analysis was performed, as all three groups were comparable (Fisher exact test) and the study focus was on functional network changes related to memory recovery.

All participants provided written informed consent in accordance with the Declaration of Helsinki. The NHNN and Institute of Neurology Joint Research Ethics Committee approved this research (18/LO/1447). Participation was voluntary, and participants received no compensation for taking part in the study. Exclusion criteria included contraindication to MRI, non-proficient English speaker, and intelligence quotient (IQ) < 70. Postoperative seizure outcome was assessed using the International League Against Epilepsy (ILAE) classification[21]. Seizure frequency was collected from seizure diaries at preoperative and postoperative time-points; comprising the total number of focal impaired awareness seizures per month and focal to bilateral tonic–clonic seizures.

### Neuropsychological tests

Patients and controls underwent standardized neuropsychometry at corresponding time points; before, at median 3-month and 12-month after surgery, and up to 10 years postoperatively (median 9 years) (see SI for details).

Intellectual functioning was evaluated using Full-Scale IQ (FSIQ) of the Wechsler Adult Intelligence Scale (WAIS)[55] in controls, and with the National Adult Reading Test (NART-2)[56] for metrics of premorbid IQ in patients[57], and reliable estimates of patients' FSIQ[58]. Patients' verbal and visual memory was assessed using verbal and design learning subtests of the BIRT Memory and Information Processing Battery version I (BMIPB-I)[59], as previously done[9,16,57,60], These memory measures are sensitive to temporal structures' integrity[61]. In controls,

BMIPB-I was employed up to January 2021, and BMIPB–II from February 2021 onwards.

Memory scores were standardized into z-scores, using age-specific norms of corresponding BMIPB version, accounting for version change and age-related differences[62]. Memory change represented the difference between z-scores of short-term and long-term follow-ups. Improvement or decline were considered clinically significant based on reliable change index (RCI) upper and lower limits, using 95% confidence intervals, as described in neuropsychological and imaging studies[18,57,61,62]. Using the mean and standard deviation of the whole sample, the RCI probes meaningful change by adjusting for test reliability and practice effect in a test-retest context[62].

## Magnetic resonance data acquisition

Preoperatively and short-term postoperatively, participants were scanned on a 3T GE Signa Excite HDx MRI scanner, with a 20-channel head coil. Long-term postoperatively, data was acquired using 3T GE Discovery MR750, with a 32-channel head coil. Refer to SI for detailed acquisition parameters.

Preoperatively and at short-term (long-term) follow-up, memory fMRI timeseries included T2*-weighted gradient echo planar images (EPI) acquired using 36 (50) contiguous oblique axial slices per volume, 24-cm field of view, 2.5 (2.4) mm slice thickness with 0.3 (0.1) mm gap, with TE of 25 (22) ms and TR of 2750 ms[7].

At each scanning time-point, the field of view covered the temporal and frontal lobes, and slices were aligned on the sagittal view with the long axis of the hippocampus[9].

## Functional memory paradigm

We applied the same material-specific memory fMRI paradigm as preoperatively[7,16], detailed in SI.

In summary, black-and-white faces and words were visually presented on a magnetic resonance-compatible screen, viewed through a mirror during a single scanning session at each time-point. Forty minutes after scanning, participants were tested on the same 100 words and faces intermixed with an additional 50 novel words/faces as foils, separately. Participants categorized items as remembered, familiar (if unsure), or novel via a button-box, and performance was recorded as either successfully remembered, familiar or forgotten. At each scanning timepoint, an identical memory fMRI paradigm was performed with different items.

## Data analysis

**Preprocessing.** For both 3-12-month and 10-year follow-ups and every subject, the anatomical 3D-T1 scan underwent field bias correction with Advanced Normalization Tools[63] and was registered to a scanner-specific template in MNI space. The scanner-specific template was created from 30 healthy subjects, 15 individuals with left hippocampal sclerosis and 15 people with right hippocampal sclerosis, using high-resolution whole-brain EPI[7].

Short- and long-term functional imaging time-series were realigned to the mean image and time-corrected using Statistical Parametric Mapping 12 (SPM12; Wellcome Department of Cognitive Neurology, Institute of Neurology, London, UK; http://www.fil.ion.ucl.ac.uk/spm/). Normalization into standard anatomical space was done using EasyReg[64], a deep-leaning registration method accessible via Freesurfer[65]. Transformation parameters from the T1-to-template registration were used to warp fMRI time-series to template using nearest-neighbor interpolation. Normalized time-series were smoothed on SPM12 using 8 mm full-width at half-maximum Gaussian kernel[60].

## Longitudinal connectivity analysis

**Event-related contrasts.** Event-related spmT maps of subsequent memory effects were generated for each subject and separately for words or faces on SPM12 via random-effects analysis of a blocked

design general linear model (GLM)[9]. Six regressors of interest were created; words and faces subsequently remembered, familiar, or forgotten. Six motion parameters were added as confounds. Resulting event-related statistical maps were used for subsequent single-level connectivity analyses.

**MTL seeds.** To ensure unbiased ROI selection for group comparisons, seed selection was based on AAL-based anatomical masks (WFU-Pick-Atlas toolbox v3.0). Healthy controls had the left and right hippocampi as MTL seeds, based on their role in successful memory[7,20]. For patients, one MTL seed included the non-resected contralesional hippocampus. Ipsilesionally, the remnant MTL seed entailed remnant hippocampus and parahippocampal gyrus, based on left and right ATLR group resection masks[57,66]. All participants underwent standard ATLR with little within-group difference in the size of resection cavity.

**Generalized PPI analysis.** A generalized form of PPI (gPPI) spanned the entire experimental space[50], modeling beta-estimates of all subsequent memory conditions (further details in SI).

At the participant-level, seed-to-whole-brain connectivity analysis was conducted on MATLAB[50]. For verbal and visual memory separately, the subject-level gPPI model included three regressors: time-course of each event-related task-condition, timeseries of one MTL seed, and of the PPI term (task*seed interaction). All six task-conditions were modeled to better probe successful verbal and visual memory effect[50]. The average seed time-course was extracted within each anatomical MTL mask/seed. Physiological and psychological variables were treated as nuisance regressors. T-contrasts were generated for each MTL seed and words/faces subsequently remembered, revealing whole-brain cortical areas that were significantly more correlated with the seed during successful memory encoding than during uncertain/failed conditions, based on the PPI term prediction[50].

For each participant, separate GLMs were performed for each MTL seed. The same task and seed regressors were used for all participants and at 3–12-month and 10-year follow-ups. Resulting single-level gPPI t-contrasts of successful subsequent memory were used for group-level random-effects analyses (see below section).

## Statistical analyses

**Clinical and neuropsychological data.** Data was analyzed using R 4.0.5. Demographics were evaluated using Fisher's exact test for sex proportion and one-way ANOVA for parametric z-scores (preoperative memory and IQ) with post-hoc tests correction using Tukey's HSD adjustment. Kruskal–Wallis tests were performed for nonparametric continuous variables (age, ILAE outcomes, ASM intake and change)[60].

Linear mixed-effect models were conducted to assess Group*Time interaction effects on memory z-scores, corrected for multiple comparisons using False Discovery Rate. For each list and design-learning task, linear models included the fixed effects of Group (left ATLR, right ATLR, controls), Time (3-12–month and 10-year memory z-scores), and Group*Time interaction, along with subject-specific random intercepts to account for both individual variability in baseline memory and non-independence of repeated measures[67].

Due to small numbers, Fisher-Pitman permutation tests were performed to assess the impact of HS pathology preoperatively, and ASM cessation and seizure-freedom at 10 years on long-term memory function.

**Longitudinal assessment of the functional memory network.** All data was analyzed with SPM12. Shorter-term memory data consisted of a combination of both 12-month and 3-month timepoints for subjects who did not return at 12 months. Sensitivity analyses (see SI) showed no significant difference between longitudinal connectivity changes in a mixed 3-12month cohort compared to those in a 12-month-only

cohort. This justified combining 3- and 12-month postoperative scans into the short-term timepoint.

Postoperative clinical factors like change in medication or ASM cessation can impact memory function and the corresponding mechanisms underlying cognitive network recovery. As such, individual ASM intake at both 3-12 months and 10 years was treated as continuous regressor of no interest in all neuroimaging analyses.

**Long-term changes in functional connectivity from 3–12-month to 10-year follow-ups.** Two-way mixed ANOVAs within a flexible factorial design, with both IQ and change in ASM intake as confound regressors, were performed to investigate changes in MTL-seeded memory connectivity between short-term and long-term follow-ups in the individual patient groups compared to changes in test-retest over that timeline in healthy subjects. A distinct analysis was conducted for faces and words successfully remembered, seeding from each MTL separately.

For each subject, the relevant first-level gPPI *t*-contrasts for each of the scanning timepoint (3–12 months and 10 years) were entered. Each flexible factorial design involved a random subject factor, a three-levels group factor (controls, left and right ATLR), and a two-levels condition factor (short-term and long-term postoperative scans), allowing the investigation of a Group x Condition interaction for the successful memory contrasts. Differences in activations across scanning sessions were compared between ATLR groups and controls in *t*-contrasts: 10-year connectivity > 3–12-month connectivity in left or right ATLR versus controls, and 10-year connectivity <3–12-month connectivity in left or right ATLR versus controls.

The effect of scanner change between short-term and long-term timepoints was addressed using specific parameters of the flexible factorial design. This design modeled any possible heteroskedasticity, allowing for differences in residual variance between scanners. Longitudinal differences in task-based functional coupling between brain regions represented interaction effects (time-point x task condition) that are orthogonal to the main effect of time-point, making them distinguishable. In summary, flexible factorial *t*-contrasts modeled the within-subject differences in MTL-seeded, whole-brain functional connectivity across postoperative follow-ups in each ATLR group, beyond the connectivity changes seen in controls. This model allowed control of both age-related and scanner-change effects.

**Correlation between functional reorganization in TLE and memory recovery.** Three-way ANCOVAs, controlling for IQ and changes in ASM intake, were conducted for each MTL seed and successful memory contrast (words/faces) to investigate differences in functional connectivity from 3–12-month and 10-year follow-ups related with improvement in memory functions over this timeline.

Specifically, individual changes in memory z-scores were correlated with changes in task-modulated fMRI connectivity over time, modeling the fMRI biomarkers more strongly associated with those whose memory successfully improved compared to those who did not. Differences in z-scores (i.e., BMIPB I or II verbal and visual learning scores converted into age-normalized z-scores) from 3 to 12 months to 10 years were used as continuous variables.

**Statistical thresholds.** Given our a priori hypothesis of increased local MTL connectivity (including the fusiform gyrus)[9,13,20], MTL connectivity was corrected for multiple comparisons at $P < 0.05$ voxel-wise, controlling for family-wise error rate via small volume correction[9,68,69]. This included a 6 mm-radius sphere in contralesional MTL regions[7,16], and 3 mm-radius sphere in remnant hippocampus and parahippocampal gyrus to avoid resection cavity-related activation. All reported seed-to-remnant hippocampus/parahippocampus connectivity was validated against artifacts using exclusive MTL group-resection masks.

Group comparison and correlation analyses generated highly specific MTL-to-whole-brain, longitudinal, event-related *t*-contrasts[70]. Thus, at the extra-MTL level, functional connectivity is reported at an exploratory $P < 0.001$ threshold (uncorrected), alike previous longitudinal, event-related and network fMRI studies[9,16,68] and correlation results are reported at exploratory $P < 0.001$, masked within binary group masks of the main PPI effect, in line with PPI correlations studies[71–73].

### Resection extent

To test for potential effects of variations in the extent of temporal lobe resection, postoperative T1 scans were co-registered to the pre-operative T1 native space using Elastix[74] with affine and bspline transformations. Individual resection masks were automatically generated using *Resseg*[75]; a self-supervised, convolutional neural network approach to segment the temporal lobe cavity. The extent of anterior temporal lobe structures' resection was assessed, accounting for individual variations in brain sizes, by dividing the resection cavity volume by the total brain's volume for each subject. The proportion of hippocampus volume resected was calculated using the formula: $(V_1 - V_2) / V_1$, with $V_1$ and $V_2$ standing for the hippocampus volume at the preoperative baseline and at the 10-year follow-up, respectively.

Mann–Whitney tests were conducted to assess whether the size of resection of both the anterior temporal lobe structures and the epileptogenic hippocampus differed according to surgical laterality. To investigate the effect of differences in the volume of surgically spared hippocampus on long-term recovery of verbal and visual memory, linear regressions were performed within each resection group. The ipsilateral remnant hippocampus volume at 10 years was the explanatory variable and the individual change in list- and design-learning z-score from 3-12 months to 10 years the outcome variable.

### Reporting summary

Further information on research design is available in the Nature Portfolio Reporting Summary linked to this article.

### Data availability

Source data are provided with this paper and can be accessed at https://github.com/marine-fleury/memory-plasticity.git[76]. The processed neuroimaging data are available at https://osf.io/8suqt[77]. Source data are provided with this paper.

### Code availability

*R* code is provided and can be accessed at https://github.com/marine-fleury/memory-plasticity.git[76].

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

## Acknowledgements

The authors thank Andrea Hill, Epilepsy Society radiographer, for her crucial help in data collection, and Jane de Tisi for gathering long-term clinical information. We are grateful to Dr. Gerard Hall and Dr. Bernardo Pimentel for methods and clinical discussions. Special thanks to the UCL Wellcome Center for Human Neuroimaging, particularly Dr. Peter Zeidman and Prof. Karl Friston, and to Prof. Carolin Moessnang (SRH University Heidelberg) for their advice on experimental design and statistical analysis. We extend our special gratitude to all participants for their invaluable cooperation. This work is supported by National Institute for Health Research UCLH Biomedical Research Centre (grant 229811, M.K.S.), The Wellcome Trust (grant 083148, J.S.D.), Wellcome Trust Innovation Program (106882/Z/15/Z, 218380/Z/19/Z, J.S.D.) and MRC (grant G0802012, MR/M00841X/1, G.P.W.; grant MR/X031039/1, M.K.S.), and UKRI Future Leaders Fellowship (MR/T04294X/1, P.T.). M.F. and M.K.S. are supported by the UCLH BRC and the Epilepsy Society.

## Author contributions

Neuroimage data collection: M.N.F., M.K.S., L.P.B., G.P.W., SBuck. Neuropsychology testing: P.J.T., SBaxendale, and M.N.F. Conception and study design: M.K.S. and M.N.F. Data processing, statistical analysis, and results interpretation: M.N.F. Physics support: L.P.B. and P.T. Neuroimage processing support: L.P.B. Data analysis support: F.X. and L.P.B. Neurosurgeries: AM. Data interpretation support: D.G., M.J.K., J.S.D., and M.K.S. Manuscript and supporting materials write-up: M.N.F. Manuscript preparation support: L.P.B., P.T., F.X., D.G., G.P.W., M.J.K., M.K.S. Project supervision: M.K.S. and J.S.D. All authors approved the final manuscript.

## Competing interests

The authors declare no competing interests.
