## [Peer Review file · Nature Communications]

Long-Term Memory Plasticity in A Decade-Long Connectivity Study Post Anterior Temporal Lobe Resection

Corresponding Author: Ms Marine Fleury

Version 0:

Reviewer comments:

Reviewer #1

(Remarks to the Author)

This longitudinal study aimed to identify neural mechanisms underlying verbal and visual memory network plasticity and recovery long-term after epilepsy surgery. The authors assessed changes in task-associated functional connectivity from 3–12 months to 10 years after anterior temporal lobe resection (ATLR) using generalized psychophysiological interaction (PPI) proposed by Friston et al, and correlated fMRI memory network alterations with improvement in episodic memory function. They focused on local mesial temporal lobe (MTL) connectivity, including connections to the fusiform gyrus based on the authors' previous work.

They found that from before to 3–12 months post-ATLR, healthy controls exhibited a verbal memory learning/test-retest effect absent in postsurgical patients, suggesting an immediate surgical insult on cognitive function. Beyond 3–12 months, their predominantly seizure-free cohort demonstrated cognitive stability at the group level, consistent with neuropsychological research. In addition, they showed that from 3–12 months to 10 years postoperatively, long-term reorganization toward posterior fusiform and right posterior parahippocampal gyri was significantly supportive of postoperative improvement in visual and verbal memory, supporting the role of posterior MTL regions in verbal memory formation. They concluded that specific changes in memory networks and identified regions crucial for long-term verbal and visual memory recovery.

I have the following questions/comments:

How many patients had the 10-year follow-up memory and fMRI evaluation? This should be clear in the main text. The supplemental materials could include details on the number of subjects in all time points.

Please describe better how the reliable change index was defined. Was this based on the controls' scores?

Reliable Change Index calculations created with data from healthy control groups may exaggerate the decline when used with epilepsy patients. This should be commented on in the discussion.

Preoperatively and short-term postoperatively images were acquired in a scanner different from the long-term follow-up, as expected due to the end-of-life and upgrades in these machines. In addition, the head coils were different (20-channel head coil in the first scanners versus 32-channel head coil in the follow-up scanners). fMRI data are quite sensitive, and the signal varies according to different scanners and head coils with more channels. How were these differences handled?

The last paragraph of the supplementary text is truncated.

“Single-level gPPI t-contrasts of successful subsequent memory were applied to group-level random-effects analyses for functional connectivity estimation and comparison across 143 groups (see below section).”

There is no section below that.

Please add the pertinent information about the group comparisons.

Minor typo on Page 7 of the main text under Healthy controls: “...improved, 50% no change, while 37.5% declined. At the group level, there was a clinically 119 significant improvement in verbal memory from preoperative to 3–12 months timepoints ...”

- It should be ‘baseline’ instead of ‘preoperative.’

Reviewer #2

(Remarks to the Author)

The paper is of some interest give the data regarding neural mechanisms of memory recovery and network plasticity over nearly a decade post-surgery. Fleury et al. assessed memory network changes in 25 patients (12 left-sided resections VS 13 left-sided resections) relative to 10 healthy matched controls, using longitudinal, task-based functional MRI and neuropsychology assessments. In the seizure-free cohort, group-level memory either stabilized or improved in the long-term post-resection. Increases in functional connectivity that supported long-term memory recovery projected to medial temporal and extra-temporal structures. Reduced involvement of the thalamus may serve as an imaging biomarker of cognitive network convalescence. Although I agree with the authors that the findings of the study have the potential to impact surgical intervention to avoid crucial regions of memory reorganization, there major limitations that would need to be addressed to increase the reliability and stability of the findings.

(1) The sample size of the study was too small to make a robust conclusion ; it could not provide a group with a poor prognosis.

(2) Despite the efforts made by the authors to mitigate scanner-related effect, I do not think this is sufficient to address this issue, especially when the sample size is small.

(3) Using functional magnetic resonance imaging (fMRI) data from two scanners to explore changes in brain function over a span of approximately 10 years is significantly influenced by numerous confounding factors. Therefore, it's necessary to increase the sample size and collect fMRI data at multiple time points post surgically, as well as another dataset to further validate the stability of the results.

(4) For those patients who did not attend for a 12-month follow-up, the authors used the 3-month data instead, which may influence the results, since the existing literature suggest that patients may have distinct structural and functional reorganizations at the time of 3-month post surgery and 12-month. It is better to scan the patient the same time-interval.

(5) The aging effect of memory decline was not considered in the analysis especially at the ten-year follow-up.

Reviewer #3

(Remarks to the Author)

Despite the importance of early memory network changes following anterior temporal lobe resection for temporal lobe epilepsy, long-term plasticity and its impact on memory function are unclear. Hence, the authors investigated neural mechanisms of memory recovery and network plasticity over nearly a decade post-surgery.

From 3–12 months to 10 years postoperatively, they assessed memory network changes in 25 patients (12 left-sided resections) relative to 10 healthy matched controls, using longitudinal, task-based functional MRI and standard neuropsychology assessments.

They observed specific changes in memory networks and identified regions crucial for long term verbal and visual memory recovery. Since their postoperative cohort is predominantly seizure-free, they highlight the potential for sustained cognitive improvement and reduced medication needs over a decade after epilepsy surgery.

These findings endorse strategic approaches in epilepsy treatment: advocating for conservative surgeries and promoting the long-term use of cognitive rehabilitation for ongoing recovery.

This study is very difficult to perform since they have to follow their patient until 10 years postoperatively. I fully understand how difficult to collect the long term results following surgery.

In that sense, they should be commended.

However, I have several major concerns about their results with their interpretations.

On T3 time point, 10/12 (left ATLR) and 12/13 (right ATLR) patients are seizure free. However, off-medication is still 4 and 6, respectively. Therefore, it needs to be explained why this discrepancy occurs.

For the connectivity analysis, they used the MTL seeds. However, considering that some of the MTL seeds are already removed, it needs to be explained how they put their seeds on the remaining MTL. In Fig. 2 and 3, their MTL seeds look rather posteriorly located. Is there any difference in seeds location depending on whether the MTL seed is put on the remaining one or the intact one?

Also there should be some explanation why they do not perform the whole brain analysis.

Also in verbal memory, the extra-MTL connectivity is reduced following the resection, whereas it increased in visual memory. I am wondering how and why this opposite reorganization ensues.

The change in MTL-connectivity is the difference between short term and long term connectivity. It should be compared with the difference between preoperative and short term connectivity as well.

In Fig. 1, the difference between the T2 and T3 is only in the verbal memory following the left section. The other differences are between the T1 and T2 in visual memory. Hence, why the MTL-connectivity changes despite no statistically significant difference between most T2 and T3 time period.

Relatedly, the memory outcome is diverse, roughly each one third improved, stationary or deteriorated. Hence in this scenario, the individual analysis may be better to the group analysis.

Also I have a concern about their cohort as well.

1. Whether they have hippocampal sclerosis preoperatively? If some of them have HS, the preoperative MTL connectivity might be different from that in the patients without HS.

2. Is there any difference in the extent of surgical resection of anterior temporal lobe and hippocampus?

3. Also what is the pathologies of the patients? Cortical dysplasia, tumors, or some others?

4. Also the period between the onset of epilepsy and the surgery time is controlled in their cohort. If the duration of medically intractable epilepsy differs in their cohort, the preoperative reorganization would be different as well.
5. They assessed the results from the 25 patients following anterior temporal resection. However, this is not the whole story, since their cohort might include others who may not be followed up or not be examined with fMRI. It should be disclosed.
6. There is no handedness information in both patients group and the control.

Reviewer #4

(Remarks to the Author)

The present study showed that patients who undergo anterior temporal lobe resection for epilepsy can experience long-term improvements in memory function up to a decade post-surgery. The brain demonstrates plasticity, allowing memory networks to reorganize and recover. The findings support the use of less invasive surgeries and ongoing cognitive rehabilitation to enhance long-term memory recovery in epilepsy patients. These findings significantly enhance our understanding of the brain's capacity for adaptation and functional reorganization, especially regarding memory, after epilepsy surgery. It reinforces the notion that with careful surgical planning and dedicated postoperative care, including cognitive rehabilitation, patients can look forward to a future where improvement in cognitive abilities is not only possible but sustainable over the long term.

1. Despite providing several important insights, this study carries weaknesses in that the study's small sample size limits the statistical reliability and necessitates cautious interpretation regarding the generalizability of the findings. In addition, the use of different scanner types during follow-up assessments could undermine measurement consistency, potentially limiting the comparability of the study results. Although this represents a missed opportunity, the findings from the available data are sufficiently striking and coherent to merit reporting.

2. The study acknowledges that surgical procedures can have varied effects on cognitive functions. Longitudinal studies with pre- and post-operative assessments, including a range of cognitive domains, could delineate the specific cognitive functions affected by surgery. Including a comparison group of patients with similar conditions but not undergoing surgery would help isolate the effects of the surgical intervention itself.

3. Given the heterogeneity observed in patient populations, which encompasses a spectrum of underlying conditions and comorbidities, these factors could notably affect cognitive outcomes. For example, the author may wish to elaborate on the potential confounding effects of medication changes on memory dysfunction. This is particularly significant in studies where neuropsychopharmacology interventions could be a key variable. Therefore, it may be necessary to closely examine the impact of medications on memory functions for each subject and understand how these changes might have shaped the study's conclusions.

4. Finally, the study's reliance on anatomical masks for determining seed regions for analysis could introduce limitations. While this method offers a straightforward approach based on specific brain areas, it might not fully reflect the actual neurobiological pathways, thus constraining the interpretation of the study results. More sophisticated methods for seed region selection or approaches that consider the whole-brain connectome may be required to support current findings.

Reviewer #5

(Remarks to the Author)

This study presents long-term memory network changes in patients who have undergone brain resection and the impact of plasticity on memory formation.

The authors evaluated using fMRI and memory performance data before brain resection, three to 12 months after resection, and 10 years later.

The results of this study observed specific changes in the memory network and identified regions that are important for long-term verbal and visual memory recovery.

Notable is that this study tracks fMRI and memory performance data in people with epilepsy (PWE) for up to ten years following anterior temporal lobe resection (ATLR).

I was very impressed with the patience and hard work of the researchers who waited until 10 years after the lobotomy to get the data.

The data obtained and analyzed in this study is worth a decade and is useful for analyzing long-term memory network changes and the impact of plasticity on memory formation.

However, the results of the study are preliminary and the evidence for the results presented is lacking.

1. Memory formation change

Currently, the explanation for these results is incomplete and requires further analysis.

Patients' memory performance after ATLR was not consistent across all patients, so it is necessary to analyze the data separately based on memory performance.

For example, individual L-ATLR subjects had 50% improvement and 33% decline in verbal memory and 42% improvement and 50% decline in visual memory performance, respectively.

When the memory performance result is divided into improvement, maintenance, and decline, the chance level is 33%. Therefore, a 33% decrease in verbal memory performance and a 42% improvement in visual memory performance are both above the chance level and therefore significant.

However, the authors do not explain and analyze these memory performance results, so further analysis is needed.

Also, while memory decline is typically expected with ATLR, this study suggests that memory performance improved at the group level.

As an explanation for this memory plasticity, it would be interesting to further analyze the fMRI data before and after ATLR. For example, if the connectivity supporting verbal or visual memory formation before ATLR was A, then after ATLR the A connectivity was disconnected and replaced by B connectivity.

It would be interesting to explain the neurological mechanisms or reasons for these connectivity changes or substitutions.

2. Changes in functional connectivity 3-12 months to 10 years post-surgery relative to changes in controls

The explanation for these results is incomplete and requires further analysis.

The results for the statistical analysis of functional connectivity given in Figures 2-(A) and 3-(A) are uncorrected.

This does not appear to solve the multiple comparison problem that occurs with fMRI voxel data.

Therefore, further analysis should be done to correct the p-value of that result to solve the multiple comparison problem.

Furthermore, the neurological implications of changes in functional connectivity in verbal and visual memory need to be explained.

Currently, it appears to simply list the results.

Therefore, it is necessary to supplement the explanation in the results or discussion.

3. Correlation between longitudinal changes in connectivity and memory recovery 3-12 months to 10 years postoperatively

The correlation of connectivity with verbal memory and visual memory performance needs to be presented in a figure.

The authors state in the main text that there was a significant correlation and presented a table. However, it would be nice to have a figure that makes sense intuitively.

Overall, the research is interesting and important.

Therefore, it would be good to supplement the explanation of the results and supplement the results with additional analysis.

Version 1:

Reviewer comments:

Reviewer #1

(Remarks to the Author)

The paper is much improved. I have no further questions.

Reviewer #2

(Remarks to the Author)

The authors have addressed my concerns.

Reviewer #3

(Remarks to the Author)

I appreciate the authors for their elaborate responses to my comments.

Although they have responded appropriately to my comments, their response to my comment 7.5 about the entire cohort would be better included in the supplement.

Otherwise, I have no further comment.

Reviewer #4

(Remarks to the Author)

I appreciate the author's point-by-point revisions and their acknowledgment of the generalizability issue, which enhance the understanding of the study's limitations and contributions.

I have a few additional comments:

1. You noted that ASM cessation improved visual memory recovery but not verbal memory. Could you speculate on this discrepancy? Are there specific antiepileptic medications that might have a more significant impact on cognitive recovery?

2. The finding that ongoing seizures 10 years post-surgery did not affect memory recovery seems counterintuitive. Could you discuss potential explanations for this?

3. I would appreciate a more detailed explanation of your choice to use the Reliable Change Index (RCI) for determining clinically significant change. How might results differ with alternative methods? This clarification would provide valuable insight into your findings.

Reviewer #5

(Remarks to the Author)

Dear Author.

I would like to express my sincere gratitude for your dedicated effort in revising and resubmitting the manuscript.

I previously provided the following comments:

1. Additional analysis on memory formation changes, particularly in relation to ATR and various contributing factors.
2. Further investigation into the relationship between memory formation changes, plasticity, and connectivity.
3. Correction for the multiple comparison problem in the statistical analysis of connectivity in the boxel data.
4. A detailed explanation of the neuroscientific significance of connectivity changes in verbal and visual memory.
5. Presentation of figures to illustrate the results.

In conclusion, you have addressed my comments comprehensively, and I appreciate the additional analyses you performed beyond my initial suggestions.

Regarding points 1 and 2, you appropriately constructed statistical models to examine the factors influencing memory formation changes and incorporated connectivity data to strengthen the credibility of the findings on memory formation changes and plasticity.

For point 3, you suggested that the results remained significant even after applying FWE correction for multiple comparisons, which is commendable.

In point 4, you provided further clarification on the dominant hemisphere associated with verbal and visual functions.

Lastly, in point 5, you introduced appropriate figures to support the results effectively.

I think the revisions have been executed appropriately, and I have only one minor suggestion.

- On page 8: For visual memory, it states "38% improved, 15% no change, and 38% declined.

The percentages sum to 91% (38+38+15), whereas the total should be 100%. Please verify and correct the values.

Other than this, I have no additional comments.

Once again, thank you for your revision, and I look forward to seeing the final version of this important research.

Best regards,
Seong jin Lee

REVIEWER COMMENTS

Reviewer 1

This longitudinal study aimed to identify neural mechanisms underlying verbal and visual memory network plasticity and recovery long-term after epilepsy surgery. The authors assessed changes in task-associated functional connectivity from 3–12 months to 10 years after anterior temporal lobe resection (ATLR) using generalized psychophysiological interaction (PPI) proposed by Friston et al, and correlated fMRI memory network alterations with improvement in episodic memory function. They focused on local mesial temporal lobe (MTL) connectivity, including connections to the fusiform gyrus based on the authors' previous work.

They found that from before to 3–12 months post-ATLR, healthy controls exhibited a verbal memory learning/test-retest effect absent in postsurgical patients, suggesting an immediate surgical insult on cognitive function. Beyond 3–12 months, their predominantly seizure-free cohort demonstrated cognitive stability at the group level, consistent with neuropsychological research. In addition, they showed that from 3–12 months to 10 years postoperatively, long-term reorganization toward posterior fusiform and right posterior parahippocampal gyri was significantly supportive of postoperative improvement in visual and verbal memory, supporting the role of posterior MTL regions in verbal memory formation. They concluded that specific changes in memory networks and identified regions crucial for long-term verbal and visual memory recovery.

I have the following questions/comments:

1. How many patients had the 10-year follow-up memory and fMRI evaluation? This should be clear in the main text. The supplemental materials could include details on the number of subjects in all time points.

We thank you for your review. All people with epilepsy (PWE) included in this study, specifically 25 ATLR cases (12 left-sided), returned for both the 10-year memory neuropsychometry and 10-year fMRI assessments. Both neuropsychology and memory fMRI paradigm were conducted on the same day, as such, anyone returning for follow-up underwent a comprehensive assessment, including both memory neuropsychometry and memory fMRI. We have clarified this point in the manuscript.

There were eight left-sided and 10 right-sided ATLR cases who completed assessments at all postoperative follow-ups, including the preoperative baseline, 3-month, 12-month, and 10-year assessments. Four individuals with left-sided and three with right-sided ATLR returned for the 3-month and 10-year assessments but did not undergo the 12-month follow-up. Therefore, to optimize acquired data, a 'short-term' assessment timepoint encompassed all data collected at 12 months, alongside data at 3 months from PWE unable to attend 12-month follow-up. This information is detailed in the Methods. In response to your comment, we have added a table in supplementary materials to more clearly outline the number of people at each assessment time point.

We have performed an additional sensitivity analysis, included in supplementary materials, which showed no significant difference between the neuroplasticity of the short-term cohort comprising of both 3-month and 12-month fMRI data and the brain changes of a cohort entailing 12-month data only. This justified using a combined 'short-term' timepoint with both 12-month and 3-month data when needed. We have however included this point in our discussion.

Changes made in the manuscript are highlighted in italics below.

Methods, Subjects, Page 31

“Twenty-five individuals with medically refractory TLE underwent ATLR from 2009 to 2012 at the National Hospital for Neurology and Neurosurgery (NHNN), London, United Kingdom.⁹ There were 12 left-sided ATLR (seven males, median preoperative age 38 years, interquartile range (IQR) 28-41) and 13 right-sided (four males, median preoperative age 38 years, IQR 29-50).

Neuropsychology assessment, as well as structural and memory functional MRI (fMRI) were acquired at four timepoints: preoperatively, and at a median 3-month (IQR = 3-4) and 12-month (IQR = 11-13.5) post-surgery, and up to 10 years postoperatively (median = 9, IQR = 8-10). *Neuropsychological and memory fMRI evaluations were conducted on the same day, ensuring comprehensive assessments at each follow-up.* Eight left-sided and 10 right-sided ATR cases completed assessments at all postoperative follow-ups. *All subjects included in this study returned for the 10-year neuropsychometry and fMRI assessments.*”

Methods, Statistical analyses, Longitudinal assessment of the functional memory network, Page 37

“All data was analysed with SPM12. *Shorter-term memory data consisted of a combination of both 12-month and 3-month timepoints for subjects who did not return at 12 months. Sensitivity analyses (see supplementary material) showed no significant difference between longitudinal connectivity changes in a mixed ‘3-12month’ cohort compared to those in a ‘12-month only’ cohort. This justified combining 3- and 12-month postoperative scans into the ‘short-term’ timepoint.*”

Discussion, Strengths and limitations, Page 29

“*The shorter-term timepoint consisted of a combination of 12-month and 3-month data for subjects who did not return at 12 months. We performed sensitivity analyses to verify that there was no significant difference in neuroplasticity effects between the mixed ‘3–12-month’ and ‘12-month only’ cohorts. Future studies should assess the plasticity of memory networks at more frequent, regular intervals to better understand the temporal evolution of the described plasticity changes.*”

Supplementary Results, Sensitivity analysis: Twelve months to 10 years plasticity, Pages 5-6

“*Current literature suggests that PWE may undergo distinct structural and functional reorganizations at 3 months compared to 12 months after surgery. The shorter-term follow-up (T2) consisted of a combination of both 12-month and 3-month timepoints for subjects who did not return at 12 months. To assess the impact on the robustness of long-term plasticity findings (T2 to T3 (10 years)), we compared longitudinal connectivity changes in the mixed 3-12-month follow-up cohort with those in a 12-month only cohort.*

For each subject, we subtracted the preprocessed, normalised contrast image¹ of successful memory connectivity from the subject-level PPI analysis at T3 from the contrast image at T2 and vice versa, using the SPM toolbox ‘ImCalc’. This yielded two ‘connectivity difference’ contrast images per subject, MTL seed, and memory task: increased connectivity and decreased connectivity at T3 compared to T2 (3 or 12 months, depending on the subject) over the whole brain. At the second level of the random effects analysis, Two-sample t-tests then compared these functional connectivity changes between the ‘3-12-month’ and ‘12-month only’ groups, separately for each ‘connectivity difference’ T-contrast, memory task, MTL seed, and ATR group.

In left ATR, this included 12 subjects in the combined ‘3-12-months’ cohort compared to eight in the ‘12-month only’ group, while in right ATR, there were 13 subjects in the combined cohort and 10 in the ‘12-month only’ group. At an exploratory threshold $P < 0.05$, uncorrected, there were no significant differences in longitudinal connectivity changes (no suprathreshold difference) between the ‘3-12-month’ and the ‘12-month only’ cohorts during successful encoding of either words or faces for each ‘connectivity difference’ T-contrast, seeding from either MTL in any ATR group. This is possibly due to the low number of 3-month compared to 12-month participants. Due to the lack of difference in connectivity changes, these groups were combined into a short-term postoperative group.”

Supplementary Methods, Subjects, Page 7

“*All subjects included in this study returned for the 10-year neuropsychometry and fMRI assessments. Supplementary table 1 below outlines the number people at each follow-up.*”

Table S1 Number of healthy controls and people who had anterior temporal lobe resection at each timepoint.

	Baseline	3 months	12 months	10 years
Controls	10	9	7	10
Left resection	12	12	8	12
Right resection	13	13	10	13

The number of individuals who had both neuropsychometry and neuroimaging assessments done at each experimental timepoint is outlined. This includes the preoperative assessment, and 3-month, 12-month and 10-year follow-ups after people underwent left- or right-sided resection, and the equivalent first and follow-up tests in healthy matched controls.

2. Please describe better how the reliable change index was defined.

2.1. Was this based on the controls' scores?

We thank you for your enquiry. The reliable change index was estimated using all participants' data, including scores of both PWE and healthy controls that were normalized (z-scores). We have added this information to our manuscript.

The RCI was calculated using a similar method as in Baxendale and Thompson (2020)¹. Baxendale and Thompson (2020) used standardised scores (z-scores) to do a RCI with 80% confidence interval (CI). We used a 95% CI (more stringent than 80% CI) to estimate longitudinal change in BMIPB z-scores, alike Chan et al. 2017². The published BMIPB test-retest reliability and standard deviation of normal population were used for converting raw scores into z-scores, which differed based on published BMIPB ageing norms. This allowed to adjust for both practice effect and age-related differences. We also corrected for change over time using the change in the study's healthy controls, as suggested by Chelune 1993³.

We used a critical value of 1.96 from the normal distribution to estimate the RCI with 95% CI using the formula: **Mean ± Critical Value x (standard deviation/√sample size)**. This entailed the below.

- 1) Calculating the change in z-scores between short and long-term follow-ups for every participant (including both PWE and healthy controls).
- 2) Computing the mean change in z-scores across follow-ups and standard deviation (STD) in the whole sample, including PWE and controls.
- 3) Estimate the 95% CI: 1.96 x (STD of change in z-scores/√sample size).
- 4) Find the RCI boundaries based on 95% CI: add or subtract the 95% CI margin to the mean change of the whole sample, i.e., mean change ± 95% CI margin.

As such, a reliable change in z-scores across follow-ups included any change that is above or below the RCI boundaries (mean change ± 95% CI margin).

Changes to the manuscript are highlighted in italics below.

Methods, Neuropsychological tests, Page 33

“Using the mean and standard deviation of the whole sample, the RCI probes meaningful change by adjusting for test reliability and practice effect in a test-retest context.”⁵⁹

2.2. Reliable Change Index calculations created with data from healthy control groups may exaggerate the decline when used with epilepsy patients. This should be commented on in the discussion.

We appreciate the limitation of only using healthy control data which is why we adopted an RCI approach that is based on the standard deviation of the whole sample (PWE and controls). All PWE and controls were scanned at equivalent timepoints. Using this method, our data did not reflect an overestimation of decline in PWE. Based on the RCI from the short-term to long-term follow-ups, our healthy control sample showed a significant decline at the group-level in verbal memory that was not seen in the patient population – people who underwent left ATR showed a significant verbal memory improvement while there was cognitive stability at the group-level in right ATR for both verbal and visual memory based on the RCI.

In healthy controls, there was a significant improvement in verbal memory from baseline to short-term assessments. This suggests that the observed memory decline in the longer term may be due to a practice effect seen at shorter-term follow-up, which resolves over time, indicating overall cognitive stability. We have discussed this in our manuscript. Lastly, all neuroimaging analyses were performed using IQ as a regressor of no interest (indicated in Methods) further removing potential bias due to differences in individual intelligence levels.

Changes to the manuscript are highlighted in italics below.

Discussion, Long-term postoperative memory changes, Pages 22-23

“Over almost a decade, healthy controls exhibited *an initial improvement in verbal memory from the baseline to 3–12-month assessments followed by a significant decline at 10 years. This pattern suggests an early practice effect that resolved over time, with similar mean performances at baseline and 10 years indicating overall stable cognitive function.*

Neuropsychology studies demonstrate that ATLR is characterized by an immediate ‘hit’ on episodic memory functions^{1,22}, which in seizure-free people is followed by either cognitive stabilization or ongoing improvement as convalescence time increases (beyond two years).^{1,2,23} In line with current literature, postsurgical PWE did not present with the learning/test-retest effect in verbal memory, suggesting an early surgical insult on memory function. Yet beyond 3–12 months, our predominantly seizure-free cohort demonstrated cognitive stability at the group-level and patient-specific recovery was evident.”

Discussion, Strengths and limitations, Page 29

“Healthy controls showed a significant test-retest *influence* on verbal memory, *at the initial short-interval follow-up, which was not seen in PWE likely due to the immediate surgical impact on function.*”

3. Preoperatively and short-term postoperatively images were acquired in a scanner different from the long-term follow-up, as expected due to the end-of-life and upgrades in these machines. In addition, the head coils were different (20-channel head coil in the first scanners versus 32-channel head coil in the follow-up scanners). fMRI data are quite sensitive, and the signal varies according to different scanners and head coils with more channels. How were these differences handled?

We thank you for your comment and understanding of the challenges inherent to the experimental design. There is likely to be scanner effects using different scanners, however we have modelled for this as below.

- (a) We have data collected at 3 months, 1 year and about 10 years post-surgery. Scanner effect and effect of ageing was controlled for by (1) having healthy controls that were age-matched and scanned at similar intervals (same scanner as PWE interval scans), and (2) using a flexible factorial design in the general linear model that compares changes across time-points in PWE compared to changes across timepoints in controls. Modelling the data in this way allows the investigation of changes due to the experimental condition rather than scanner effect changes that are controlled for. For example, PWE who had left surgery 10y – 1y over and above changes in controls from 10y – 1y : contrast [1 -1 -1 1 0 0].
- (b) To further address this important point raised by the reviewers, we sought the opinion of Dr Peter Zeidman, chair of the SPM Methods Group, at the UCL Wellcome Centre for Human Neuroimaging. He suggested that we have adequately modelled for this and provided an explanation below. We have however added this limitation inherent to our longitudinal long-term study design to our discussion.

“Due to the change of scanner, [we] cannot exclude that any difference in functional connectivity may also be due to change in scanner. However, the reported longitudinal differences in task-based functional coupling between brain regions represent interaction effects (time-point x task condition) that are orthogonal to the main effect of time-point and therefore are distinguishable.

Additionally, in modelling the GLM using a flexible factorial design, [we] allowed for differences in noises between scanner, i.e., [we] modelled any possible heteroskedasticity. Modelling heteroskedasticity means that [we] allowed for a difference in residual variance between scanners, meaning that the variance of the errors from each scanner is not constant over time.”

We have made clearer that these effects are controlled for by the parameters of flexible factorial analysis. We have also added the limitation of change of scanner to our limitations.

Methods, Statistical analyses, Long-term changes in functional connectivity from 3–12-month to 10-year follow-ups, Pages 38-39

“The effect of scanner change between short-term and long-term timepoints was addressed using specific parameters of the flexible factorial design. This design modelled any possible heteroskedasticity, allowing for differences in residual variance between scanners. Longitudinal differences in task-based functional coupling between brain regions represented interaction effects (time-point x task condition) that are orthogonal to the main effect of time-point, making them distinguishable. In summary, flexible factorial t-contrasts modelled the within-subject differences in MTL-seeded, whole-brain functional connectivity across postoperative follow-ups in each ATR group, beyond the connectivity changes seen in controls. This model allowed control of both age-related and scanner-change effects.”

Discussion, Strengths and limitations, Page 30:

“Over a 10-year period, scanner changes were inevitable. To address this, we used a flexible factorial design that accounted for potential heteroskedasticity and consulted with the UCL Wellcome Centre for Human Neuroimaging; longitudinal differences in connectivity reflected interaction effects between time-point and memory task, which are orthogonal to the main time-point effect and therefore distinguishable from effects related to scanner change.”

4. The last paragraph of the supplementary text is truncated. “Single-level gPPI t-contrasts of successful subsequent memory were applied to group-level random-effects analyses for functional connectivity estimation and comparison across groups (see below section).” There is no section below that. Please add the pertinent information about the group comparisons.

We thank you for highlighting this. This represents the last paragraph of Supplementary Methods as the detailed information on group comparisons is provided in the main manuscript (as shown below). We have amended the supplementary text to reflect this; changes are highlighted in italics.

Supplementary Methods, Data analysis, Page 10

“Single-level gPPI t-contrasts of successful subsequent memory were applied to group-level random-effects analyses for functional connectivity estimation and comparison across groups. Group analyses are described in the main manuscript’s Methods – longitudinal assessment of the functional memory network.”

Methods, Statistical analyses, Long-term changes in functional connectivity from 3–12-month to 10-year follow-ups, Page 37

“Mixed ANOVAs within a flexible factorial design with both IQ and change in ASM intake as confound regressors, were performed to investigate changes in MTL-seeded memory connectivity between short-term and long-term follow-ups in the individual patient groups compared to changes in test-retest over that timeline in healthy subjects. A distinct analysis was conducted for faces and words successfully remembered, seeding from each MTL separately.

For each subject, the relevant first-level gPPI t-contrasts for each of the scanning timepoint (3–12 months and 10 years) were entered. Each flexible factorial design involved a random subject factor, a three-levels group factor (controls, left and right ATR), and a two-levels condition factor (short-term and long-term postoperative scans), allowing the investigation of a Group x Condition interaction for the successful memory contrasts. Differences in activations across scanning sessions were compared between ATR groups and controls in t-contrasts: 10-year connectivity > 3–12-month connectivity in left or right ATR versus controls, and 10-year connectivity < 3–12-month connectivity in left or right ATR versus controls [...].”

Methods, Statistical analyses, Correlation between functional reorganization in TLE and memory recovery, Page 38

“Three-way ANCOVAs, controlling for IQ and changes in ASM intake, were conducted for each MTL seed and successful memory contrast (words/faces) to investigate differences in functional connectivity from 3–12-month and 10-year follow-ups related with improvement in memory functions over this timeline.

Specifically, individual changes in memory z-scores were correlated with changes in task-modulated fMRI connectivity over time, modelling the fMRI biomarkers more strongly associated with those whose memory successfully improved compared to those who did not. Differences in z-scores (i.e., BMIPB I or II verbal and visual learning scores converted into age-normalized z-scores) from 3–12 months to 10 years were used as continuous variables.”

- Minor typo on Page 7 of the main text under Healthy controls: “...improved, 50% no change, while 37.5% declined. At the group level, there was a clinically significant improvement in verbal memory from preoperative to 3–12 months timepoints ...” - It should be **‘baseline’** instead of “preoperative.”

We thank you for highlighting this matter. We have amended the manuscript accordingly. Changes are highlighted in italics as below.

Results, Long-term change in postoperative memory, Page 8

“From 3–12 months to 10 years, healthy controls’ memory was as follows: [...] At the group-level, there was a clinically significant improvement in verbal memory from *baseline* to 3–12 months assessments, which was reversed from 3–12 months to 10 years (i.e., significant decline).”

Reviewer 2

The paper is of some interest give the data regarding neural mechanisms of memory recovery and network plasticity over nearly a decade post-surgery. Fleury et al. assessed memory network changes in 25 patients (12 left-sided resections VS 13 left-sided resections) relative to 10 healthy matched controls, using longitudinal, task-based functional MRI and neuropsychology assessments. In the seizure-free cohort, group-level memory either stabilized or improved in the long-term post-resection. Increases in functional connectivity that supported long-term memory recovery projected to medial temporal and extra-temporal structures. Reduced involvement of the thalamus may serve as an imaging biomarker of cognitive network convalescence. Although I agree with the authors that the findings of the study have the potential to impact surgical intervention to avoid crucial regions of memory reorganization, there major limitations that would need to be addressed to increase the reliability and stability of the findings.

- The sample size of the study was too small to make a robust conclusion: it could not provide a group with a poor prognosis.

We thank you for your comment. Despite small sample sizes, we were able to demonstrate statistically significant gPPI activation maps, which suggests reliable results. For the scope of this study, we report group differences using specific contrast images from a flexible factorial design, hence the chosen statistical thresholding, but group analyses of memory activations show robust connectivity at family-wise error (FWE) rate. As an example, we have inserted the result table of group analyses of successful memory connectivity 10 years after surgery, at FWE cluster-extent thresholding.

Table: Cross-sectional connectivity of the successful memory networks at the 10-year follow-up, corrected using family-wise error rate.

		Left ATLR	Right ATLR	Controls
Verbal Memory	Left MTL seed	Left post hippocampus -20 -28 -6, $P = 0.003^b$	Left ant hippocampus -30 -8 -22, $P = 0.017^a$	Left ant hippocampus -30 -28 -12, $P = 0.019^a$
		Left post fusiform (temporal) -44 -46 -20, $P = 0.001^a$	Left post parahippocampal G -24 -30 -12, $P = 0.026^a$	Left mid-post hippocampus -26 -24 -16, $P = 0.050^a$
		Left post parahippocampal G -28 -38 -12 $P = 0.015^b$	Left post fusiform (temporal) -38 -24 -18, $P < 0.001^a$	Left mid fusiform -28 -40 -20, $P = 0.030^a$
		Right mid hippocampus 18 -24 -12, $P < 0.002^a$	Right post fusiform (temp-occ.) 24 -70 -6, $P = 0.004^a$	Right post fusiform (occ.) 22 -50 -12, $P = 0.043^a$
		Right mid-post fusiform (temp-occ.) 36 -54 -18, $P < 0.001^a$	Right post parahipp./fusiform G 36 -38 -14 $P = 0.013^a$	
		Left inferior temporal gyrus -50 -52 -18, $FWEc = 0.025$	Left inferior parietal gyrus -58 -34 50, $FWEc = 0.001$	
		Left post cingulate cortex -8 -42 8, $FWEc = 0.021$		

	Right precuneus 4 -54 70, FWEc = 0.047			
Right MTL seed	Left post hippocampus -36 -28 -4, $P = 0.021^b$	Left amygdala -28 -8 -12, $P = 0.044^a$	Left post fusiform (occ.) -26 -68 -14, $P = 0.007^a$	
	Left post fusiform (temp-occ.) -34 -62 -16 $P < 0.001^a$	Left mid hippocampus -26 -22 -12, $P = 0.018^a$	Left mid fusiform (temporal) -28 -46 -20, $P = 0.036^a$	
	Right ant hippocampus 32 -18 -20, $P = 0.029^a$	Left post fusiform (temp-occ.) -22 -76 -14, $P = 0.001^a$	Right ant-mid hippocampus 34 -18 -8, $P = 0.035^a$	
	Right post hippocampus 26 -30 -10, $P = 0.019^a$	Left mid fusiform gyrus (temp.) -28 -38 -24, $P = 0.002^a$	Right post hippocampus 30 -30 -8, $P = 0.025^a$	
	Right post parahippocampal G 18 -40 -4, $P = 0.015^a$	Left ant parahippocampal G -30 -12 -26, $P=0.026^a$	Right post fusiform (occ.) 46 -70 -18, $P = 0.043^a$	
	Right mid fusiform (temp.) 28 -38 -22, $P = 0.029^a$	Left ant hippocampus -34 -14 -22, $P=0.007^a$		
	Right post fusiform (temp-occ) 24 -84 -16, $P = 0.001^a$	Right post hippocampus 26 -36 -6, $P = 0.049^b$		
		Right mid-post hippocampus 34 -18 -8, $P = 0.047^b$		
		Right post parahippocampal G 24 -44 -4, $P = 0.018^b$		
		Right post fusiform (temp-occ.) 28 -78 -16, $P=0.025^a$		
	Right mid-post fusiform (temp.) 38 -36 -28, $P<0.001^a$			
	Left inferior parietal G -56 -24 48, FWEc < 0.001			
	Right mid frontal G 40 -2 60, FWEc = 0.035			
Left MTL seed	Left post hippocampus -20 -32 -6, $P = 0.012^b$	Left ant hippocampus -30 -8 -22, $P = 0.017^a$	Left ant hippocampus/amygdala -26 -8 -20, $P = 0.041^a$	
	Left post fusiform (lateral-occ.) -44 -46 -20, $P = 0.001^a$	Left mid parahippocampal G -30 -26 -1, $P = 0.021^a$	Left mid hippocampus -26 -24 -16, $P = 0.050^a$	
	Left post parahippocampal G -26 -38 -12 $P = 0.017^b$	Left post fusiform (temp-occ.) -34 -62 -8, $P = 0.005^a$	Right post fusiform (temp-occ.) 24 -50 -12, $P = 0.043^a$	
	Right mid [para]hippocampus 18 -24 -12, $P= 0.002^a$	Left mid-post fusiform (temp) -38 -24 -18, $P < 0.001^c$	Right mid parahippocampal G 26 -26 -16, $P = 0.043^a$	
	Right mid fusiform (temporal) 34 -44 -18, $P = 0.024^a$	Right post fusiform (temp-occ.) 24 -70 -6, $P = 0.004^a$		
	Right post fusiform (temp-occ.) 36 -54 -18, $P < 0.001^a$	Right post parahippocampal G 36 -38 -12, $P = 0.008^b$		
	Left inferior TG/lateral fusiform -44 -46 -20, FWEc = 0.005	Left lingual G 2 -80 -2, FWEc = 0.002		
Visual Memory		Right Cuneus 14 -72 24, FWEc=0,002		
	Left post fusiform (occ.) -34 -82 -16, $P < 0.001^a$	Left mid hippocampus -34 -22 -14, $P = 0.001^a$	Right mid-post hippocampus 28 -28 -6, $P = 0.029^a$	
	Right mid hippocampus 18 -24 -10, $P = 0.008^a$	Left post fusiform (temp-occ.) -32 -52 -6, $P = 0.006^a$		
	Right ant parahippocampal G 22 -2 -24, $P = 0.002^a$	Left mid parahippocampal G -28 -28 -14, $P=0.038^a$		
	Right post fusiform (temp-occ.) 34 -48 -16, $P = 0.001^a$	Right post fusiform (temporal) 30 -34 -20, $P = 0.003^a$		
		Right post fusiform (lateral-occ.) 42 -58 -22, $P = 0.008^a$		
		Right post parahippocampal G 34 -34 -16, $P<0.001^b$		
		Left mid temporal G -44 -52 -4, FWEc = 0.013		
	Right MTL seed			

Mean functional connectivity as measured by one-sample t-tests separately for each MTL seed and subsequent memory fMRI task. MTL-to-neocortex connectivity is reported at voxel-wise cluster-defining threshold $P < 0.001$, corrected using cluster-extent family-wise error (FWE) rate < 0.05 (i.e., FWEc). ^a $P < 0.05$, FWE correction using a 6mm sphere in contralateral MTL regions. ^b $P < 0.05$, FWE correction using a 3mm sphere in remnant MTL regions.

The memory task has been refined to allow for reliable individual mapping of preoperative memory function that can be used presurgically. Indeed, individual frontal temporal activations within a mask are predictive of verbal memory outcome with 87.5% sensitivity and 80% specificity⁴. The finding of significant activations both at the individual and group-level suggests the robustness of the fMRI task. Other study with small sample size showing significant results include a previous longitudinal study; using similar PPI analysis (ANOVA and paired t-test) of verbal fluency task in 14 people who had add-on perampanel⁵.

This is the first study that reports gPPI connectivity for memory; other groups have looked at connectivity of resting-state fMRI in group of 9 people with gliomas⁶ to map language function and another compared connectivity concordance of verbal generation fMRI with that of resting-state fMRI in 3 people with gliomas⁷, and found comparable patterns of activation. Task-based connectivity compared to resting-state maximizes the reliability of functional connectivity specifically for task-engaged regions in the functional network of interest⁸. Smaller numbers than our study number have been used in activation-based univariate analyses (9 left TLE, 8 right TLE)⁹ to look at changes in scene-encoding fMRI patterns long-term post-epilepsy surgery. Although similar, functional connectivity analysis is a more sensitive method than univariate analysis for describing network-level functional architecture^{10,11}.

Nevertheless, the majority of our patients were seizure free. We appreciate the limitations of our study that primarily consisted of a homogeneous seizure-free cohort which limits the generalisability of our results to seizure-free patients. We have added this limitation, highlighted in italics below.

Discussion, Strengths and limitations, Page 28:

“[...] The postsurgical cohort was primarily seizure-free, limiting the generalisability of our results to people with good prognosis. Future studies with larger numbers should investigate the modulatory effect of specific ASMs and differential seizure outcomes on network recovery.”

2. Despite the efforts made by the authors to mitigate scanner-related effect, I do not think this is sufficient to address this issue, especially when the sample size is small.

We thank you for your comment. We have discussed sample sizes as above in response 1.

The scanner related challenge is inherent to the experimental design. There is likely to be scanner effects using different scanners, however we have modelled for this as below.

- (a) We have data collected at 3 months, 1 year and about 10 years post-surgery. Scanner effect and effect of ageing was controlled for by (1) having healthy controls that were age-matched and scanned at similar intervals (same scanner as PWE interval scans), and (2) using a flexible factorial design in the general liner model that compares changes across time-points in PWE compared to changes across timepoints in controls. Modelling the data in this way allows the investigation of changes due to the experimental condition rather than scanner effect changes that are controlled for. For example, PWE who had left surgery 10y – 1y over and above changes in controls from 10y – 1y : contrast [1 -1 -1 1 0 0].
- (b) To further address this important point raised by the reviewers, we sought the opinion of Dr Peter Zeidman, chair of the SPM Methods Group, at the UCL Wellcome Centre for Human Neuroimaging. He suggested that we have adequately modelled for this and provided an explanation discussed in response to reviewer 1’s comment (reviewer 1 response 3, Page 6).

We have made clearer that these effects are controlled for by the parameters of flexible factorial analysis. We have also added the limitation of change of scanner to our limitations (see reviewer 1 response 3, Page 6).

3. Using functional magnetic resonance imaging (fMRI) data from two scanners to explore changes in brain function over a span of approximately 10 years is significantly influenced by numerous confounding factors. Therefore, it's necessary to increase the sample size and collect fMRI data at multiple time points post surgically, as well as another dataset to further validate the stability of the results.

We thank you for your comment. There will invariably be changes related to scanner differences and we have therefore modelled this matter using a flexible factorial design (see above response 2 and reviewer 1 response 3, Page 6). Scanner effect and effect of ageing is controlled for by having healthy controls that are age-matched and scanned at similar intervals in the design. We have made clearer that these effects are controlled for by using flexible factorial design in our analysis.

Regarding the timeline of data collection, we have data acquired at 3 months, 12 months and about 10 years post-surgery. Ethical approval was sought for the 3 stipulated timepoints after surgery.

We have performed a sensitivity analysis, included in supplementary materials, to evaluate the robustness of our results by comparing the neuroplasticity of the short-term cohort comprising of 3-month and 12-month data with the neuroplasticity of a cohort entailing 12-month data only. This showed no significant difference between the longitudinal connectivity changes between cohorts.

We however agree that future research should look at further timepoints in between and they will be included for future studies. We have included this in limitations.

With regards to sample size, the memory task has been refined to allow for reliable individual mapping of preoperative memory function that can be used presurgically. Indeed, individual frontal temporal activations within a mask are predictive of verbal memory outcome with 87.5% sensitivity and 80% specificity⁴. The finding of significant activations both at the individual and group-level suggests the robustness of the fMRI task. Other study with small sample size showing significant results include a previous longitudinal study; using similar PPI analysis (ANOVA and paired t-test) of verbal fluency task in 14 people who had add-on perampanel⁵.

This is the first study that reports changes in gPPI connectivity for memory; other groups have looked at connectivity of resting-state fMRI in group of 9 people with gliomas⁶ to map language function and another compared connectivity concordance of verbal generation fMRI with that of resting-state fMRI in 3 people with gliomas⁷, and found comparable patterns of activation. Task-based connectivity compared to resting-state maximizes the reliability of functional connectivity specifically for task-engaged regions in the functional network of interest⁸. Smaller numbers than our study number have been used in activation-based univariate analyses (9 left TLE, 8 right TLE)⁹ to look at changes in scene-encoding fMRI patterns long-term post-epilepsy surgery. Although similar, functional connectivity analysis is a more sensitive method than univariate analysis for describing network-level functional architecture^{10,11}. We however appreciate the limitation of our small sample size, and it is discussed in our limitations.

Changes made in the manuscript are highlighted in italics below. For changes related to sensitivity analysis, please refer to reviewer 1 response 1, Pages 3-4, and for those related to scanner change see reviewer 1 response 3, Page 6.

Discussion, Strengths and limitations, Pages 28-29

“The postsurgical cohort was primarily seizure-free, limiting the generalisability of our results to people with a good prognosis. Future studies with larger numbers should investigate the modulatory effect of specific ASMs and differential seizure outcomes on network recovery. [...]

The shorter-term timepoint consisted of a combination of 12-month and 3-month data for subjects who did not return at 12 months. We performed sensitivity analyses to verify that there was no significant difference in neuroplasticity effects between the mixed ‘3–12-month’ and ‘12-month only’ cohorts. Future studies should assess the plasticity of memory networks at more frequent, regular intervals to better understand the temporal evolution of the described plasticity changes”

4. For those patients who did not attend for a 12-month follow-up, the authors used the 3-month data instead, which may influence the results, since the existing literature suggest that patients may have distinct structural and functional reorganizations at the time of 3-month post-surgery and 12-month.

It is better to scan the patient the same time-interval.

We thank you for your comment. In response to your review, we have performed a sensitivity analysis, testing in each ATR group for differences in longitudinal, long-term connectivity changes between our combined cohort of 3-month and 12-month data and a cohort of only people with 12-month data. In each ATR group, there were no significant differences between the functional connectivity changes from 3-12 months to 10 years and those from 12 months to 10 years. This is probably related to the low number of people who did not return at 12 months (4 in left ATR and 3 in right ATR). We have added this analysis to supplementary information. We however appreciate the limitation of combining both 3-month and 12-month data for assessing shorter-term memory. People do not always want to return and future studies should look at memory plasticity at more frequent, regular intervals. We have included this in our limitations.

Please see reviewer 1 response 1, Pages 3-4 for the sensitivity analysis’ Supplementary Results and related changes to Methods and Discussion.

5. The aging effect of memory decline was not considered in the analysis especially at the ten-year follow-up.

Thank you for your comment. The effect of ageing on memory function was accounted by scanning healthy matched controls at similar intervals as PWE and including them in the longitudinal analysis of connectivity changes in PWE. With a flexible factorial design, the general linear model removes all changes that are also seen in controls so that only longitudinal changes that are statistically significant in PWE compared to controls are visualized on the glass brain. As such, the flexible factorial design allows to model changes in PWE over and above changes in controls, accounting for ageing effects (see responses 2 and 3). We have made this clearer in our methods.

Methods, Statistical analyses, Long-term changes in functional connectivity from 3–12-month to 10-year follow-ups, Pages 37-38

“The effect of scanner change between short-term and long-term timepoints was addressed using specific parameters of the flexible factorial design. This design modelled any possible heteroskedasticity, allowing for differences in residual variance between scanners. Longitudinal differences in task-based functional coupling between brain regions represented interaction effects (time-point x task condition) that are orthogonal to the main effect of time-point, making them distinguishable. In summary, flexible factorial t-contrasts modelled the within-subject differences in MTL-seeded, whole-brain functional connectivity across postoperative follow-ups in each ATR group, beyond the connectivity changes seen in controls. This model allowed control of both age-related and scanner-change effects.”

Methods, Statistical analyses, Correlation between functional reorganization in TLE and memory recovery, Page 38

“Three-way ANCOVAs, controlling for IQ and changes in ASM intake, were conducted for each MTL seed and successful memory contrast (words/faces) to investigate differences in functional connectivity from 3–12-month and 10-year follow-ups related with improvement in memory functions over this timeline.

Specifically, individual changes in memory z-scores were correlated with changes in task-modulated fMRI connectivity over time, modelling the fMRI biomarkers more strongly associated with those whose memory successfully improved compared to those who did not. Differences in z-scores (i.e., BMIPB I or II verbal and visual learning scores converted into age-normalized z-scores) from 3–12 months to 10 years were used as continuous variables.”

Reviewer 3

Despite the importance of early memory network changes following anterior temporal lobe resection for temporal lobe epilepsy, long-term plasticity and its impact on memory function are unclear. Hence, the authors investigated neural mechanisms of memory recovery and network plasticity over nearly a decade post-surgery.

From 3–12 months to 10 years postoperatively, they assessed memory network changes in 25 patients (12 left-sided resections) relative to 10 healthy matched controls, using longitudinal, task-based functional MRI and standard neuropsychology assessments.

They observed specific changes in memory networks and identified regions crucial for long term verbal and visual memory recovery. Since their postoperative cohort is predominantly seizure-free, they highlight the potential for sustained cognitive improvement and reduced medication needs over a decade after epilepsy surgery.

These findings endorse strategic approaches in epilepsy treatment: advocating for conservative surgeries and promoting the long-term use of cognitive rehabilitation for ongoing recovery.

This study is very difficult to perform since they have to follow their patient until 10 years postoperatively. I fully understand how difficult to collect the long term results following surgery. In that sense, they should be commended.

However, I have several major concerns about their results with their interpretations.

1. On T3 time point, 10/12 (left ATR) and 12/13 (right ATR) patients are seizure free. However, off-medication is still 4 and 6, respectively. Therefore, it needs to be explained why this discrepancy occurs.

Thank you for your enquiry. Generally, after epilepsy surgery, medications are reduced 12 months postoperatively. This is a discussion between PWE and their clinical consultant. In some cases, there was reluctance to reduce ASM intake due to the potential impact on driving. In others, some PWE decided to reduce the doses but not to come off ASM altogether. This study team was not involved in the clinical decisions regarding ASMs.

2. For the connectivity analysis, they used the MTL seeds. However, considering that some of the MTL seeds are already removed, it needs to be explained how they put their seeds on the remaining MTL. In Fig. 2 and 3, their MTL seeds look rather posteriorly located. Is there any difference in seeds location depending on whether the MTL seed is put on the remaining one or the intact one?

Seed selection was defined by atlas-based anatomical masks: mask of the non-resected hippocampus and left/right ATR group masks of the surgically spared hippocampus and parahippocampus. Using anatomical masks for PPI seeds allows non-bias seed selection¹². PPI only models the change in interaction between the seed ROI and other brain regions, therefore ensuring that any non-task specific baseline interactions related to anatomical proximity (e.g., between masked regions) will be disregarded if it is not specifically task-related¹².

We informed the seed selection based on memory literature. Since the 1900s, substantial evidence underscores the significance of structures of the MTL in memory formation. Before surgery for intractable epilepsy, memory fMRI has demonstrated that in people with TLE, fMRI activation maxima may shift from the typical anterior hippocampus to the posterior hippocampus^{13,14}. Following unilateral ATR, memory fMRI studies report fMRI activation maxima in the posterior, surgically spared, hippocampus and parahippocampus during successful word and face encoding^{15,16}. Accordingly in our study, the global fMRI maxima at the group level also laid in the posterior hippocampus. Prior to performing 1st level gPPI analysis, we also performed a visual check ensuring that every participant had an activation peak within each hippocampus and remnant MTL mask used for gPPI seed selection.

Consequently, to study the pathological postoperative memory connectome, the MTL seeds entailed (1) the non-resected hippocampus to address eventual neurobiological shift in activation fMRI maximum from anterior to posterior hippocampus, and (2) the surgically spared (remnant) posterior hippocampus and parahippocampus to address potential activation fMRI maxima shift from the hippocampus to parahippocampus.

3. There should be some explanation why they do not perform the whole brain analysis.

We thank you for your comment. Although the seeds were taken from the MTL, this was a whole-brain connectivity analysis.

Different approaches to functional connectome generation exist, involving data-driven and model-based methods. Psychophysiological interaction (PPI)¹⁷ is a data-driven, directed functional connectivity approach that uses regression to explore increases in the relationship between a region of interest (ROI) and the rest of the brain resulting from experimental manipulation¹⁷. Anatomically adjacent brain regions share similar time courses that are correlated at rest^{18,19}. To model cortical correlations that vary with task performance, PPI analysis generates an interaction regressor (the PPI term) derived from the timeseries of a specific 'seed' ROI and the task time course. The influence of one neural system (i.e., the seed) on another is modelled at the whole-brain level, while accounting for the experimental context (i.e., context-dependent)^{12,17}. That is, PPI models the moderator effect of task performance on the functional connectivity between a ROI *a priori* known to be part of the whole-brain, functional connectome and the rest of the brain.

We informed the seed selection based on memory literature (see above response 2). Seed selection was performed using anatomical masks of the non-resected hippocampus and surgically spared hippocampus and parahippocampus for non-bias seed selection¹². PPI only models the change in interaction between the seed ROI and other brain regions, therefore ensuring that any non-task specific baseline interactions related to anatomical proximity (e.g., between masked regions) will be disregarded if it is not specifically task-related¹².

As such, seed-to-voxel-wise PPI analysis allows to investigate task-modulated changes in the whole-brain connectivity using a starting point in a brain region *a priori* known to be part of the functional network to model the whole-brain memory connectome.

4. In verbal memory, the extra-MTL connectivity is reduced following the resection, whereas it increased in visual memory. I am wondering how and why this opposite reorganization ensues.

We thank you for your review. Task lateralization appears to vary according to cognitive demands, as shown in previous studies: left hemispheric lateralization during successful word encoding and bilateral temporal involvement during successful face encoding¹³. Verbal mnemonics exhibit greater lateralization to the speech dominant hemisphere and incurs semantics for learning, whereas the visual memory network entails a more bilateral network with learning strategies that can differ among individuals. Sidhu et al. (2016)¹⁶ demonstrated that 12 months after surgery, increases in extra-temporal activations contralateral to the side of surgery (i.e., in the anterior cingulum, orbitofrontal cortex, and insula) support postoperative verbal and visual memory functions. By 10 years, ongoing neocortical plasticity within the anterior cingulum is observed in the verbal memory network after left ATR, potentially due to a greater impact of left hemispheric resection compared to the right. Long-term reorganization of the visual memory network continues after both left and right ATR, likely due to its extensive and bilateral nature encompassing medial temporal, temporal, and frontal regions relevant to visual tasks.

We have discussed this task-dependent differential trend in longer term extra-MTL plasticity as below. Changes to the manuscript are highlighted in italics.

Discussion, Long-term support from structures of goal-directed behaviours, Page 26

“For verbal memory recovery, the importance of right frontal engagement was enhanced 3–12 months after both ATR compared to preoperatively, including the right middle cingulate and anterior orbitofrontal cortices. The absence of long-term frontal connectivity decreases suggests this plasticity was sustained and adaptive 10 years post-surgery. There were ongoing connectivity increases between remnant MTL and contralesional anterior cingulum long-term after left ATR, possibly translating for a greater impact of left than right-sided resection on verbal mnemonic processes.

Reorganization of verbal memory function to the cingulate cortex is critical in TLE. PWE exhibit memory fMRI activations in the anterior cingulate cortex that support successful word encoding both before and short-term after surgery^{7,40} and correlate with improvement 12 months after left ATR⁹.

[...]”

Discussion, Task-specific patterns of extra-MTL plasticity, Pages 27-28

“The long-term plasticity of extra-temporal memory networks varied with memory. Visual memory recovery underpinned a bilateral network involvement with compensatory mechanisms engaging both temporal and frontal regions a decade postoperatively, while verbal memory displayed a more lateralized pattern of compensation, primarily involving the reinstatement of posterior MTL plasticity disconnected during surgery.

Preoperatively, successful visual memory encoding correlates with activation fMRI maxima in both anterior hippocampi, and with the bilateral middle and inferior temporal cortices and right frontal areas such as the right orbitofrontal cortex in healthy subjects^{7,16}. This extensive network’s architecture may explain the observed, long-term increases in connectivity with neocortical areas typically part of the episodic memory network. Such neuroplasticity adaptively compensated for functional loss in the ipsilateral MTL following right ATR, and for the disruption of the bilateral temporal cognitive network after left ATR.

Verbal memory encoding is predominantly lateralized to the left hemisphere in healthy subjects^{7,48,49}. As such, individuals undergoing right-sided resection may maintain nearly intact, established neocortical networks of verbal memory arising from the left MTL. This may account for the observed long-term plasticity after right ATLR, which, beyond 3–12 months, primarily involved increased connectivity within the remnant right MTL.

These differential plasticity patterns underscore the dynamic nature of cognitive network architectures, adapting according to the task's functional demands and learning strategies involved, as previously proposed⁵⁰."

5. The change in MTL-connectivity is the difference between short term and long term connectivity. It should be compared with the difference between preoperative and short term connectivity as well.

We thank you for highlighting this matter. To address your comment, we have analysed the changes in MTL-seeded connectivity from preoperative to 3-12month post-surgery which supported long-term improvement in memory.

We found that early after surgery, disconnected preoperative connectivity is replaced by new connectivity patterns to anterior contralateral MTL regions which efficiently supports longer term verbal and visual memory functions, while connections with posterior MTL regions are not immediately efficient. In PWE who continue to improve, posterior MTL regions like the fusiform and parahippocampus become efficiently involved the cognitive network over the longer term. For verbal memory recovery, increased plasticity in the shorter term to the right frontal cortex, particularly in the right cingulum and anterior orbitofrontal cortex and is sustained over the longer term is supportive of function in both ATLR groups. This possibly serves as a foundation for ongoing longer-term connectivity increases within the right cingulum in the left ATLR group. For visual memory recovery, shorter term plasticity in the contralateral middle temporal cortex continues over the longer term, while ipsilateral increases in functional couplings with the right basal ganglia (external globus pallidus) is expanded to the contralateral hemisphere over the longer term in PWE with long term visual memory improvement. We have added this supplementary analysis to our results and discussed the main findings as below. Changes are highlighted in italics.

Results, Shorter-term plasticity (3-12 months) supportive of longer-term improvement in postoperative memory, Pages 20-21

"Post-hoc assessments evaluated further the neural mechanisms specific to long-term recovery of memory. We investigated the changes from before to 3-12 months after ATLR that were supportive of ongoing improvement over the longer term (3-12 months to 10 years) in PWE, compared to changes observed in healthy controls. Similar methods and thresholding to the primary longer-term plasticity analyses were employed (see Methods) and results are detailed with MNI coordinates and T-statistics in SI.

In left ATLR compared to controls, long-term improvement in verbal memory was associated with increased functional connectivity short-term compared to preoperatively, from the remnant MTL seed with the right middle cingulate and right superior parietal cortices, and reduced short-term connectivity with the remnant posterior fusiform gyrus. From the contralesional hippocampus seed, people with long-term verbal memory improvement exhibited reduced connectivity with the right inferior frontal cortex (gyrus and inferior OFC) and right angular gyrus, but increased connectivity with the right anterior and posterior OFC, right middle frontal gyrus, and bilateral parietal areas at 3-12 months compared to pre-surgery.

Long-term improvement in visual memory in left ATLR compared to controls correlated with reduced short-term connectivity from the remnant MTL seed with the right posterior fusiform gyrus, compared to preoperatively. From the contralesional hippocampus, 3-12-month connectivity that was reduced with the right superior temporal gyrus and increased with the right middle temporal gyrus correlated with improvement in longer term memory.

In right ATLR compared to controls, people with improvement in longer term memory showed increased short-term connectivity between the contralesional hippocampus and anterior MTL (left amygdala), right middle temporal and right middle cingulate cortices, and reduced with the right posterior MTL (remnant posterior fusiform) and right angular gyrus, compared to pre-surgery. From the remnant

MTL seed, people with long-term visual memory recovery exhibited enhanced connectivity with the right anterior OFC at 3-12 months compared to before surgery.

Long-term improvement in visual memory in right ATLR correlated with increased connectivity seeding from the contralesional hippocampus with the right cuneus, and seeding from the remnant MTL with the right the calcarine and basal ganglia (external globus pallidus), and decreased remnant MTL connectivity with the left inferior OFC.”

Discussion, Efficient medial temporal plasticity over a decade: change of trajectory, Pages 23-24

“Early after surgery, activation-based fMRI studies denote plasticity effects in the contralesional hippocampus and related structures like the amygdala and anterior parahippocampal gyrus, supportive of memory function beyond 5 years post-ATLR.⁹⁻¹¹ In our study, impaired preoperative connectivity was followed by new functional connections between the contralesional hippocampus and amygdala 3–12 months after right ATLR, which efficiently supported long-term improvement in verbal memory.

Conversely, increased memory fMRI activation in the surgically spared posterior hippocampus and associated regions is reportedly not functionally efficient four months post-ATLR²⁰. This aligns with the observed ‘adaptive’ decrease in remnant fusiform connectivity at 3–12 months compared to preoperatively, irrespective of surgical laterality and memory type. The surgically induced, partial severance of the inferior longitudinal fasciculus (ILF) may result in this observed functional disconnection, as the ILF connects anterior and posterior MTL regions and its resection can affect verbal and visual cognition^{29,30}.

It is of note that PWE with continued memory improvement beyond 3–12 months developed new functional connections with posterior and surgically spared regions (i.e., fusiform and parahippocampal gyri), that were efficiently integrated in the cognitive network at 10 years. Conversely, people with HS exhibited reduced connectivity to the remnant fusiform gyrus, a region herein demonstrated to be critical for memory recovery. Preoperatively, people with HS pathology show greater functional network disruption than those without^{16,31}. Memory function at 10 years was significantly worse in people with HS, suggesting that this represented a maladaptive reduction in long-term connectivity.

Overall, we propose that early post-surgery, there is efficient new contralateral connectivity with anterior medial temporal regions, laying the foundation for longer-term plasticity, as posterior remnant regions become involved in recovery of mnemonic processes.”

Discussion, Resection extent and successful network reorganizations, Pages 24-25

In line with our long-term volumetric findings, Bonelli and colleagues^{20,32} found that short-term post-ATLR, memory function does not correlate with the volumes of preoperative and remnant postoperative hippocampus. This lack of correlation may be attributed to the observed consistency in resection extent irrespective of surgical laterality, resulting in a clinical cohort with homogeneous remnant MTL volumes.

Moving beyond volumetric analyses, connectivity measures highlighted crucial differences in plasticity patterns. These included successful network reorganization toward structures that putatively underly the processing and specialized pattern recognition of graphic and contextual stimuli.^{33,34} Memory formation may rely on the conceptual representation of features supported by specialized visual medial temporal regions. Recent evidence denotes positive engagement of the fusiform’s visual word form and face areas during recognition of both letters and faces, with functional lateralization independent of material type³⁴. This raises questions about external drivers of hemispheric dominance, beyond category-selective representations³⁶ – for instance, the surgical removal of homologous medial temporal regions.

Our findings emphasize the value of using connectivity metrics to assess individual mechanistic variations in cognitive recovery and the importance of preserving posterior MTL areas during surgery. Enhancing their integration with the broader brain network through preoperative pre-habilitation and postoperative rehabilitation presents a promising clinical avenue to facilitate long-term cognitive recovery.”

Supplementary Results, Post-hoc assessment: Shorter-term plasticity (3-12 months) supportive of longer-term improvement in postoperative memory, Pages 4-5:

“To further elucidate the neural mechanisms underlying long-term memory recovery, we investigated the changes in connectivity from before to shortly after ATR that remain supportive of memory recovery over the long term (beyond 3-12 months in PWE compared to changes observed in healthy controls. Similar methods to the primary long-term plasticity analysis were employed, using separate mixed ANOVAs within a flexible factorial design for each MTL seed and memory task, with both IQ levels and change in ASM intake as confounding variables. Individual memory change (list and design learning) from 3-12 months to 10 years was treated as the regressor of interest.

Left ATR relative to controls

Words remembered. Short-term after surgery compared to preoperatively, functional connectivity from the remnant MTL seed that was reduced with the remnant posterior fusiform gyrus (MNI= -38 -46 -18, T=3.51) but was increased with the right middle cingulate cortex (MNI = 18 -32 48, T=3.40) and right superior parietal gyrus (MNI=22 -48 70, T=3.56) correlated with people who showed improvement in verbal memory over the long term. From the contralesional hippocampal seed, long-term verbal memory improvement correlated with increased 3-12-month connectivity with the right anterior and posterior OFC (anterior: MNI=22 38 -16, T=3.50, posterior: MNI=34 34 -14, T=4.28), right middle frontal (MNI=34 16 42, T=3.98) and bilateral parietal areas (left supramarginal: MNI=-62 -42 24, T=4.28; right superior parietal: MNI=30 -66 54, T=3.85) compared to before surgery. There were also adaptive decreases in right hippocampal connectivity with the right inferior frontal gyrus (MNI=40 16 14, T=3.68), right inferior OFC (MNI=36 30 -6, T=3.53) and right angular gyrus (MNI=44 -48 36, T=3.57).

Faces remembered. Short-term after surgery compared to preoperatively, people who had reduced functional connectivity from the remnant MTL seed with the right posterior fusiform gyrus (MNI=26 -66 -4, T=3.66) showed improvement in longer term visual memory. From the contralesional hippocampus seed, reduced 3-12-month connectivity with the right superior temporal gyrus (MNI=44 -36 14, T=3.57) but increased with the right middle temporal gyrus (MNI=58 -62 8, T=3.69) correlated with people who improved over the long term.

Right ATR relative to controls

Words remembered. Short-term post-surgery compared to preoperatively, people who had connectivity from the contralesional left hippocampus that was reduced with the remnant posterior fusiform gyrus (MNI=30 -82 -6, T=3.48), left middle frontal gyrus(-26 24 36), and right parietal cortex (angular gyrus: MNI=56 -60 36, T=3.58) but increased with the left amygdala (MNI=-18 -4 16, T=4.95), right middle temporal gyrus (MNI=42 -42 -2, T=3.51) and right middle cingulate cortex (MNI=18 -40 30, T=3.40) showed improvement in longer term memory. From the remnant right MTL seed, people with more functional connectivity with the right anterior OFC (MNI=38 36 -16, T=4.53) at 3-12-month compared to preoperatively showed improvement in long-term verbal memory.

Faces remembered. From the contralesional hippocampus seed, people who had increased functional connectivity with the right cuneus (MNI=20 -70 20, T=3.72) at 3-12months compared to before surgery also exhibited long-term visual memory improvement. From the remnant MTL seed, reduced connectivity with the left inferior OFC (MNI=-44 32 -12, T=3.51) and (MNI=22 -70 16, T=4.11) and right basal ganglia (external globus pallidus: MNI= 14 2 -8, T=3.82) correlated with people with improvement in long-term visual memory.”

6. In Fig. 1, the difference between the T2 and T3 is only in the verbal memory following the left section. The other differences are between the T1 and T2 in visual memory.

6.1. Hence, why the MTL-connectivity changes despite no statistically significant difference between most T2 and T3 time period.

We thank you for your comment. Although at the group level there were no significant changes between T2 and T3, on the individual level, we report changes as below:

For visual memory from T2 to T3, we saw that clinically significant recovery - as established by the reliable change index with 95% confidence interval - was seen in 42% of left ATR and 38% of right ATR despite not reaching significance at the group level. Therefore, we assessed the connectivity changes that more strongly related with those who had visual memory improvement than those who showed either visual memory decline or stability, allowing to gain insight on the neurobiological mechanisms of successful visual memory recovery. The main aim of this study was to study brain plasticity underpinning cognitive recovery long-term after epilepsy surgery. This is important down to

the level of an individual neuropsychological function, with regards to its applications to both tailored rehabilitation and presurgical counselling.

6.2. Relatedly, the memory outcome is diverse, roughly each one third improved, stationary or deteriorated. Hence in this scenario, the individual analysis may be better to the group analysis.

We thank you for your suggestion.

For Neuroimaging data, our group-level analysis entailed the correlation of individual change in memory score from short to long-term follow-ups with individual change in task-modulated fMRI connectivity over the same timeline. This group-level correlation of fMRI biomarkers with individual change in memory performance allows to assess which changes in fMRI correlates have a stronger relationship with people whose memory successfully improved over the long term in comparison with those who did not. As such, our model combines the flexibility of an individual analysis (i.e., the repeated measures allow to assess individual changes in both fMRI and memory function) while providing the statistical power to differentiate between those with better and worse memory and therefore to highlight the changes in connectivity that were more dominant in those with successful memory recovery. This fell under the scope of the project, which had a specific interest in the neural mechanisms underlying successful, as opposed to failed/inefficient, memory recovery over the long term. We have highlighted this in our methods.

For the Neuropsychology data, we have re-analysed the longitudinal changes in memory across groups using linear mixed effect models to better depict individual variations, in response to your review. Similar to a repeated measures ANOVA, linear mixed effect models test for interaction effects of Group*Time factors on memory function in a mixed design (between-subject Group and within-subject Time factors), however with greater flexibility for modelling individual variances in time effects. This approach assesses fixed effects (Time, Group, Time*Group interaction) while accounting for random subject-specific effects²⁰. We have found similar results to previous findings. Changes to the manuscript are highlighted in italics.

Results, Group differences in memory outcome, Page 11

*“Longitudinal Group*Time interaction effects on memory function were assessed using linear mixed-effect models, and the impact of preoperative HS pathology, ASM cessation and seizure-freedom at 10 years using Fisher-Pitman permutation tests. Group averages in memory z-scores and statistics are shown Table 2. Supplementary information (SI) details preoperative neuropsychology, confidence intervals, and effects of clinical factors.*

Group differences: at both 3–12-month and 10-year assessments, healthy individuals’ verbal memory was significantly better than that of people in left ATLR ($\beta = 2.7, P = 0.0002$) and right ATLR ($\beta = 2.0, P = 0.003$) groups. For visual memory, controls performed significantly better than individuals in the right ATLR ($\beta = 1.5, P = 0.005$) but not left ATLR ($\beta = 0.63, P = 0.27$) group. Across postoperative timepoints, there was no significant difference in verbal or visual memory z-scores between ATLR groups (see SI).

*Interaction effect: For verbal memory, the effect of time on memory z-scores differed between controls and left ATLR, with a greater memory change (increase) over time in left ATLR than controls ($\beta = 1.1, P = 0.016$). There was no significant Group*Time interaction effect between controls and the right ATLR group for verbal memory ($\beta = 0.74, P = 0.071$), and between controls and either ATLR group for visual memory (left ATLR: $\beta = 0.020, P = 0.96$; right ATLR: $\beta = 0.38; P = 0.60$).”*

Methods, Statistical analyses, Clinical and neuropsychological data, Page 36

*“Linear mixed-effect models were conducted to assess Group*Time interaction effects on memory z-scores, corrected for multiple comparisons using False Discovery Rate. For each list and design-learning task, linear models included the fixed effects of Group (left ATLR, right ATLR, controls), Time (3-12-month and 10-year memory z-scores), and Group*Time interaction, along with subject-specific random intercepts to account for both individual variability in baseline memory and non-independence of repeated measures.”*

Methods, Statistical analyses, Correlation between functional reorganization in TLE and memory recovery, Page 38

“Three-way ANCOVAs, *controlling for IQ and changes in ASM intake*, were conducted for each MTL seed and successful memory contrast (words/faces) to investigate differences in functional connectivity from 3–12-month and 10-year follow-ups related with improvement in memory functions over this timeline.

Specifically, individual changes in memory z-scores were correlated with changes in task-modulated fMRI connectivity over time, modelling the fMRI biomarkers more strongly associated with those whose memory successfully improved compared to those who did not. Differences in z-scores (i.e., BMIPB I or II verbal and visual learning scores converted into age-normalized z-scores) from 3–12 months to 10 years were used as continuous variables.”

7. Also I have a concern about their cohort as well.

7.1. Whether they have hippocampal sclerosis preoperatively? If some of them have HS, the preoperative MTL connectivity might be different from that in the patients without HS.

We thank you for your comment. To address the possible effect of preoperative pathology on long-term plasticity, we performed post-hoc analyses separately in each ATR group and for each MTL seed and subsequent memory fMRI task. Two samples t-tests were conducted separately in each ATR group and for each MTL seed and subsequent memory fMRI task comparing the longitudinal connectivity changes in people with HS pathology with those in people without HS. We found that people with HS show reduced plasticity (i.e., less longitudinal increases in connectivity), with posterior MTL regions that are supportive of memory functions. After both left and right ATR, this included reduced long-term plasticity with the remnant posterior fusiform gyrus seeding from the remnant MTL seed for successful word encoding, and seeding from the contralesional hippocampal seed for successful face encoding. We have included this supplementary analysis in our manuscript and discussed it as below. We have also assessed the impact of HS pathology on memory function long-term post-ATR. Fisher-Pitman permutation tests were performed to account for small numbers and consequently uneven sample distribution between people with and without HS. We found that people with HS preoperatively showed poorer verbal and visual memory functions 10 years post-surgery.

Changes to the manuscript are highlighted in italics.

Methods, Statistical analyses, Clinical and neuropsychological data, Page 36

“Due to small numbers, Fisher-Pitman permutation tests were performed to assess the impact of HS pathology preoperatively, and ASM cessation and seizure-freedom at 10 years on long-term memory function.”

Results, Group difference in memory outcome, Page 11

“One-way ANOVAs were used on parametric z-scores and post-hoc tests were corrected using Tukey's Honestly Significant Difference (HSD) adjustment. *Due to small numbers, Fisher-Pitman permutation tests were performed to assess the impact of preoperative HS pathology, and ASM cessation and seizure-freedom at 10 years. [...] Supplementary information (SI) details preoperative neuropsychology, confidence intervals, and effect of clinical factors.*

Impact of clinical factors: *Ongoing medication at 10 years and preoperative HS pathology adversely impacted long-term memory functions. This included worse visual memory recovery in people who continued ASMs (visual: $Z = 2.08$; $P = 0.036$), and worse 10-year verbal and visual memory in those with compared to those without HS (verbal: $Z = -2.70$, $P = 0.0045$; visual: $Z = -2.11$, $P = 0.029$.)”*

Results, Impact of hippocampal sclerosis on long-term plasticity effects, Pages 19-20:

“Post-hoc two-sample t-tests evaluated the potential effect of HS on long-term memory plasticity. The longitudinal connectivity changes in people with HS were compared with those in people without HS, separately in each ATR group for each MTL seed and subsequent memory fMRI task. Due to the maladaptive impact of HS on preoperative network plasticity¹⁶, we hypothesized that people with HS will have less network changes to regions supportive of memory functions from 3-12 months to 10 years. Results are reported within a binary thresholded mask of the longitudinal PPI's main effect, using the same thresholding methods as in above analyses. Refer to SI for detailed methods and results with MNI coordinates and T-statistics.

For successful word encoding after left ATLR, individuals with HS compared to those without HS exhibited reduced plasticity (i.e., less longitudinal increases in connectivity) from 3-12 months to 10 years between the remnant MTL seed and the bilateral posterior fusiform gyrus. There was no difference in plasticity patterns from the contralesional hippocampus seed.

For successful face encoding after left ATLR, people with HS had reduced plasticity between the contralesional hippocampus seed and the remnant posterior fusiform gyrus. There was no difference in plasticity effects from the remnant seed.

For successful word encoding after right ATLR, individuals with HS showed less plasticity between the remnant MTL seed and the remnant fusiform gyrus than those without HS.

For successful face encoding after right ATLR, people with HS had more plasticity (i.e., greater increases in connectivity) from 3-12 months to 10 years between the contralesional hippocampus seed and the left posterior fusiform gyrus, but reduced plasticity with the remnant posterior fusiform gyrus, compared to those without HS. From the remnant MTL seed, people with HS had less plasticity with the remnant posterior parahippocampus than those without HS.”

Discussion, Efficient medial temporal plasticity over a decade: change of trajectory, Pages 24

“It is of note that PWE with continued memory improvement beyond 3–12 months developed new functional connections with posterior and surgically spared regions (i.e., fusiform and parahippocampal gyri), that were efficiently integrated in the cognitive network at 10 years. Conversely, people with HS exhibited reduced connectivity to the remnant fusiform gyrus, a region herein demonstrated to be critical for memory recovery. Preoperatively, people with HS pathology show greater functional network disruption than those without^{16,31}. Memory function at 10 years was significantly worse in people with HS, suggesting that this represented a maladaptive reduction in long-term connectivity.”

Supplementary Results, Group difference in memory outcome, Page 2

“Impact of clinical factors: [...]Individuals with HS preoperatively showed worse 10-year verbal and visual memory functions than those without HS pathology (verbal: $Z = -2.70$, $P = 0.0045$; visual: $Z = -2.11$, $P = 0.029$). Post-hoc analyses showed significant HS effect after left ATLR on verbal memory ($Z = -1.86$, $P = 0.045$) but not visual memory ($Z = -1.53$, $P = 0.14$) nor right ATLR (verbal: $Z = -1.86$, $P = 0.061$; visual: $Z = -1.87$, $P = 0.061$).”

Supplementary Results, Post-hoc assessment: Impact of hippocampal sclerosis on long-term plasticity effects, Page 3:

“Left ATLR: two samples t-test

Word remembered. *People with HS compared to those without HS had reduced plasticity (i.e., less connectivity increases) from 3-12 months to 10 years between the remnant MTL seed and the bilateral posterior fusiform gyrus (remnant: $MNI = -32 -58 -6$, $T=7.58$; right: $MNI=26 -76 -2$, $T=4.54$). There was no difference in plasticity patterns from the contralesional hippocampus seed, nor any enhanced plasticity (more connectivity increases) between those with HS and those without HS.*

Faces remembered. *There was no difference from the remnant MTL seed between plasticity patterns of those with and without HS. People with HS had reduced plasticity between the contralesional hippocampus seed and the remnant posterior fusiform gyrus ($MNI:-22 -48 -16$, $T=3.74$).*

Right ATLR: two samples t-test

Word remembered. *No difference from the contralesional hippocampus seed. People with HS compared to those without HS had less plasticity (i.e., less longitudinal increases in connectivity) between the remnant MTL seed and the remnant fusiform gyrus ($MNI=34 -38 -14$, $T=2.96$).*

Faces remembered. *People with HS compared to those without HS had more plasticity (i.e., greater increases in connectivity) from 3-12 months to 10 years between the contralesional hippocampus seed and the left posterior fusiform gyrus ($MNI: -38 -46 -12$, $T=4.20$), but reduced connectivity with the remnant posterior fusiform gyrus ($MNI: 34 -46 -12$, $T=3.53$). People with HS compared to those without HS had reduced plasticity (less connectivity increases) between the remnant MTL seed and the remnant posterior parahippocampus ($MNI=36 -42 -8$, $T=3.52$).”*

7.2. Is there any difference in the extent of surgical resection of anterior temporal lobe and hippocampus?

Thank you for your question. All surgeries were performed by the same surgeon, all participants had a standard ATR with little variation in extent of resection. Post-surgical scans were corrected to account for individual resection and brain sag.

In response to your review, we have conducted supplementary tests to assess for differences in the extent of anterior temporal lobe and hippocampus resection between groups. Aligning with Bonelli et al., 2013 findings, there were no statistical differences between groups in the resection extents. For completeness, we also assessed within in each ATR group, whether the 10-year volume of the remnant, surgically spared hippocampus affected the recovery of verbal and visual memory. We found that it did not significantly relate to recovery from 3-12 months to 10 years after both ATR. This is probably associated with the little variations in volumes of remnant hippocampus among participants, as shown above, and highlights the value of connectivity analyses over volumetric analyses to assess individual variations in mechanistic aspects of cognitive recovery. Changes to the manuscript are highlighted below in italics.

Methods, Resection extent, Page 39

“To test for potential effects of variations in the extent of temporal lobe resection, postoperative T1 scans were co-registered to the preoperative T1 native space using Elastix⁷¹ with affine and bspline transformations. Individual resection masks were automatically generated using Resseg⁷²; a self-supervised, convolutional neural network approach to segment the temporal lobe cavity. The extent of anterior temporal lobe structures’ resection was assessed, accounting for individual variations in brain sizes, by dividing the resection cavity volume by the total brain’s volume for each subject. The proportion of hippocampus volume resected was calculated using the formula: $(V1 - V2) / V1$, with V1 and V2 standing for the hippocampus volume at the preoperative baseline and at the 10-year follow-up, respectively.

Mann-Whitney tests were conducted to assess whether the size of resection of both the anterior temporal lobe structures and the epileptogenic hippocampus differed according to surgical laterality. To investigate the effect of differences in the volume of surgically spared hippocampus on long-term recovery of verbal and visual memory, linear regressions were performed within each resection group. The ipsilateral remnant hippocampus volume at 10 years was the explanatory variable and the individual change in list- and design-learning z-score from 3-12 months to 10 years the outcome variable.”

Results, Differences in resection extent, Page 12

“There was no significant difference in the extent of hippocampus resection (Mann-Whitney tests, $W = 91$, $P = 0.13$), and of anterior temporal lobe structures’ resection ($W = 83$, $P = 0.32$) between left and right ATR groups. Figure 2 illustrates the preoperative and 10-year volumes of the surgically spared, remnant hippocampus in each resection group.

Correspondingly, linear regressions showed no predictive effect of the remnant hippocampus volume at 10 years on verbal and visual memory recovery from 3-12 months to 10 years, irrespective of memory type and surgical laterality; verbal memory, left ATR $P = 0.48$, $\beta = -0.0003$, right ATR $P = 0.53$, $\beta = 0.0002$; visual memory, left ATR $P = 0.08$, $\beta = 0.0006$, right ATR $P = 0.42$, $\beta = -0.0003$. This lack of predictive effect is possibly associated with the non-significant variations in the extents of hippocampus resection.”

Figure 2 Volume of the epileptogenic hippocampus before and 10 years after anterior temporal lobe resection (ATLR). In each group, the boxplots and its whiskers show mean, standard error, and maximum and minimum volumes of the ipsilateral hippocampus, preoperatively and at the 10-year follow-up in people with epilepsy. Violin distribution shows the data spread.”

Discussion, Resection extent and successful network reorganizations, Pages 24-25

“In line with our long-term volumetric findings, Bonelli and colleagues^{20,32} found that short-term post-ATLR, memory function does not correlate with the volumes of preoperative and remnant postoperative hippocampus. This lack of correlation may be attributed to the observed consistency in resection extent irrespective of surgical laterality, resulting in a clinical cohort with homogeneous remnant MTL volumes.

Moving beyond volumetric analyses, connectivity measures highlighted crucial differences in plasticity patterns. These included successful network reorganization toward structures that putatively underly the processing and specialized pattern recognition of graphic and contextual stimuli.^{33,34} Memory formation may rely on the conceptual representation of features supported by specialized visual medial temporal regions. Recent evidence denotes positive engagement of the fusiform’s visual word form and face areas during recognition of both letters and faces, with functional lateralization independent of material type³⁵. This raises questions about external drivers of hemispheric dominance, beyond category-selective representations³⁶ – for instance, the surgical removal of homologous medial temporal regions.

Our findings emphasize the value of using connectivity metrics to assess individual mechanistic variations in cognitive recovery and the importance of preserving posterior MTL areas during surgery. Enhancing their integration with the broader brain network through preoperative pre-habilitation and postoperative rehabilitation presents a promising clinical avenue to facilitate long-term cognitive recovery.”

7.3. Also what is the pathologies of the patients? Cortical dysplasia, tumors, or some others?

We thank you for your enquiry. All subjects who underwent left ATLR were MRI positive; 10/12 had hippocampal sclerosis (HS) preoperatively, 1 cavernoma and 1 ependymoma. Among right ATLR subjects, 11/13 were MRI positive, 8/13 had HS, 1 cavernoma, 1 dysembryoplastic neuroepithelial tumor, and 3 gliosis. This has been added to results and changes are highlighted in italics below.

Results, Subjects, Page 7

“Patient demographics and clinical features at 3–12 months and 10 years are outlined Table 1. *All subjects who underwent left ATLR were MRI positive; 10/12 had hippocampal sclerosis (HS) preoperatively, 1 cavernoma, and 1 ependymoma. Among right ATLR subjects, 11/13 were MRI positive, 8/13 had HS, 1 cavernoma, 1 dysembryoplastic neuroepithelial tumor, and 3 gliosis (only*

evident on pathology in 2). Long-term postoperatively, seizure-freedom (i.e., ILAE outcome 1) was observed in 83% of left and 92% of right ATR.

7.4. Also the period between the onset of epilepsy and the surgery time is controlled in their cohort. If the duration of medically intractable epilepsy differs in their cohort, the preoperative reorganization would be different as well.

We thank you for your review. We agree that preoperative reorganization is associated with age of onset, duration, seizure burden (Fleury 2022 *Epilepsia*²¹, and Sidhu 2015 *Neurology*²²). However, modelling changes from 12 months to 10 years after surgery compared to changes in controls mitigates the plasticity effects associated with these individual parameters known to influence preoperative reorganization. In this analysis we have looked at differences from short-term postop to long-term postoperatively, regardless of what has happened in preoperative reorganization. Our analysis uses a flexible factorial design, capturing the individual changes in connectivity at two intervals following surgery, regardless of preoperative plasticity. What is common to all, is the interval of plasticity changes after resection.

However, factors that can affect the postoperative memory network include changes in ASM and also the effect of pathology, as pointed by the reviewer (point 7.1). We have therefore looked into these factors separately.

For ASM we have re-analysed our results using the number ASMs as regressor of no interest. The changes in memory networks from 3-12 months to 10 years specifically associated with long-term memory improvement were also correlated with change in memory scores while treating individual ASM intake at both 3-12 months and 10 years as continuous regressors of no interest.

Fisher-Pitman permutation tests were performed to assess the impact of preoperative HS pathology, and ASM cessation and seizure-freedom at 10 years on long-term memory functions. Permutation tests allowed to account for small numbers and uneven sample distribution between people with and without HS, and those who ceased and continued ASM.

Please refer to response 7.1 (pages 18-19) for the manuscript changes related to the impact of HS.

Methods, Statistical analyses, Clinical and neuropsychological data, Page 36

“Due to small numbers, Fisher-Pitman permutation tests were performed to assess the impact of HS pathology preoperatively, and ASM cessation and seizure-freedom at 10 years on long-term memory function.”

Methods, Statistical analyses, Longitudinal assessment of the functional memory network, Page 36:

“Postoperative clinical factors like change in medication or ASM cessation can impact memory function and the corresponding mechanisms underlying cognitive network recovery. As such, individual ASM intake at both 3-12 months and 10 years was treated as continuous regressor of no interest in all neuroimaging analyses.”

Methods, Statistical analyses, Long-term changes in functional connectivity from 3-12 months to 10-year follow-ups, Pages 37

“Mixed ANOVAs within a flexible factorial design with both IQ and change in ASM intake as confound regressors, were performed to investigate changes in MTL-seeded memory connectivity between short-term and long-term follow-ups in the individual patient groups compared to changes in test-retest over that timeline in healthy subjects.”

Methods, Statistical analyses, Correlation between functional reorganization in TLE and memory recovery, Page 38

“Three-way ANCOVAs, controlling for IQ and changes in ASM intake, were conducted for each MTL seed and successful memory contrast (words/faces) to investigate differences in functional connectivity from 3–12-month and 10-year follow-ups related with improvement in memory functions over this timeline.”

Results, Group differences in memory outcome, Page 11

“Impact of clinical factors: Continued medication at 10 years and HS pathology adversely impacted long-term memory functions. This included worse visual memory recovery in people who continued ASMs (visual: $Z = 2.08$; $P = 0.036$), and worse 10-year verbal and visual memory in those with compared to those without HS (verbal memory: $Z = -2.70$, $P = 0.0045$; visual memory: $Z = -2.11$, $P = 0.029$).”

Results, Changes in functional connectivity 3-12 months to 10 years post-surgery relative to changes in controls, Page 13

“Table 3 shows detailed results from the flexible factorial design analysis.

For successful word encoding in left ATR compared to controls, there was increased connectivity from the remnant MTL seed with both bilateral posterior parahippocampal and fusiform gyri from 3–12 months to 10 years postoperatively. Extra-temporally, connectivity was longitudinally increased between the remnant MTL and both the right anterior cingulate cortex and bilateral occipital cortices. For successful face encoding, there was enhanced functional connectivity between the remnant left MTL seed and posterior MTL regions, including the remnant parahippocampal and bilateral fusiform gyri, long-term compared to short-term after left ATR. Neocortically, there was increased connectivity between the contralesional hippocampal seed and the right superior frontal gyrus from 3–12 months to 10 years.

For successful word encoding in right ATR compared to controls, there was increased long-term functional connectivity compared to short-term, from the remnant right MTL seed with the remnant posterior parahippocampal and fusiform gyri, and extra-temporally with the left inferior parietal gyrus. For successful face encoding, functional connectivity from 3–12 months to 10 years after right ATR was enhanced mainly from the contralesional hippocampal seed; posteriorly with the remnant parahippocampal and bilateral fusiform gyri, and extra-temporally with the right inferior frontal and right supramarginal gyri. The remnant right MTL seed showed stronger functional couplings with the remnant posterior parahippocampus.”

Results, Correlation between longitudinal changes in connectivity and memory recovery 3-12 months to 10 years postoperatively, Pages 14-15

“From 3–12 months to 10 years after left ATR compared to controls, improvement in verbal memory was significantly correlated with increased functional connectivity between the remnant left MTL seed and both the right posterior parahippocampus and the remnant posterior fusiform gyrus. At the extra-MTL level, verbal memory improvement was significantly associated with enhanced functional couplings between the remnant MTL seed and the right anterior cingulate cortex, along with reduced connectivity with the left parietal gyrus, 10 years relative to 3–12 months post-surgery.

Visual memory improvement after left ATR compared to controls correlated with longitudinal increases in connectivity between the remnant left MTL seed and bilateral posterior fusiform gyri, and between the contralesional hippocampal seed and both remnant hippocampus and fusiform gyrus, from short-term to long-term follow-ups. Neocortically, people with long-term improvement in visual memory exhibited a longitudinal increase in contralesional frontal and bilateral temporal connectivity; between the remnant MTL and bilateral temporal areas (right middle and left inferior), and between the contralesional hippocampal seed and the right anterior OFC and right superior frontal gyrus.

Similar to left ATR, from 3–12 months to 10 years after right ATR, verbal memory improvement was significantly correlated with increased connectivity with posterior MTL regions; from the remnant right MTL seed with the remnant posterior fusiform, and from the contralesional left hippocampal seed with the remnant parahippocampal gyrus. At the extra-MTL level, long-term verbal memory improvement was significantly supported by reduced connectivity between the remnant MTL and left amygdala, right thalamus, and right supramarginal gyrus.

Visual memory improvement after right ATR compared to controls correlated with longitudinal increases in functional connectivity with the bilateral posterior MTL. This involved enhanced connectivity from the remnant right MTL seed with the remnant parahippocampal and contralesional

fusiform gyri, and from the contralesional hippocampal seed with the contralesional parahippocampal and bilateral fusiform gyri. At the extra-MTL level, improvement in visual memory significantly correlated with increased functional connections between the contralesional hippocampal seed and the bilateral middle and inferior temporal gyri, contralesional basal ganglia (external globus pallidus), right inferior frontal and right supramarginal gyri, 10 years compared to 3–12 months after right ATR.

Discussion, Strengths and limitations, Page 28

“Next, in our analyses we took into account number of ASMs at each timepoint and included these as regressors of no interest. This modelled for any changes attributed to the change in number of ASMs at each timepoint. Due to the variability of ASMs prescribed clinically however, we were unable to assess the specific impact of individual ASMs on memory network changes.”

Supplementary Results, Group difference in memory outcome, Page 2

“Impact of clinical factors:

From 3-12 months to 10 years post-ATR, people who ceased medication recovered significantly better in visual memory function than those who continued ASMs, but not in verbal memory (visual: $Z = 2.08$; $P = 0.036$; verbal: $Z = -0.87$; $P = 0.40$). Post-hoc analyses showed significant ASM effect on visual memory after right but not left ATR (right: $Z = 2.37$, $P = 0.012$; left: $Z = 0.28$, $P = 0.79$). Ongoing seizures 10 years post-surgery did not affect memory recovery (visual: $Z = 1.44$; $P = 0.16$; verbal: $Z = 0.39$; $P = 0.71$).

Individuals with HS preoperatively showed worse 10-year verbal and visual memory functions than those without HS pathology (verbal: $Z = -2.70$, $P = 0.0045$; visual: $Z = -2.11$, $P = 0.029$). Post-hoc analyses showed significant HS effect after left ATR on verbal memory ($Z = -1.86$, $P = 0.045$) but not visual memory ($Z = -1.53$, $P = 0.14$) nor right ATR (verbal: $Z = -1.86$, $P = 0.061$; visual: $Z = -1.87$, $P = 0.061$).

- 7.5. They assessed the results from the 25 patients following anterior temporal resection. However, this is not the whole story, since their cohort might include others who may not be followed up or not be examined with fMRI. It should be disclosed.

We thank you for your review. This is a longitudinal analysis, wherein, preoperatively we had 90 patients with TLE (43 left) and 29 controls²¹. 65 people had ATR, including 30 left-sided. All participants were invited for long-term study, and only 25 PWE and 10 controls were keen to participate and return for long-term study. Of the 40 that did not return (18 left ATR), clinical data was available for 35, among which 74% (26/35) were seizure free with ILAE outcome 1 at the 12-month clinical follow-up (including 75% of left ATR and 74% of right ATR). They however did not consent to being included to the long-term follow-up scanning; we are therefore unable to report any further outcomes for those who did not wish to participate.

- 7.6. There is no handedness information in both patients group and the control.

Thank you for highlighting this. We have added this information in our Results.

Results, Subjects, Page 7

“Nine of 10 healthy controls were right-handed, and one left-handed, as were 10 of 12 in the left ATR group, two left-handed. In the right ATR group, 11 of 13 were right-handed (1 ambidextrous and 2 left-handed). Patient demographics and clinical features at 3–12 months and 10 years are outlined in Table 1.”

Reviewer 4

The present study showed that patients who undergo anterior temporal lobe resection for epilepsy can experience long-term improvements in memory function up to a decade post-surgery. The brain demonstrates plasticity, allowing memory networks to reorganize and recover. The findings support the use of less invasive surgeries and ongoing cognitive rehabilitation to enhance long-term memory recovery in epilepsy patients. These findings significantly enhance our understanding of the brain's

capacity for adaptation and functional reorganization, especially regarding memory, after epilepsy surgery. It reinforces the notion that with careful surgical planning and dedicated postoperative care, including cognitive rehabilitation, patients can look forward to a future where improvement in cognitive abilities is not only possible but sustainable over the long term.

1. Despite providing several important insights, this study carries weaknesses in that

1.1. the study's small sample size limits the statistical reliability and necessitates cautious interpretation regarding the generalizability of the findings.

Thank you for your comment. The memory task has been refined to allow for reliable individual mapping of preoperative memory function that can be used presurgically. Indeed, individual frontal temporal activations within a mask are predictive of verbal memory outcome with 87.5% sensitivity and 80% specificity⁴. The finding of significant activations both at the individual and group-level suggests the robustness of the fMRI task. Other study with small sample size showing significant results include a previous longitudinal study; using similar PPI analysis (ANOVA and paired t-test) of verbal fluency task in 14 people who had add-on perampanel⁵.

This is the first study that reports changes in gPPI connectivity for memory; other groups have looked at connectivity of resting-state fMRI in group of 9 people with gliomas⁶ to map language function and another compared connectivity concordance of verbal generation fMRI with that of resting-state fMRI in 3 people with gliomas⁷, and found comparable patterns of activation. Task-based connectivity compared to resting-state maximizes the reliability of functional connectivity specifically for task-engaged regions in the functional network of interest⁸. Smaller numbers than our study number have been used in activation-based univariate analyses (9 left TLE, 8 right TLE)⁹ to look at changes in scene-encoding fMRI patterns long-term post-epilepsy surgery. Although similar, functional connectivity analysis is a more sensitive method than univariate analysis for describing network-level functional architecture^{10,11}.

We however appreciate the limitation of our small sample size on generalisability, and it is discussed in our limitations. Changes made in the manuscript are highlighted in italics below.

Discussion, Strengths and limitations, Page 28

“The postsurgical cohort was primarily seizure-free, limiting the generalisability of our results to people with good prognosis. Future studies with larger numbers should investigate the modulatory effect of specific ASMs and differential seizure outcomes on network recovery.”

1.2. In addition, the use of different scanner types during follow-up assessments could undermine measurement consistency, potentially limiting the comparability of the study results. Although this represents a missed opportunity, the findings from the available data are sufficiently striking and coherent to merit reporting.

We thank you for your comment and understanding of the challenges inherent to the experimental design. There is likely to be scanner effects using different scanners. Response 3 to reviewer 1 (Page 6) describes how we have modelled for this, and the related changes made to the manuscript.

2. The study acknowledges that surgical procedures can have varied effects on cognitive functions. Longitudinal studies with pre-and post-operative assessments, including a range of cognitive domains, could delineate the specific cognitive functions affected by surgery. Including a comparison group of patients with similar conditions but not undergoing surgery would help isolate the effects of the surgical intervention itself.

In our study we do not have participants who did not undergo surgery. There is clinical data, but non-surgical participants have not consented to this study. Hence, we are unable to report it. However, performing a literature review, we found that:

1) Regarding the cognitive functions in people with refractory epilepsy who do not undergo surgery: In a study of more than 135 individuals with severe intractable epilepsy, Thompson and Duncan (2005)²³ demonstrated that complex partial seizures predicted progressive cognitive decline over multiple domains (memory and executive functions) between the baseline assessment and 10-year follow-up. Uncontrolled focal seizures did not affect IQ levels. Of note, periods of remission were

associated with better cognitive outcomes at the long-term assessment, indicating the benefit of good seizure control on cognitive prognosis.

2) Regarding the cognitive functions in the long term after ATR:

In a longitudinal study with a 5-to-22-year follow-up that compared memory and executive functions in people undergoing ATR with non-operated people with focal epilepsy, Helmstaedter, Elger and Vogt (2018)²⁴ showed that in both operated and non-operated groups, seizures control and reduction in ASM load were respectively associated with recovery of memory and executive functions. This was independent of the surgery effect, indicating no disadvantage for the surgical group. Later age at time of surgery was predictive of poorer memory recovery, and following the initial hit of surgery on verbal memory function, progressive memory decline was only seen in those with uncontrolled seizures. Baxendale, Thompson and Duncan (2012)²⁵ showed in a 5-year follow-up study, that the effect of surgical intervention includes cognitive decline that relates to temporal lobe functions in people with uncontrolled seizures, in contrast to the generalised cognitive decline in refractory focal epilepsy; this aligns with our neuropsychological findings.

We have added relevant literature to the introduction and discussion. Changes are highlighted in italics.

Introduction, Page 4:

“In medication-refractory focal epilepsy, seizures are associated with progressive cognitive decline over multiple domains, including memory and executive functions.¹⁻³ Surgical resection can bring seizure control^{1,4} which is the strongest predictor of cognitive remission, regardless of the effect of surgery on function.^{1,2} Anterior temporal lobe resection (ATR) is the most commonly performed surgical treatment for refractory temporal lobe epilepsy (TLE).⁵ This involves the partial removal of temporal neocortical and mesial temporal structures including the anterior hippocampus, parahippocampal and fusiform gyri; structures critical for successful memory formation.^{6,7} Although ATR may prevent the progressive decline observed in refractory epilepsy, up to 40% of individuals experience significant episodic memory deficits postoperatively.^{5,8} Identifying regions crucial for effective memory reorganization that could be spared in epilepsy surgery may help mitigate long-term postoperative memory loss. Additionally, understanding post-operative plasticity could assist the development of strategies to promote long-term memory recovery.”

Discussion, Long-term postoperative memory changes, Pages 22-23

“Neuropsychology studies demonstrate that ATR is characterized by an immediate ‘hit’ on episodic memory functions^{1,22}, which in seizure-free people is followed by either cognitive stabilization or ongoing improvement as recovery time increases (beyond two years).^{1,2,23} In line with current literature, postsurgical PWE did not present with the learning/test-retest effect in verbal memory, suggesting an early surgical insult on memory function. Yet beyond 3–12 months, our predominantly seizure-free cohort demonstrated cognitive stability at the group-level and patient-specific recovery was evident. From 3–12-months to 10 years, 50% of left ATR and 38% of right ATR showed clinically significant improvement in verbal and visual memory, respectively. [...]. Ten years postoperatively, 75% of left ATR and 58% of right ATR had good verbal and visual memory outcomes compared to preoperatively, with improved performance or return to baseline. Enhancement of baseline performance possibly translates to the removal of nociferous cortex and corresponding release of functions and reserve capacities previously suppressed by epileptic activity.^{24,25”}

3. Given the heterogeneity observed in patient populations, which encompasses a spectrum of underlying conditions and comorbidities, these factors could notably affect cognitive outcomes. For example, the author may wish to elaborate on the potential confounding effects of medication changes on memory dysfunction. This is particularly significant in studies where neuropsychopharmacology interventions could be a key variable. Therefore, it may be necessary to closely examine the impact of medications on memory functions for each subject and understand how these changes might have shaped the study’s conclusions.

Thank you for your comment.

Ten years postoperatively, 10 PWE were off medications and 15 were on ASMs. Out of 12 left ATR, 4 were off medications (including cessation of Tegretol retard, Zonisamide, Topiramate, Levetiracetam,

Lamotrigine, and Carbamazepine) while 6 out of 13 right ATR were off (including cessation of Tegretol retard, Levetiracetam, Lamotrigine, Keppra, and Oxcarbazepine). Of those who remained on ASMs, 3 continued to have seizures.

We have assessed the relationship of ASM cessation and ongoing seizures with memory changes from 1 to 10 years. Fisher-Pitman permutation tests indicated that ASM intake (on/off) had a significant effect on recovery of visual memory with people who ceased medication recovering significantly better than those on medication at 10 years ($Z = 2.08$, $P = 0.036$), but not for verbal memory changes. There was no significant difference in recovery of verbal nor visual memory between individuals with ongoing seizures and those who were seizure-free at 10 years. This is described in Results and Supplementary Results.

We have re-analysed our results using the effect of change in ASM intake as a confounding factor. That is,

- (1) we have re-run the longitudinal connectivity analyses with ASM intake (number of ASMs at each short-term and long-term timepoint) as a continuous regressor of no interest
- (2) the changes in memory networks from 3-12 months to 10 years specifically associated with long-term memory improvement were correlated with change in memory scores while treating individual ASM intake at both 3-12 months and 10 years as continuous regressors of no interest.

For both analyses, the study's primary conclusions were corroborated. This was added to methods and our results, tables and figures were amended accordingly.

Changes to the manuscript related to the addition of ASM as regressor of no interest are described response 7.4 to reviewer 3 (pages 22-24). Other changes are highlighted below in italics.

Methods, Statistical analyses, Correlation between functional reorganization in TLE and memory recovery, Page 38

“Three-way ANCOVAs, *controlling for IQ and changes in ASM intake*, were conducted for each MTL seed and successful memory contrast (words/faces) to investigate differences in functional connectivity from 3–12-month and 10-year follow-ups related with improvement in memory functions over this timeline.

Specifically, individual changes in memory z-scores were correlated with changes in task-modulated fMRI connectivity over time, modelling the fMRI biomarkers more strongly associated with those whose memory successfully improved compared to those who did not. Differences in z-scores (i.e., BMIPB I or II verbal and visual learning scores converted into age-normalized z-scores) from 3–12 months to 10 years were used as continuous variables.”

4. Finally, the study's reliance on anatomical masks for determining seed regions for analysis could introduce limitations. While this method offers a straightforward approach based on specific brain areas, it might not fully reflect the actual neurobiological pathways, thus constraining the interpretation of the study results. More sophisticated methods for seed region selection or approaches that consider the whole-brain connectome may be required to support current findings.

Thank you for your comment.

Our seed region in the MTL was based firstly on literature in that MTL is critical for encoding processes. Since the 1900s, substantial evidence underscores the significance of structures of the MTL in memory formation. Before surgery for intractable epilepsy, memory fMRI has demonstrated that in TLE, activation maxima may shift from typical anterior hippocampus to for instance the posterior hippocampus and after surgery also in the parahippocampus¹³⁻¹⁶. Consequently, using MTL seeds that entail (1) the non-resected hippocampus to address eventual neurobiological shift in activation fMRI maximum from anterior to posterior hippocampus, and (2) the surgically spared (remnant) posterior hippocampus and parahippocampus to address potential activation fMRI maxima shift from the hippocampus to parahippocampus emerged as a suitable approach for studying the pathological memory connectome following epilepsy surgery.

Second, we visually verified that each participant had a global maximum in the MTL mask applied – most postsurgical PWE showed a maxima in the posterior, ipsilateral MTL mask. This is in line with

previous postoperative studies, that showed plasticity to involve increase activations within the remnant hippocampus (Fleury 2022, Sidhu 2016).

We also discussed this matter with experts in PPI analysis, more specifically Prof. Carolin Moessnang (Professor of Clinical Psychology and Psychotherapy at SRH University Heidelberg) and Dr. Peter Zeidman (Chair of the Methods Group at the UCL Wellcome Centre for Human Neuroimaging and leads the Neurovascular Modelling Group). Based on their guidance, we have summarized the key points of our discussion below.

While seed selection can be literature-based or data-driven, the latter approach may introduce bias¹². Using an atlas-based mask as an anatomical boundary supports non-bias seed selection¹². Anatomically adjacent brain regions share similar time courses that are correlated at rest^{18,19}.

PPI specifically models the change in interaction between the seed ROI and other brain regions, as such, non-task specific baseline interactions due to anatomical proximity (e.g., between regions in the anatomical mask) will be disregarded, and only variations in timeseries that specifically relate to task modulation are modelled¹². To model cortical correlations that vary with task performance, PPI analysis generates an interaction regressor (i.e., the PPI term) derived from the timeseries of a specific seed ROI and the time course of the experimental (memory) task. This approach models the influence of one neural system (the seed) on another while accounting for the experimental context (memory task performance)^{12,17}. In essence, PPI considers the modulatory effect of the specific task condition on the functional interaction between ROI and whole-brain regions.

Connectivity analysis using seed to voxel-wise PPI analysis allows to model the functional memory connectome at the whole-brain level. PPI is a data-driven, directed functional connectivity approach that employs regression to investigate changes in the relationship between a region of interest (ROI) and the rest of the brain resulting from experimental manipulations¹⁷. Benchmark evidence shows its best suitability to model task-modulated functional networks at the voxel-wise level compared to other types of functional connectivity analyses (O'Reilly et al., 2012). PPI models voxel-wise connectivity specific to task performance based on temporal fluctuations in the timeseries of a region of interest known to be task-modulated. Consequently, seed to voxelwise PPI allows to assess the specific effects of task modulation on the whole-brain regions by using a seed region, *a priori* known to be involved or activated during task performance, such as the MTL herein.

Reviewer 5

This study presents long-term memory network changes in patients who have undergone brain resection and the impact of plasticity on memory formation.

The authors evaluated using fMRI and memory performance data before brain resection, three to 12 months after resection, and 10 years later.

The results of this study observed specific changes in the memory network and identified regions that are important for long-term verbal and visual memory recovery.

Notable is that this study tracks fMRI and memory performance data in people with epilepsy (PWE) for up to ten years following anterior temporal lobe resection (ATLR).

I was very impressed with the patience and hard work of the researchers who waited until 10 years after the lobotomy to get the data.

The data obtained and analyzed in this study is worth a decade and is useful for analyzing long-term memory network changes and the impact of plasticity on memory formation.

However, the results of the study are preliminary and the evidence for the results presented is lacking.

1. Memory formation change.

- 1.1 Currently, the explanation for these results is incomplete and requires further analysis. Patients' memory performance after ATLR was not consistent across all patients, so it is necessary to analyze the data separately based on memory performance. For example, individual L-ATLR

subjects had 50% improvement and 33% decline in verbal memory and 42% improvement and 50% decline in visual memory performance, respectively. When the memory performance result is divided into improvement, maintenance, and decline, the chance level is 33%. Therefore, a 33% decrease in verbal memory performance and a 42% improvement in visual memory performance are both above the chance level and therefore significant. However, the authors do not explain and analyze these memory performance results, so further analysis is needed.

We thank you for your review.

In a longitudinal study with a 5-to-22-year follow-up that compared memory and executive functions in 161 people undergoing ATR with those in 208 non-operated people with focal epilepsy, Helmstaedter, Elger and Vogt (2018)²⁴ showed that in both operated and non-operated groups, seizure control and reduction in ASM load were respectively associated with recovery of memory and executive functions. This was independent of the surgery effect, indicating no disadvantage for the surgical group. Later age at time of surgery was predictive of poorer memory recovery, and following the initial hit of surgery on verbal memory function, progressive memory decline was only seen in those with uncontrolled seizures. Similarly, Baxendale, Thompson and Duncan (2012)²⁵ showed in a 5-year follow-up study, that the long-term effect of surgical intervention includes cognitive decline that relates to temporal lobe functions, as opposed to IQ levels, in people with uncontrolled seizures.

We used a RCI of the whole sample of controls and PWE included in our study to assess if PWE had clinically significant decline or change in memory between assessments. The way changes are reported for reliable clinical relevance is based on RCI in previous neuropsychology and fMRI-based studies^{1,26-28}. Accordingly, from 3-12 months to 10 years, changes in z-scores that deviated more than the RCI boundaries (verbal memory [-0.40; 0.31] and visual memory [-0.16; 0.58]), based on the whole samples' mean and standard deviation, were determined as clinically significant.

We have performed further analyses to look into factors that are associated with individual differences in memory performance. Factors associated with preoperative reorganization included age of onset and epilepsy duration and seizure burden (Fleury 2022 Epilepsia, and Sidhu 2015 Neurology). In this analysis we have looked at differences from short-term to long-term postoperatively, regardless of what has happened in preoperative reorganization. Our analysis uses a flexible factorial design, capturing the individual changes in connectivity at two intervals following surgery, regardless of preoperative plasticity. What is common to all, is the interval of plasticity changes after resection. However, factors that can affect the postoperative memory network include changes in ASM and also the effect of pathology. We have therefore looked into these factors separately.

1. To address the possible effect of preoperative pathology on long-term plasticity, we performed post-hoc analyses separately in each ATR group and for each MTL seed and subsequent memory fMRI task. Two samples t-tests were conducted separately in each ATR group and for each MTL seed and subsequent memory fMRI task comparing the longitudinal connectivity changes in people with HS pathology with those in people without HS.

We found that people with HS show reduced plasticity (i.e., less longitudinal increases in connectivity), with posterior MTL regions that are supportive of memory functions. After both left and right ATR, this included reduced long-term plasticity with the remnant posterior fusiform gyrus seeding from the remnant MTL seed for successful word encoding, and seeding from the contralesional hippocampal seed for successful face encoding.

2. For ASM, we have re-analysed our results using the number ASMs as regressor of no interest. The changes in memory networks from 3-12 months to 10 years specifically associated with long-term memory improvement were also correlated with change in memory scores while treating individual ASM intake at both 3-12 months and 10 years as continuous regressors of no interest.

Ten years postoperatively, 10 PWE were off medications and 15 were on ASMs. Out of 12 left ATR, 4 were off medications (including cessation of Tegretol retard, Zonisamide, Topiramate, Levetiracetam, Lamotrigine, and Carbamazepine) while 6 out of 13 right ATR were off (including

cessation of Tegretol retard, Levetiracetam, Lamotrigine, Keppra, and Oxcarbazepine). Of those who remained on ASMs, 3 continued to have seizures.

3. In response to your comment, we have also investigated the relationship between change in memory function from T2 to T3 in people who are off medicines versus people are on medicine, and the association of ongoing seizures with memory changes from 3-12 months to 10 years, along with the relationship of preoperative HS pathology on long-term memory. Fisher-Pitman permutation tests indicated that ASM intake (on/off) had a significant effect on recovery of visual memory with people who ceased medication recovering significantly better than those on medication at 10 years but not for verbal memory changes. There was no significant difference in recovery of verbal nor visual memory between individuals with ongoing seizures and those were are seizure-free at 10 years. People with HS performed significantly worse 10 years post-ATLR for both verbal and visual memory. This is described in Results and Supplementary Results.

Changes to the manuscript related to post-hoc assessment of HS impact on network changes are described response 7.1 to reviewer 3 (pages 18-19), those associated with the addition of ASM as regressor of no interest are described response 7.4 to reviewer 3 (pages 22-24). Other changes are highlighted below in italics.

Methods, Statistical analyses, Correlation between functional reorganization in TLE and memory recovery, Page 38

“Three-way ANCOVAs, *controlling for IQ and changes in ASM intake*, were conducted for each MTL seed and successful memory contrast (words/faces) to investigate differences in functional connectivity from 3–12-month and 10-year follow-ups related with improvement in memory functions over this timeline.

Specifically, individual changes in memory z-scores were correlated with changes in task-modulated fMRI connectivity over time, modelling the fMRI biomarkers more strongly associated with those whose memory successfully improved compared to those who did not. Differences in z-scores (i.e., BMIPB I or II verbal and visual learning scores converted into age-normalized z-scores) from 3–12 months to 10 years were used as continuous variables.”

- 1.2 Also, while memory decline is typically expected with ATR, this study suggests that memory performance improved at the group level.

We thank you for your review. Performing a literature review, we found in a longitudinal study with a 5-to-22-year follow-up that seizures control and reduction in ASM load were respectively associated with recovery of memory and executive functions (Helmstaedter, Elger and Vogt, 2018)²⁴. This conclusion was drawn from comparing memory and executive functions in people undergoing ATR with functions in non-operated people with focal epilepsy and the same observation was made in both operated and non-operated groups. This was independent of the surgery effect, indicating no disadvantage for the surgical group. Later age at time of surgery was predictive of poorer memory recovery, and following the initial hit of surgery on verbal memory function, progressive memory decline was only seen in those with uncontrolled seizures (Helmstaedter, Elger and Vogt, 2018). Similarly, Baxendale, Thompson and Duncan (2012)²⁵ showed in a 5-year follow-up study, that the effect of surgical intervention includes cognitive decline that relates to temporal lobe functions in people with uncontrolled seizures, in contrast to the generalised cognitive decline in refractory focal epilepsy. The findings of both long-term longitudinal studies align with our neuropsychology results. We have added relevant literature to the introduction and discussion. Changes are highlighted in italics.

Introduction, Page 4:

“In medication-refractory focal epilepsy, seizures are associated with progressive cognitive decline over multiple domains, including memory and executive functions.¹⁻³ Surgical resection can bring seizure control^{1,4} which is the strongest predictor of cognitive remission, regardless of the effect of surgery on function.^{1,2} Anterior temporal lobe resection (ATR) is the most commonly performed surgical treatment for refractory temporal lobe epilepsy (TLE).⁵ This involves the partial removal of

temporal neocortical and mesial temporal structures including the anterior hippocampus, parahippocampal and fusiform gyri; structures critical for successful memory formation.^{6,7} Although ATLR may prevent the progressive decline observed in refractory epilepsy, up to 40% of individuals experience significant episodic memory deficits postoperatively.^{5,8} Identifying regions crucial for effective memory reorganization that *could* be spared in epilepsy surgery may help mitigate long-term postoperative memory loss. Additionally, understanding post-operative plasticity could assist the development of strategies to promote long-term memory recovery.”

Discussion, Long-term postoperative memory changes, Pages 22-23

“Neuropsychology studies demonstrate that ATLR is characterized by an immediate ‘hit’ on episodic memory functions^{1,22}, which in seizure-free people is followed by either cognitive stabilization or ongoing improvement as recovery time increases (beyond two years).^{1,2,23} In line with current literature, postsurgical PWE did not present with the learning/test-retest effect in verbal memory, suggesting an early surgical insult on memory function. Yet beyond 3–12 months, our predominantly seizure-free cohort demonstrated cognitive stability at the group-level and patient-specific recovery was evident. From 3–12-months to 10 years, 50% of left ATLR and 38% of right ATLR showed clinically significant improvement in verbal and visual memory, respectively. [...]. Ten years postoperatively, 75% of left ATLR and 58% of right ATLR had good verbal and visual memory outcomes compared to preoperatively, with improved performance or return to baseline. Enhancement of baseline performance possibly translates to the *removal of nociferous cortex and corresponding release of functions and reserve capacities previously suppressed by epileptic activity.*^{24,25”}

- 1.3 As an explanation for this memory plasticity, it would be interesting to further analyze the fMRI data before and after ATLR. For example, if the connectivity supporting verbal or visual memory formation before ATLR was A, then after ATLR the A connectivity was disconnected and replaced by B connectivity. It would be interesting to explain the neurological mechanisms or reasons for these connectivity changes or substitutions.

We thank you for your suggestion. To address this, we have analysed the changes in preoperative to 3-12month connectivity that correlated with memory improvement over the long term. Preoperative PPI that is supportive of preoperative memory function is published (Fleury et al., 2022)²¹.

We found that early after surgery, disconnected preoperative connectivity is replaced by new connectivity patterns to anterior contralateral MTL regions which efficiently supports longer term verbal and visual memory functions, while connections with posterior MTL regions are not immediately efficient. In PWE who continue to improve, posterior MTL regions like the fusiform and parahippocampus become efficiently involved the cognitive network over the longer term. For verbal memory recovery, increased plasticity in the shorter term to the right frontal cortex, particularly in the right cingulum and anterior orbitofrontal cortex and is sustained over the longer term is supportive of function in both ATLR groups. This possibly serves as a foundation for ongoing longer-term connectivity increases within the right cingulum in the left ATLR group. For visual memory recovery, shorter term plasticity in the contralateral middle temporal cortex continues over the longer term, while ipsilateral increases in functional couplings with the right basal ganglia (external globus pallidus) is expanded to the contralateral hemisphere over the longer term in PWE with long term visual memory improvement.

We have added this supplementary analysis to our results and discussed the main findings as below. Changes are highlighted in response 5 to reviewer 3, pages 14-15.

2. Changes in functional connectivity 3-12 months to 10 years post-surgery relative to changes in controls. The explanation for these results is incomplete and requires further analysis.

2.1 The results for the statistical analysis of functional connectivity given in Figures 2-(A) and 3-(A) are uncorrected. This does not appear to solve the multiple comparison problem that occurs with fMRI voxel data. Therefore, further analysis should be done to correct the p-value of that result to solve the multiple comparison problem.

We thank you for your comment. The correlation of longitudinal connectivity changes with memory improvement is a highly specific event-related *T*-contrast²⁹, with the analysis of functional connectivity also being a more sensitive method than univariate analyses for describing network-level functional architecture^{10,11}. As a group, the connectivity survives FWE. As an example, we have inserted the result table of group analyses of successful memory connectivity 10 years after surgery, with at FWE cluster-extent thresholding.

Table: Cross-sectional connectivity of the successful memory networks at the 10-year follow-up, corrected using family-wise error rate.

	Left ATLR	Right ATLR	Controls	
Left MTL seed	Left post hippocampus -20 -28 -6, $P = 0.003^b$	Left ant hippocampus -30 -8 -22, $P = 0.017^a$	Left ant hippocampus -30 -28 -12, $P = 0.019^a$	
	Left post fusiform (temporal) -44 -46 -20, $P = 0.001^a$	Left post parahippocampal G -24 -30 -12, $P = 0.026^a$	Left mid-post hippocampus -26 -24 -16, $P = 0.050^a$	
	Left post parahippocampal G -28 -38 -12, $P = 0.015^b$	Left post fusiform (temporal) -38 -24 -18, $P < 0.001^a$	Left mid fusiform -28 -40 -20, $P = 0.030^a$	
	Right mid hippocampus 18 -24 -12, $P < 0.002^a$	Right post fusiform (temp-occ.) 24 -70 -6, $P = 0.004^a$	Right post fusiform (occ.) 22 -50 -12, $P = 0.043^a$	
	Right mid-post fusiform (temp-occ.) 36 -54 -18, $P < 0.001^a$	Right post parahipp./fusiform G 36 -38 -14, $P = 0.013^a$		
	Left inferior temporal gyrus -50 -52 -18, FWEc = 0.025	Left inferior parietal gyrus -58 -34 50, FWEc = 0.001		
	Left post cingulate cortex -8 -42 8, FWEc = 0.021			
	Right precuneus 4 -54 70, FWEc = 0.047			
	Verbal Memory	Left post hippocampus -36 -28 -4, $P = 0.021^b$	Left amygdala -28 -8 -12, $P = 0.044^a$	Left post fusiform (occ.) -26 -68 -14, $P = 0.007^a$
		Left post fusiform (temp-occ.) -34 -62 -16, $P < 0.001^a$	Left mid hippocampus -26 -22 -12, $P = 0.018^a$	Left mid fusiform (temporal) -28 -46 -20, $P = 0.036^a$
Right ant hippocampus 32 -18 -20, $P = 0.029^a$		Right post fusiform (temp-occ.) -22 -76 -14, $P = 0.001^a$	Right ant-mid hippocampus 34 -18 -8, $P = 0.035^a$	
Right post hippocampus 26 -30 -10, $P = 0.019^a$		Left mid fusiform gyrus (temp.) -28 -38 -24, $P = 0.002^a$	Right post hippocampus 30 -30 -8, $P = 0.025^a$	
Right post parahippocampal G 18 -40 -4, $P = 0.015^a$		Left ant parahippocampal G -30 -12 -26, $P = 0.026^a$	Right post fusiform (occ.) 46 -70 -18, $P = 0.043^a$	
Right mid fusiform (temp.) 28 -38 -22, $P = 0.029^a$		Left ant hippocampus -34 -14 -22, $P = 0.007^a$		
Right post fusiform (temp-occ.) 24 -84 -16, $P = 0.001^a$		Right post hippocampus 26 -36 -6, $P = 0.049^b$		
		Right mid-post hippocampus 34 -18 -8, $P = 0.047^b$		
		Right post parahippocampal G 24 -44 -4, $P = 0.018^b$		
		Right post fusiform (temp-occ.) 28 -78 -16, $P = 0.025^a$		
Right MTL seed		Right mid-post fusiform (temp.) 38 -36 -28, $P < 0.001^a$		
		Left inferior parietal G -56 -24 48, FWEc = 0.001		
		Right mid frontal G 40 -2 60, FWEc = 0.035		
	Visual Memory	Left post hippocampus -20 -32 -6, $P = 0.012^b$	Left ant hippocampus -30 -8 -22, $P = 0.017^a$	Left ant hippocampus/amygdala -26 -8 -20, $P = 0.041^a$
		Left post fusiform (lateral-occ.) -44 -46 -20, $P = 0.001^a$	Left mid parahippocampal G -30 -26 -1, $P = 0.021^a$	Left mid hippocampus -26 -24 -16, $P = 0.050^a$
		Left post parahippocampal G -26 -38 -12, $P = 0.017^b$	Left post fusiform (temp-occ.) -34 -62 -8, $P = 0.005^a$	Right post fusiform (temp-occ.) 24 -50 -12, $P = 0.043^a$
		Right mid [para]hippocampus 18 -24 -12, $P = 0.002^a$	Left mid-post fusiform (temp.) -38 -24 -18, $P < 0.001^a$	Right mid parahippocampal G 26 -26 -16, $P = 0.043^a$
		Left post parahippocampal G 34 -44 -18, $P = 0.024^a$	Right post fusiform (temp-occ.) 24 -70 -6, $P = 0.004^a$	
		Right post fusiform (temp-occ.) 36 -54 -18, $P < 0.001^a$	Right post parahippocampal G 36 -38 -12, $P = 0.008^b$	

	Left inferior TG/lateral fusiform -44 -46 -20, FWEc = 0.005	Left lingual G 2 -80 -2, FWEc = 0.002	
		Right Cuneus 14 -72 24, FWEc=0,002	
	Left post fusiform (occ.) -34 -82 -16, $P < 0.001^a$	Left mid hippocampus -34 -22 -14, $P = 0.001^a$	Right mid-post hippocampus 28 -28 -6, $P = 0.029^a$
	Right mid hippocampus 18 -24 -10, $P = 0.008^a$	Left post fusiform (temp-occ.) -32 -52 -6, $P = 0.006^a$	
	Right ant parahippocampal G 22 -2 -24, $P = 0.002^a$	Left mid parahippocampal G -28 -28 -14, $P = 0.038^a$	
Right MTL seed	Right post fusiform (temp-occ.) 34 -48 -16, $P = 0.001^a$	Right post fusiform (temporal) 30 -34 -20, $P = 0.003^a$	
		Right post fusiform (lateral-occ.) 42 -58 -22, $P = 0.008^a$	
		Right post parahippocampal G 34 -34 -16, $P < 0.001^b$	
		Left mid temporal G -44 -52 -4, FWEc = 0.013	

Mean functional connectivity as measured by one-sample t -tests separately for each MTL seed and subsequent memory fMRI task. MTL-to-neocortex connectivity is reported at voxel-wise cluster-defining threshold $P < 0.001$, corrected using cluster-extent family-wise error (FWE) rate < 0.05 (i.e., FWEc). ^a $P < 0.05$, FWE correction using a 6mm sphere in contralesional MTL regions. ^b $P < 0.05$, FWE correction using a 3mm sphere in remnant MTL regions.

However, the results of longitudinal connectivity changes are yielded from a T -contrast looking at changes in PWE compared with longitudinal changes in controls over the same timeline, justifying for our thresholding methods. Other studies looking at PPI correlations used similar thresholding methods^{16,21,30}. For *a priori* regions of interest, we performed FWE correction. To address the multiple comparison problem, we have

- (1) used a flexible factorial design: significant results are only those that are not seen in controls and as such, any network changes also occurring in healthy controls are removed from the statistical glass brain. This is included in our Methods as shown below.
- (2) verified that reported correlations are still significant within a binary mask of the main PPI effect, as done in previous PPI correlation studies³¹⁻³³. We have added this to our Methods. Changes to our manuscript are highlighted in italics below.

Methods, Statistical analyses, Long-term changes in functional connectivity from 3–12-month to 10-year follow-ups, Pages 37-38:

“In summary, flexible factorial t -contrasts modelled in each ATLR group, the within-subject differences in MTL-seeded, whole-brain functional connectivity across postoperative follow-ups, beyond the connectivity changes seen in controls. *This model allowed control of both age-related and scanner-change effects.*”

Methods, Statistical analyses, Statistical thresholds, Page 35:

“Group comparison and correlation analyses generated highly specific MTL-to-whole-brain, longitudinal, event-related t -contrasts⁵⁰. Thus, at the extra-MTL level, functional connectivity is reported at an exploratory $P < 0.001$ threshold (uncorrected), alike previous longitudinal, event-related and network fMRI studies^{5,12,49} and correlation results are reported at exploratory $P < 0.001$, masked within binary group masks of the main PPI effect, in line with PPI correlations studies^{28,51,52}.”

- 2.2 Furthermore, the neurological implications of changes in functional connectivity in verbal and visual memory need to be explained. Currently, it appears to simply list the results. Therefore, it is necessary to supplement the explanation in the results or discussion.

We thank you for your review. In response, we have amended our manuscript by adding literature on neurological implications of specific verbal and visual network changes, and network changes that are task specific are further discussed.

Task lateralization appears to vary according to cognitive demands, as shown in previous studies: left hemispheric lateralization during successful word encoding and bilateral temporal involvement during successful face encoding¹³. Sidhu et al. (2016)¹⁶ demonstrated that 12 months after surgery, increases in extra-temporal activations contralateral to the side of surgery (i.e., in the anterior cingulum, orbitofrontal cortex, and insula) support postoperative verbal and visual memory functions. By 10 years, ongoing neocortical plasticity within the anterior cingulum is observed in the verbal memory

network after left ATR, potentially due to a greater impact of left hemispheric resection compared to the right. Long-term reorganization of the visual memory network continues after both left and right ATR, likely due to its extensive and bilateral nature encompassing medial temporal, temporal, and frontal regions relevant to visual tasks.

We have discussed this task-dependent differential trend in extra-MTL plasticity as below. Changes to the manuscript are highlighted in italics and all changes of above-mentioned sections are tracked in the main manuscript's discussion.

Discussion, Long-term support from structures of goal-directed behaviours, Pages 25-27

“Long-term visual memory recovery following right ATR was associated with new functional connections with the external globus pallidus bilaterally; ipsilaterally in the short term post-surgery and also contralesionally by 10 years.

Although known modulators of the motor control system, the globus pallidus interna and externa are also part of frontal-striatal loops subserving spatial memory encoding and controlling working memory capacity to store relevant information^{37,38}. In concert with the preparatory engagement of bilateral frontal regions, left external pallidal activity allows subsequent information filtering during encoding of visuo-spatial working memory and its task-related recruitment reduces unnecessary storage of distractors³⁷. Correspondingly, people with right-sided basal ganglia injury exhibit deficits in encoding and recall of visual working memory under equally weighted attentional demands³⁹. This implies corpus striatal engagement beyond memory processes that depend on selective attention mechanisms.

For verbal memory recovery, the importance of right frontal engagement was enhanced 3–12 months after both ATR compared to preoperatively, including the right middle cingulate and anterior orbitofrontal cortices. The absence of long-term decreases in frontal connectivity suggests this plasticity was sustained and adaptive 10 years post-surgery. There were ongoing connectivity increases between remnant MTL and contralesional anterior cingulum long-term after left ATR, possibly reflecting a greater impact of left than right-sided resection on verbal mnemonic processes.

Reorganization of verbal memory function to the cingulate cortex is critical in TLE. PWE exhibit memory fMRI activations in the anterior cingulate cortex that support successful word encoding both before and short-term after surgery^{7,40} and correlate with improvement 12 months after left ATR⁹. The anterior cingulate cortex shares direct structural connections with the anterior parahippocampal gyrus⁴¹ and is an essential node of the default mode and salience networks⁴². In healthy people, heightened functional connectivity between right cingulum and temporal regions is associated with increased cognitive load and sustained attention⁴³. Increased functional connectivity and structural integrity of the contralesional anterior cingulum may subservise better global cognition⁴⁴ and flexible learning⁴⁵, shifting attention allocation for higher processing speed, as evidenced in post-stroke cohorts.

Recent hippocampal pathology reports have shed light on the hippocampus' role in binding elements regardless of duration of retention; both for short-term working and long-term memories. They denote similar activity of hippocampal neurons underpinning long-term memory during short-term processes requiring great memory load and cognitive control^{46,47}. This activity is synchronised with frontal engagement based on cognitive demand⁴⁷. It is therefore plausible that the surgical strain on function is compensated by new functional couplings with extra-temporal areas subserving goal-directed behaviours, such as the globus pallidus and cingulate cortex, and facilitating successful formation of long-term memory.”

Discussion, Task-specific patterns of extra-MTL plasticity, Pages 27-28

“The long-term plasticity of extra-temporal memory networks varied with memory. Visual memory recovery underpinned a bilateral network involvement with compensatory mechanisms engaging both temporal and frontal regions a decade postoperatively, while verbal memory displayed a more lateralized pattern of compensation, primarily involving the reinstatement of posterior MTL plasticity disconnected during surgery.

Preoperatively, successful visual memory encoding correlates with activation fMRI maxima in both anterior hippocampi, and with the bilateral middle and inferior temporal cortices and right frontal areas such as the right orbitofrontal cortex in healthy subjects^{7,16}. This extensive network's architecture may explain the observed, long-term increases in connectivity with neocortical areas typically part of

the episodic memory network. Such neuroplasticity adaptively compensated for functional loss in the ipsilateral MTL following right ATR, and for the disruption of the bilateral temporal cognitive network after left ATR.

Verbal memory encoding is predominantly lateralized to the left hemisphere in healthy subjects^{7,48,49}. As such, individuals undergoing right-sided resection may maintain nearly intact, established neocortical networks of verbal memory arising from the left MTL. This may account for the observed long-term plasticity after right ATR, which, beyond 3–12 months, primarily involved increased connectivity within the remnant right MTL.

These differential plasticity patterns underscore the dynamic nature of cognitive network architectures, adapting according to the task's functional demands and learning strategies involved, as previously proposed⁵⁰.

2. Correlation between longitudinal changes in connectivity and memory recovery 3-12 months to 10 years postoperatively. The correlation of connectivity with verbal memory and visual memory performance needs to be presented in a figure. The authors state in the main text that there was a significant correlation and presented a table. However, it would be nice to have a figure that makes sense intuitively.

We thank you for your feedback. In response, specific regions of interest (ROIs) that are discussed in our manuscript have been added in a graphical form to better illustrate their relationship with memory function. Manuscript discussion of the ROIs is highlighted in bold below along with manuscript changes in italics. The scatterplots integrated in the main manuscript's Figures 3, 4 and 5 are shown below for illustration.

Discussion, Page 22

*“We identified key increases in functional connections to posterior MTL regions, including the **right posterior parahippocampus** and **remnant /bilateral posterior fusiform gyri**, which were central to long-term recovery of both verbal and visual memory. Long-term improvements in verbal and visual memory after left and right ATR, respectively, correlated with longitudinal connectivity increases with the contralesional **anterior cingulate cortex** and **external globus pallidus**.”*

Example Figure 1. Verbal memory correlation. Scatter plots of eigenvariates extracted from the functional connectivity changes in PWE compared with healthy controls. Top blue graphs represent adaptive connectivity increases with the contralesional parahippocampus and anterior cingulate cortex after left-sided surgery. Bottom purple graphs show adaptive connectivity increases with the remnant right parahippocampus and fusiform gyrus after right-sided resection.

Example Figure 2. Visual memory correlation. Scatter plots of eigenvariates extracted from the functional connectivity changes in PWE compared with healthy controls. This included the remnant left fusiform gyrus and contralesional right anterior orbitofrontal cortex (OFC) after left-sided surgery (top blue graphs), and the remnant right parahippocampus and contralesional left globus pallidus after right-sided resection (bottom purple graphs).

To display the neural correlates of memory function in these ROIs, we extracted eigenvariates from brain regions identified by the longitudinal ANOVA with positive memory correlations.

Before eigenvariates extraction, we created longitudinal ‘connectivity difference’ contrast images¹⁵ for every participant. That is, we subtracted for each subject, the 1st level *T*-contrast of successful memory connectivity from the subject-level PPI analysis at each T2 and T3 timepoint, using the SPM toolbox ‘ImCalc’. This yielded two ‘connectivity difference’ *T*-contrasts per subject, MTL seed, and memory task: increased connectivity and decreased connectivity at T3 compared to T2 (3 or 12 months, depending on the subject). From these contrasts, the extracted eigenvariates predicted individual memory changes, adjusted using specific *F*-contrasts (e.g., controls and left ATR effects [1 0 0; 0 1 0]). We had added this information to supplementary methods.

This method was performed after discussion with Professor Karl Friston who suggested this is the best method for confirming firstly the gPPI model validity and next the correlations with memory function

Supplementary Methods, Eigenvariates extraction for graphical presentation, Page 11:

*“Based on full factorial ANOVAs of ‘connectivity difference’ contrast images¹⁵, eigenvariates were extracted from the functional connectivity changes in ATR compared with healthy controls, controlling for change in ASM intake and IQ. These eigenvariates, adjusted for the *F*-contrast of interest, were used to predict individual memory changes, thereby confirming the gPPI construct validity in modelling neural substrates of memory function.”*

Overall, the research is interesting and important.

Therefore, it would be good to supplement the explanation of the results and supplement the results with additional analysis.

References

1. Baxendale, S. & Thompson, P. The association of cognitive phenotypes with postoperative outcomes after epilepsy surgery in patients with temporal lobe epilepsy. *Epilepsy & Behavior* **112**, 107386 (2020).

2. Chan, D., *et al.* Effect of high-dose simvastatin on cognitive, neuropsychiatric, and health-related quality-of-life measures in secondary progressive multiple sclerosis: secondary analyses from the MS-STAT randomised, placebo-controlled trial. *The Lancet Neurology* **16**, 591-600 (2017).
3. Chelune, G.J., Naugle, R.I., Lüders, H., Sedlak, J. & Awad, I.A. Individual change after epilepsy surgery: Practice effects and base-rate information. *Neuropsychology* **7**, 41 (1993).
4. Sidhu, M.K., *et al.* Memory fMRI predicts verbal memory decline after anterior temporal lobe resection. *Neurology* **84**, 1614 (2015).
5. Xiao, F., *et al.* Verbal fluency functional magnetic resonance imaging detects anti-seizure effects and affective side effects of perampanel in people with focal epilepsy. *Epilepsia* **64**, e9-e15 (2023).
6. van Lieshout, J., Debaene, W., Rapp, M., Noordmans, H.J. & Rutten, G.J. fMRI Resting-State Connectivity between Language and Nonlanguage Areas as Defined by Intraoperative Electrocortical Stimulation in Low-Grade Glioma Patients. *J Neurol Surg A Cent Eur Neurosurg* **82**, 357-363 (2021).
7. Cirillo, S., *et al.* Comparison between inferior frontal gyrus intrinsic connectivity network and verb-generation task fMRI network for presurgical language mapping in healthy controls and in glioma patients. *Brain Imaging Behav* **16**, 2569-2585 (2022).
8. Rai, S., Graff, K., Tansey, R. & Bray, S. How do tasks impact the reliability of fMRI functional connectivity? *Hum Brain Mapp* **45**, e26535 (2024).
9. Cheung, M.C., Chan, A.S., Lam, J.M. & Chan, Y.L. Pre- and postoperative fMRI and clinical memory performance in temporal lobe epilepsy. *J Neurol Neurosurg Psychiatry* **80**, 1099-1106 (2009).
10. Friston, K.J. Functional and effective connectivity: a review. *Brain connectivity* **1**, 13-36 (2011).
11. Pillet, I., Op de Beeck, H. & Lee Masson, H. A Comparison of Functional Networks Derived From Representational Similarity, Functional Connectivity, and Univariate Analyses. *Frontiers in Neuroscience* **13**(2020).
12. O'Reilly, J.X., Woolrich, M.W., Behrens, T.E.J., Smith, S.M. & Johansen-Berg, H. Tools of the trade: psychophysiological interactions and functional connectivity. *Social cognitive and affective neuroscience* **7**, 604-609 (2012).
13. Sidhu, M.K., *et al.* A functional magnetic resonance imaging study mapping the episodic memory encoding network in temporal lobe epilepsy. *Brain* **136**, 1868-1888 (2013).
14. Bonelli, S.B., *et al.* Imaging memory in temporal lobe epilepsy: predicting the effects of temporal lobe resection. *Brain* **133**, 1186-1199 (2010).
15. Bonelli, S.B., *et al.* Memory reorganization following anterior temporal lobe resection: a longitudinal functional MRI study. *Brain* **136**, 1889-1900 (2013).
16. Sidhu, M.K., *et al.* Memory network plasticity after temporal lobe resection: a longitudinal functional imaging study. *Brain* **139**, 415-430 (2016).

17. Friston, K.J., *et al.* Psychophysiological and Modulatory Interactions in Neuroimaging. *NeuroImage* **6**, 218-229 (1997).
18. Greicius, M.D., Supekar, K., Menon, V. & Dougherty, R.F. Resting-state functional connectivity reflects structural connectivity in the default mode network. *Cerebral cortex* **19**, 72-78 (2009).
19. Honey, C.J., *et al.* Predicting human resting-state functional connectivity from structural connectivity. *Proceedings of the National Academy of Sciences* **106**, 2035-2040 (2009).
20. Gueorguieva, R. & Krystal, J.H. Move Over ANOVA: Progress in Analyzing Repeated-Measures Data and Its Reflection in Papers Published in the Archives of General Psychiatry. *Archives of General Psychiatry* **61**, 310-317 (2004).
21. Fleury, M., *et al.* Episodic memory network connectivity in temporal lobe epilepsy. *Epilepsia* **63**, 2597-2622 (2022).
22. Sidhu, M.K., *et al.* Factors affecting reorganisation of memory encoding networks in temporal lobe epilepsy. *Epilepsy Res* **110**, 1-9 (2015).
23. Thompson, P.J. & Duncan, J.S. Cognitive decline in severe intractable epilepsy. *Epilepsia* **46**, 1780-1787 (2005).
24. Helmstaedter, C., Elger, C.E. & Vogt, V.L. Cognitive outcomes more than 5 years after temporal lobe epilepsy surgery: Remarkable functional recovery when seizures are controlled. *Seizure* **62**, 116-123 (2018).
25. Baxendale, S., Thompson, P.J. & Duncan, J.S. Neuropsychological function in patients who have had epilepsy surgery: A long-term follow-up. *Epilepsy & Behavior* **23**, 24-29 (2012).
26. Baxendale, S., Thompson, P., Harkness, W. & Duncan, J. Predicting Memory Decline Following Epilepsy Surgery: A Multivariate Approach. *Epilepsia* **47**, 1887-1894 (2006).
27. Sone, D., *et al.* Optimal Surgical Extent for Memory and Seizure Outcome in Temporal Lobe Epilepsy. *Annals of Neurology* **91**, 131-144 (2022).
28. Binding, L.P., *et al.* The impact of temporal lobe epilepsy surgery on picture naming and its relationship to network metric change. *NeuroImage: Clinical* **38**, 103444 (2023).
29. Telzer, E.H., *et al.* Methodological considerations for developmental longitudinal fMRI research. *Developmental Cognitive Neuroscience* **33**, 149-160 (2018).
30. Trimmel, K., *et al.* Left temporal lobe language network connectivity in temporal lobe epilepsy. *Brain* **141**, 2406-2418 (2018).
31. Doucet, G., Osipowicz, K., Sharan, A., Sperling, M.R. & Tracy, J.I. Extratemporal functional connectivity impairments at rest are related to memory performance in mesial temporal epilepsy. *Hum Brain Mapp* **34**, 2202-2216 (2013).
32. Caciagli, L., *et al.* Thalamus and focal to bilateral seizures: A multiscale cognitive imaging study. *Neurology* **95**, e2427-e2441 (2020).

33. Trimmel, K., *et al.* Decoupling of functional and structural language networks in temporal lobe epilepsy. *Epilepsia* **62**, 2941-2954 (2021).

REVIEWER COMMENTS

Reviewer 1:

The paper is much improved. I have no further questions.

Reviewer 2:

The authors have addressed my concerns.

Reviewer 3:

I appreciate the authors for their elaborate responses to my comments.

Although they have responded appropriately to my comments, their response to my comment 7.5 about the entire cohort would be better included in the supplement. Otherwise, I have no further comment.

We thank you for your comment and positive feedback on our revisions. Based on your suggestion, we have moved the detailed description of the cohort to the supplementary information section for better clarity. This information is now included as follows in the Supplementary Information, Subject section (Page 1), and is highlighted in *bold-italics* below.

Supplementary Results, *Subjects*, Page 1:

“The initial cohort for this longitudinal study consisted of 90 individuals with temporal lobe epilepsy (TLE), including 43 with left-sided TLE, and 29 control participants. Of the 90 TLE participants, 65 underwent anterior temporal lobectomy (ATLR), with 30 having left-sided surgery. All participants were invited to participate in long-term follow-up assessments, of which 25 postsurgical individuals and 10 controls returned for long-term study. Among the 40 participants who did not return (18 of whom had left-sided ATLR), clinical data was available for 35 individuals. Of these, 74% (26/35) were seizure-free with an ILAE outcome of 1 at the 12-month clinical follow-up, including 75% of the left ATLR group and 74% of the right ATLR group. However, these participants did not consent to participate in the long-term follow-up imaging studies, and as such, no additional outcomes are reported for those who chose not to take part.”

We hope this addresses your comment.

Reviewer 4:

I appreciate the author's point-by-point revisions and their acknowledgment of the generalizability issue, which enhance the understanding of the study's limitations and contributions. I have a few additional comments:

1. You noted that ASM cessation improved visual memory recovery but not verbal memory. Could you speculate on this discrepancy? Are there specific antiepileptic medications that might have a more significant impact on cognitive recovery?

We thank the reviewer for this important comment and positive feedback on our revisions. **Topiramate, Zonisamide, Carbamazepine** and **Lamotrigine** are ASMs known to affect the functional language network and impact language function (Yasuda *et al.*, 2013; Xiao *et al.*, 2022; Wandschneider *et al.*, 2017; Xiao *et al.*, 2018). These medications may disproportionately impact verbal processes, thereby hindering verbal memory recovery. In our cohort of 25 individuals assessed for the effects of ASM on cognitive function, 15 remained on ASMs at the 10-year follow-up (8 following left ATLR). Of these, 10 were on combinations of ASMs that included one or more of the aforementioned ASMs, potentially

contributing to the observed discrepancy between verbal and visual memory recovery. Specifically, among the 15 people who continued medication:

- **Left ATLR group (n=8):**
 - 6 individuals were taking Topiramate, Zonisamide, Carbamazepine, or Lamotrigine. Of these, 3 were on polytherapy with ≥ 2 ASMs (including one individual on 3 ASMs), which is associated with greater disruption of cognitive function (Witt, Elger and Helmstaedter, 2015).
 - The remaining 2 individuals were either on Valproate or Levetiracetam monotherapy.
 - **Right ATLR group (n=7):**
 - 4 individuals were taking either one of Topiramate, Zonisamide, Carbamazepine, or Lamotrigine.
 - The remaining 3 were either on Valproate, Levetiracetam, or Oxcarbazepine.
- All 7 individuals in the right ATLR group were on monotherapy.

Additional descriptive statistics were conducted, and suggested that verbal memory recovery was more disrupted in individuals taking ASMs affecting verbal processes:

- Individuals on potential cognitive disruptors (n = 10): median verbal memory change = **0** (IQR = -0.45–0.70).
- Individuals taking other ASMs (n=5): median verbal memory change = **0.63** (IQR= 0–1.36), a RCI-determined clinically significant improvement above the upper limit of the RCI (0.31) using a 95% confidence interval.

These descriptive statistics suggest that verbal memory recovery is more sensitive to ASMs that disrupt language-related functions. We have added this to the discussion as highlighted in *italics-bold* below, and included a new section in the supplementary information, highlighted in *italics*. We hope this explanation adequately addresses your query.

Discussion, Strengths and limitations, Pages 29-30:

“Future studies with larger numbers should investigate the modulatory effect of specific ASMs and differential seizure outcomes on network recovery. *In particular, verbal memory changes may be more sensitive to ASMs known to disrupt language-related function*⁵²⁻⁵⁵ (see SI), compared to visual memory recovery.”

Supplementary Results, *Language-disrupting antiseizure medication*, Pages 8-9:

“Topiramate, Zonisamide, Carbamazepine and Lamotrigine are ASMs known to affect the functional language network and impact language function²⁻⁵. These medications may disproportionately impact verbal processes, thereby hindering verbal memory recovery. In our cohort of 25 individuals assessed for the effects of ASM on cognitive function, 15 remained on ASMs at the 10-year follow-up (8 following left ATLR). Of these, 10 were prescribed combinations of ASMs that included one or more of the language-disrupting ASMs, potentially contributing to the observed discrepancy between verbal and visual memory recovery in our cohort. Specifically, among the 15 people who continued medication: Left ATLR group (n=8): six individuals were taking Topiramate, Zonisamide, Carbamazepine, or Lamotrigine. Of these, three were on polytherapy of two or more ASMs, which is associated with greater disruption of cognitive function⁶. The remaining two individuals were either on Valproate or Levetiracetam monotherapy.

Right ATLR group (n=7): four individuals were taking either one of Topiramate, Zonisamide, Carbamazepine, or Lamotrigine. The remaining three persons were either on Valproate, Levetiracetam, or Oxcarbazepine. All 7 individuals in the right ATLR group were on monotherapy.

Additional descriptive statistics were conducted, and suggested that verbal memory recovery was more disrupted in individuals taking ASMs affecting verbal processes:

- Individuals taking language-disrupting ASMs (n = 10): median verbal memory change = **0** (IQR = -0.45–0.70).
- Individuals taking other ASMs (n=5): median verbal memory change = **0.63** (IQR= 0–1.36), a clinically significant improvement above the upper limit of the RCI (RCI-improvement of 0.31) using a 95% confidence interval.

These descriptive statistics suggest that verbal memory recovery may be sensitive to ASMs that disrupt language-related functions.”

2. The finding that ongoing seizures 10 years post-surgery did not affect memory recovery seems counterintuitive. Could you discuss potential explanations for this?

We thank you for your comment. In our cohort of 25 people who were assessed for the effect of ongoing seizures, only 3 continued to have ongoing seizures (2 left ATR). It is possible that the small sample size of individuals with ongoing seizures limited our ability to detect a significant effect on memory recovery.

While the results may appear counterintuitive, the small number of individuals with ongoing seizures in our cohort may limit the generalizability of this finding. Further studies with larger sample sizes and a more detailed exploration of seizure frequency, duration, and other clinical variables may be necessary to better understand the relationship between ongoing seizures and cognitive recovery in the long term. We have added it to our limitation.

Discussion, Strengths and limitations, Page 29:

“The postsurgical cohort was primarily seizure-free, limiting the generalisability of our results to people with a good prognosis. ***The small number of participants (3/25) with ongoing seizures may also restrict the applicability of our finding that ongoing seizures did not affect memory recovery.*** Future studies with larger numbers should investigate the modulatory effect of specific ASMs and differential seizure outcomes on network recovery.”

3. I would appreciate a more detailed explanation of your choice to use the Reliable Change Index (RCI) for determining clinically significant change. How might results differ with alternative methods? This clarification would provide valuable insight into your findings.

We thank the reviewer for this insightful comment.

We acknowledge the limitations inherent in choosing the RCI, particularly as it relies on published BMIPB norms (Coughlan, Oddy and Crawford, 2007), which may be influenced by external factors affecting individual performance on the day of testing. Selecting a 95% confidence interval is inherently subjective; a more lenient confidence interval (e.g., 90% or 80%) might have identified additional changes. However, we prioritized reducing the risk of false positives for cognitive change to enhance specificity, ensuring the robustness of detected cognitive recovery and aligning with the primary research aim of understanding the neural mechanisms underlying cognitive recovery.

Alternative methods, such as analysing memory changes using the difference in z-scores without RCI could allow for the exploration of direct linear relationships between time points and individual-level changes. However, these approaches are more susceptible to confounding influences, such as variations in sleep quality on the day of testing, seizure occurrence near the time of neuropsychological assessment, or external stressors unrelated to the experimental setting; all of which can disproportionately impact measured changes and reduce the reliability of the results (Williamson *et al.*, 2005; Drane *et al.*, 2016; Hermann *et al.*, 1996). The RCI provides a theoretical advantage by evaluating changes against standardized benchmarks, enhancing reliability by accounting for variability from sources of errors, such as practice effect and test-retest reliability (Baxendale and Thompson, 2020; Hermann *et al.*, 1996). While it has its limitations, the RCI remains a valuable tool in both cognitive neuroscience and clinical practice, especially when calculation methods are explained, as included in our methods section (Blampied, 2022).

We hope this explanation clarifies our rationale for using the RCI and the balance we sought between methodological rigor and sensitivity to clinically meaningful changes. Additionally, we have acknowledged the limitations of the RCI and incorporated this into the study's limitations section.

Discussion, Strengths and limitations, Page 30:

“The postsurgical cohort was primarily seizure-free, limiting the generalisability of our results to people with a good prognosis. The small number of participants (3/25) with ongoing seizures may also restrict the applicability of our finding that ongoing seizures did not affect memory recovery. Future studies with larger numbers should investigate the modulatory effect of specific ASMs and differential seizure outcomes on network recovery. In particular, verbal memory changes may be more sensitive to ASMs known to disrupt language-related function⁵²⁻⁵⁵ (see SI), compared to visual memory recovery. *The RCI provided a standardized approach to assess cognitive change but is limited by its reliance on published norms and subjective confidence intervals, which may affect its generalizability and sensitivity.*”

Reviewer 5:

Dear Author.

I would like to express my sincere gratitude for your dedicated effort in revising and resubmitting the manuscript. I previously provided the following comments:

1. Additional analysis on memory formation changes, particularly in relation to ATR and various contributing factors.
2. Further investigation into the relationship between memory formation changes, plasticity, and connectivity.
3. Correction for the multiple comparison problem in the statistical analysis of connectivity in the boxel data.
4. A detailed explanation of the neuroscientific significance of connectivity changes in verbal and visual memory.
5. Presentation of figures to illustrate the results.

In conclusion, you have addressed my comments comprehensively, and I appreciate the additional analyses you performed beyond my initial suggestions. Regarding points 1 and 2, you appropriately constructed statistical models to examine the factors influencing memory formation changes and incorporated connectivity data to strengthen the credibility of the findings on memory formation changes and plasticity. For point 3, you suggested that the results remained significant even after applying FWE correction for multiple comparisons, which is commendable. In point 4, you provided further clarification on the dominant hemisphere associated with verbal and visual functions. Lastly, in point 5, you introduced appropriate figures to support the results effectively. I think the revisions have been executed appropriately, and I have only one minor suggestion.

- On page 8: **For visual memory, it states "38% improved, 15% no change, and 38% declined. The percentages sum to 91% (38+38+15)**, whereas the total should be 100%. Please verify and correct the values. Other than this, I have no additional comments. Once again, thank you for your revision, and I look forward to seeing the final version of this important research.

We thank you for your review and positive feedback on our revisions. We have carefully verified our statistics and corrected the values as suggested. The updated figures are now as follows, highlighted in *italics-bold* for clarity:

Results, Long-term changes in postoperative memory, Page 8:

“In right ATR, clinically significant changes were as follows: for verbal memory, 31% improved, 46% no change, and 23% declined. For visual memory, *42% improved, 17% no change, and 41% declined.*”

We hope this addresses your comment.

References

- Baxendale, S. and Thompson, P. (2020) 'The association of cognitive phenotypes with postoperative outcomes after epilepsy surgery in patients with temporal lobe epilepsy', *Epilepsy & Behavior*, 112, pp. 107386.
- Blampied, N. M. (2022) 'Reliable change and the reliable change index: still useful after all these years?', *The Cognitive Behaviour Therapist*, 15, pp. e50.
- Coughlan, A., Oddy, M. and Crawford, A. (2007) *BIRT Memory, and Information Processing Battery (BMIPB)*. London: Brain Injury Rehabilitation Trust.
- Drane, D. L., Ojemann, J. G., Kim, M. S., Gross, R. E., Miller, J. W., Faught Jr, R. E. and Loring, D. W. (2016) 'Interictal epileptiform discharge effects on neuropsychological assessment and epilepsy surgical planning', *Epilepsy & Behavior*, 56, pp. 131-138.
- Hermann, B. P., Seidenberg, M., Schoenfeld, J., Peterson, J., Leveroni, C. and Wyler, A. R. (1996) 'Empirical techniques for determining the reliability, magnitude, and pattern of neuropsychological change after epilepsy surgery', *Epilepsia*, 37(10), pp. 942-50.
- Wandschneider, B., Burdett, J., Townsend, L., Hill, A., Thompson, P. J., Duncan, J. S. and Koepp, M. J. (2017) 'Effect of topiramate and zonisamide on fMRI cognitive networks', *Neurology*, 88(12), pp. 1165-1171.
- Williamson, D., Drane, D., Stroup, E., Wilensky, S., Holmes, M. and Miller, J. 'Recent seizures may distort the validity of neurocognitive test scores in patients with epilepsy'. *Epilepsia*: BLACKWELL PUBLISHING 9600 GARSINGTON RD, OXFORD OX4 2DQ, OXON, ENGLAND, 74-74.
- Witt, J.-A., Elger, C. E. and Helmstaedter, C. (2015) 'Adverse cognitive effects of antiepileptic pharmacotherapy: Each additional drug matters', *European Neuropsychopharmacology*, 25(11), pp. 1954-1959.
- Xiao, F., Caciagli, L., Wandschneider, B., Joshi, B., Vos, S. B., Hill, A., Galovic, M., Long, L., Sone, D. and Trimmel, K. (2022) 'Effect of anti-seizure medications on functional anatomy of language: a perspective from language functional magnetic resonance imaging', *Frontiers in neuroscience*, 15, pp. 787272.
- Xiao, F., Caciagli, L., Wandschneider, B., Sander, J. W., Sidhu, M., Winston, G., Burdett, J., Trimmel, K., Hill, A., Vollmar, C., Vos, S. B., Ourselin, S., Thompson, P. J., Zhou, D., Duncan, J. S. and Koepp, M. J. (2018) 'Effects of carbamazepine and lamotrigine on functional magnetic resonance imaging cognitive networks', *Epilepsia*, 59(7), pp. 1362-1371.
- Yasuda, C. L., Centeno, M., Vollmar, C., Stretton, J., Symms, M., Cendes, F., Mehta, M. A., Thompson, P., Duncan, J. S. and Koepp, M. J. (2013) 'The effect of topiramate on cognitive fMRI', *Epilepsy Res*, 105(1-2), pp. 250-5.